

# Building consensus around the assessment and interpretation of Symbiodiniaceae diversity

Sarah W. Davies[1], Matthew H. Gamache[2], Lauren I. Howe-Kerr[3], Nicola G. Kriefall[1], Andrew C. Baker[4], Anastazia T. Banaszak[5], Line Kolind Bay[6], Anthony J. Bellantuono[7], Debashish Bhattacharya[8], Cheong Xin Chan[9], Danielle C. Claar[10], Mary Alice Coffroth[11], Ross Cunning[12], Simon K. Davy[13], Javier del Campo[14], Erika M. Díaz-Almeyda[15], Jörg C. Frommlet[16], Lauren E. Fuess[17], Raúl A. González-Pech[2,18], Tamar L. Goulet[19], Kenneth D. Hoadley[20], Emily J. Howells[21], Benjamin C. C. Hume[22], Dustin W. Kemp[23], Carly D. Kenkel[24], Sheila A. Kitchen[25], Todd C. LaJeunesse[26], Senjie Lin[27], Shelby E. McIlroy[28], Ryan McMinds[29], Matthew R. Nitschke[6], Clinton A. Oakley[13], Raquel S. Peixoto[30], Carlos Prada[31], Hollie M. Putnam[31], Kate Quigley[32], Hannah G. Reich[31], James Davis Reimer[33], Mauricio Rodriguez-Lanetty[7], Stephanie M. Rosales[34], Osama S. Saad[35], Eugenia M. Sampayo[36], Scott R. Santos[37], Eiichi Shoguchi[38], Edward G. Smith[39], Michael Stat[40], Timothy G. Stephens[8], Marie E. Strader[41], David J. Suggett[30,42], Timothy D. Swain[43], Cawa Tran[44], Nikki Traylor-Knowles[4], Christian R. Voolstra[22], Mark E. Warner[45], Virginia M. Weis[46], Rachel M. Wright[47], Tingting Xiang[39], Hiroshi Yamashita[48], Maren Ziegler[49], Adrienne M. S. Correa[3] and John Everett Parkinson[2]

[1] Department of Biology, Boston University, Boston, MA, United States
[2] Department of Integrative Biology, University of South Florida, Tampa, FL, United States
[3] Department of BioSciences, Rice University, Houston, TX, United States
[4] Rosenstiel School of Marine, Atmospheric, and Earth Science, University of Miami, Miami, FL, United States
[5] Unidad Académica de Sistemas Arrecifales, Universidad Nacional Autónoma de México, Puerto Morelos, Mexico
[6] Australian Institute of Marine Science, Townsville, Australia
[7] Department of Biological Sciences, Florida International University, Miami, FL, United States
[8] Department of Biochemistry and Microbiology, Rutgers University, New Brunswick, NJ, United States
[9] Australian Centre for Ecogenomics, School of Chemistry and Molecular Biosciences, The University of Queensland, Brisbane, QLD, Australia
[10] Nearshore Habitat Program, Washington State Department of Natural Resources, Olympia, WA, USA
[11] Department of Geology, University at Buffalo, Buffalo, NY, United States
[12] Daniel P. Haerther Center for Conservation and Research, John G. Shedd Aquarium, Chicago, IL, United States
[13] School of Biological Sciences, Victoria University of Wellington, Wellington, New Zealand
[14] Institut de Biologia Evolutiva (CSIC - Universitat Pompeu Fabra), Barcelona, Catalonia, Spain
[15] Department of Natural Sciences, New College of Florida, Sarasota, FL, United States
[16] Centre for Environmental and Marine Studies, Department of Biology, University of Aveiro, Campus Universitário de Santiago, Aveiro, Portugal
[17] Department of Biology, Texas State University, San Marcos, TX, United States
[18] Department of Biology, Pennsylvania State University, State College, PA, United States
[19] Department of Biology, University of Mississippi, University, MS, United States

Corresponding authors
Sarah W. Davies,
daviessw@gmail.com
Adrienne M. S. Correa,
ac53@rice.edu
John Everett Parkinson,
jparkinson@usf.edu

[20] Department of Biological Sciences, University of Alabama—Tuscaloosa, Tuscaloosa, AL, United States

[21] National Marine Science Centre, Faculty of Science and Engineering, Southern Cross University, Coffs Harbour, NSW, Australia

[22] Department of Biology, University of Konstanz, Konstanz, Germany

[23] Department of Biology, University of Alabama—Birmingham, Birmingham, Al, United States

[24] Department of Biological Sciences, University of Southern California, Los Angeles, CA, United States

[25] Division of Biology and Biological Engineering, California Institute of Technology, Pasadena, CA, United States

[26] Department of Biology, Pennsylvania State University, University Park, PA, United States

[27] Department of Marine Sciences, University of Connecticut, Mansfield, CT, United States

[28] Swire Institute of Marine Science, School of Biological Sciences, The University of Hong Kong, Pok Fu Lam, Hong Kong

[29] Center for Global Health and Infectious Disease Research, University of South Florida, Tampa, FL, United States

[30] Red Sea Research Center (RSRC), Division of Biological and Environmental Science and Engineering (BESE), King Abdullah University of Science and Technology, Thuwal, Saudi Arabia

[31] Department of Biological Sciences, University of Rhode Island, Kingston, RI, United States

[32] Minderoo Foundation, Perth, WA, Australia

[33] Department of Biology, Chemistry and Marine Sciences, Faculty of Science, University of the Ryukyus, Nishihara, Okinawa, Japan

[34] The Cooperative Institute For Marine and Atmospheric Studies, Miami, FL, United States

[35] Department of Biological Oceanography, Red Sea University, Port-Sudan, Sudan

[36] School of Biological Sciences, The University of Queensland, St. Lucia, QLD, Australia

[37] Department of Biological Sciences, University at Buffalo, Buffalo, NY, United States

[38] Marine Genomics Unit, Okinawa Institute of Science and Technology Graduate University, Okinawa, Japan

[39] School of Life Sciences, University of Warwick, Coventry, UK

[40] School of Environmental and Life Sciences, University of Newcastle, Callaghan, NSW, Australia

[41] Department of Biology, Texas A&M University, College Station, TX, United States

[42] Climate Change Cluster, University of Technology Sydney, Ultimo, NSW, Australia

[43] Department of Marine and Environmental Science, Nova Southeastern University, Dania Beach, FL, United States

[44] Department of Biology, University of San Diego, San Diego, CA, United States

[45] School of Marine Science and Policy, University of Delaware, Lewes, DE, United States

[46] Department of Integrative Biology, Oregon State University, Corvallis, OR, United States

[47] Department of Biological Sciences, Southern Methodist University, Dallas, TX, United States

[48] Fisheries Technology Institute, Japan Fisheries Research and Education Agency, Ishigaki, Okinawa, Japan

[49] Department of Animal Ecology & Systematics, Justus Liebig University Giessen (Germany), Giessen, Germany

## ABSTRACT

Within microeukaryotes, genetic variation and functional variation sometimes accumulate more quickly than morphological differences. To understand the evolutionary history and ecology of such lineages, it is key to examine diversity at multiple levels of organization. In the dinoflagellate family Symbiodiniaceae, which can form endosymbioses with cnidarians (*e.g.*, corals, octocorals, sea anemones, jellyfish), other marine invertebrates (*e.g.*, sponges, molluscs, flatworms), and protists (*e.g.*, foraminifera), molecular data have been used extensively over the past three decades to describe phenotypes and to make evolutionary and ecological inferences. Despite advances in Symbiodiniaceae genomics, a lack of consensus among

researchers with respect to interpreting genetic data has slowed progress in the field and acted as a barrier to reconciling observations. Here, we identify key challenges regarding the assessment and interpretation of Symbiodiniaceae genetic diversity across three levels: species, populations, and communities. We summarize areas of agreement and highlight techniques and approaches that are broadly accepted. In areas where debate remains, we identify unresolved issues and discuss technologies and approaches that can help to fill knowledge gaps related to genetic and phenotypic diversity. We also discuss ways to stimulate progress, in particular by fostering a more inclusive and collaborative research community. We hope that this perspective will inspire and accelerate coral reef science by serving as a resource to those designing experiments, publishing research, and applying for funding related to Symbiodiniaceae and their symbiotic partnerships.

**Subjects** Biodiversity, Evolutionary Studies, Marine Biology, Microbiology, Zoology
**Keywords** Symbiodiniaceae, Symbiosis, ITS2, Coral, Cnidarian, Species, Population, Community, Genetic diversity, Collaborative

# INTRODUCTION

Dinoflagellates in the family Symbiodiniaceae occupy multiple ecological niches on tropical, subtropical, and temperate reefs, ranging from species that are exclusively free-living to those that form symbioses with marine invertebrates (*LaJeunesse et al., 2018*). The biology of symbiotic Symbiodiniaceae has been a major research focus due to the integral role these mutualists play in the health of scleractinian corals and other marine invertebrates (*Glynn, 1996*; *Hughes et al., 2017*). Although many scleractinian coral species exhibit specificity for particular Symbiodiniaceae (*Baker, 2003*; *Hume et al., 2020*; *Thornhill et al., 2014*), some coral species and even individual coral colonies can associate with a diversity of algal symbionts (*Baker & Romanski, 2007*; *Silverstein, Correa & Baker, 2012*). Moreover, not all host-symbiont pairings are equally resistant or resilient to stress (*Abrego et al., 2008*; *Berkelmans & Van Oppen, 2006*; *Hoadley et al., 2019*; *Howells et al., 2013a*; *Sampayo et al., 2008*), and a change in symbiont community may enhance tolerance to future stress. Thus, efforts to characterize the genetic and functional diversity within Symbiodiniaceae not only advances our fundamental knowledge of the evolution and ecology of microeukaryotes, but also provides insights into the potential for cnidarian-Symbiodiniaceae partnerships, and ultimately for coral reefs, to respond to rapidly changing environments.

The first "*Symbiodinium*" species was formally described by *Freudenthal (1962)*. As more associations with these endosymbiotic dinoflagellates were cataloged, the utility of allozymes (*Schoenberg & Trench, 1980*) and later ribosomal markers (*LaJeunesse, 2001*; *Rowan & Powers, 1991*) to distinguish different lineages became apparent. Continued exploration of Symbiodiniaceae diversity through molecular genetics ultimately resulted in a recent systematic revision, delineating at least eleven genera and many species (*LaJeunesse et al., 2021*, *2018*; *Nitschke et al., 2020*; *Pochon & LaJeunesse, 2021*). However, despite numerous advances in our ability to resolve Symbiodiniaceae populations, often

allowing for genus, species, or even strain level identification (*Thornhill et al., 2017*), diversity assessments pose substantial challenges (Fig. 1). For example, Symbiodiniaceae density often exceeds 1–2 million cells *per* square centimeter of host tissue (*Fitt et al., 2000*). Further, hosts may associate with a single species or a mixture of multiple species and/or genera (*Baker & Romanski, 2007*; *Coffroth et al., 2010*; *Kemp, Fitt & Schmidt, 2008*; *Rowan & Knowlton, 1995*; *Thornhill et al., 2017*, *2006*; *van Oppen et al., 2005*). In addition, Symbiodiniaceae have expansive genomes (~1–5 Gbp; *Saad et al., 2020*), often including multi-copy genes and extensive gene duplication (*Lin, 2011*; *González-Pech et al., 2021*). Therefore, many approaches to resolve Symbiodiniaceae taxonomy rely on multi-copy gene markers. For example, the multi-copy internal transcribed spacer 2 (ITS2) rDNA region is most frequently used to resolve Symbiodiniaceae lineages, yet data generated by this marker straddle intergenomic and intragenomic variation (the latter of which is abbreviated as IGV), limiting its utility for some applications (*Smith, Ketchum & Burt, 2017*). This issue has fueled an active debate within the research community regarding the interpretation of ITS2 molecular data and likely contributed to underuse of other molecular markers, even though they may be more appropriate in some contexts (*LaJeunesse & Thornhill, 2011*; *Takishita et al., 2003*).

Indeed, the increasing popularity of amplicon-sequencing methods (*Arif et al., 2014*; *Green et al., 2014*; *Howe-Kerr et al., 2020*; *Hume et al., 2019*; *Quigley et al., 2014*), exploration of additional molecular markers (*Pochon et al., 2019*, *2012*; *Smith et al., 2020*; *Takabayashi, Santos & Cook, 2004*), and incorporation of whole-genome datasets (*Dougan et al., 2022*; *González-Pech et al., 2021*; *Liu et al., 2018*) have led to novel insights into Symbiodiniaceae diversity. However, such advances have led to additional challenges. For example, most genetic loci exhibit differential utility across Symbiodiniaceae genera (*Pochon, Putnam & Gates, 2014*). Furthermore, different analytical pipelines and thresholds applied to the same marker(s) among studies have led to different estimates of genetic variation and interpretation of their functional importance (*Cunning, Gates & Edmunds, 2017*; *Howells et al., 2016*; *Wham & LaJeunesse, 2016*; *Wirshing & Baker, 2016*). These issues have further fueled the debate around which markers to use and how to interpret the resulting data.

Recognizing that continued debate may complicate the process of scientific inquiry, we sought to identify areas of consensus regarding the assessment and interpretation of Symbiodiniaceae genetic diversity. Sixty-one scientists from 12 countries, spanning expertise in the taxonomy, physiology, genomics, and ecology of Symbiodiniaceae and other marine microbes, participated in a workshop funded by the National Science Foundation titled "Building consensus around the quantification and interpretation of Symbiodiniaceae diversity," held virtually in July 2021. The overall aim was to reduce barriers to those designing experiments, publishing research, and applying for funding related to Symbiodiniaceae and their partnerships. The major workshop outcomes are summarized herein, though not exhaustively. We highlight techniques that are broadly accepted by many experts in the field and point out caveats and considerations for these approaches (Box 1). Where agreement was not reached, we identify the key issues that remain unresolved and point to technologies that might help fill knowledge gaps so that

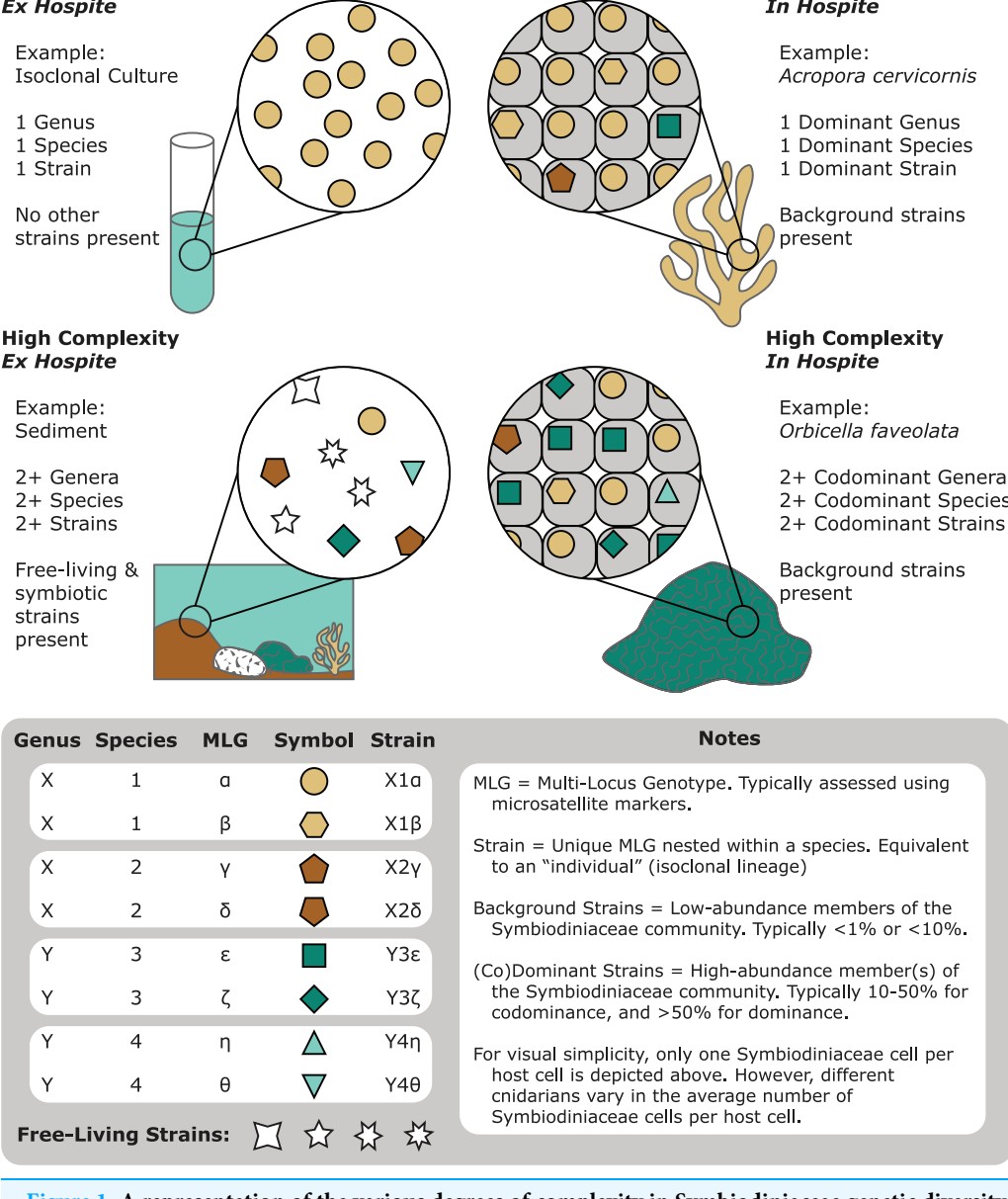

**Figure 1  A representation of the various degrees of complexity in Symbiodiniaceae genetic diversity among different habitats (*e.g.*, cultures, corals, and sediments).** Communities of Symbiodiniaceae within a given sample can encompass multiple strains, populations, species, and genera.

consensus can be achieved in the future. We conclude with suggestions for how to make the Symbiodiniaceae research community a more inclusive and welcoming space that promotes innovation as we navigate the coral reef crisis. Above all, we wish to stress that the choice of genetic marker(s) and analytical framework(s) for interpreting Symbiodiniaceae diversity will always depend on the research question at hand, along with the availability of resources (*e.g.*, for sample preservation, processing, and computation),

**Box 1 Major workshop outcomes and consensus highlights.**

**General**

a. Different research questions require different levels of resolution of Symbiodiniaceae diversity (*e.g.* species, populations, communities).

b. Molecular markers evolve at different rates and vary in their ability to resolve different Symbiodiniaceae taxonomic levels and lineages, requiring careful selection of the appropriate marker(s) for a given question.

c. Many markers and analytical approaches are available, each with their own strengths and weaknesses. As genomic resources for Symbiodiniaceae continue to be developed and technologies advance, so will options for analyzing and interpreting diversity.

d. Collaborations among research groups can ameliorate methodological and analytical disconnect within the Symbiodiniaceae community, while also reducing costs associated with answering complex and integrative research questions.

**1. Species-Level Assessment of Symbiodiniaceae Diversity**

a. Resolving Symbiodiniaceae to the species level is important. Species identification forms the basis of comparative physiological, ecological, and evolutionary investigations within Symbiodiniaceae.

b. A robust Symbiodiniaceae taxonomy is required to facilitate scientific communication, link past and future research, and establish legal frameworks for conservation. Funding to develop and maintain up-to-date public taxonomic tools and databases should be increased.

c. There are distinctions between *describing* a new species, *recognizing* a new species, and *identifying* a known species. Reef researchers benefit from incorporation and consideration of the current taxonomy whenever possible.

d. Expanding publicly accessible Symbiodiniaceae culture collections and their formal genetic, morphological, and physiological description will drive taxonomic, ecological, physiological, and genomic research. Supporting these resources for use by the scientific community should be a priority for long-term funding.

**2. Population-Level Assessment of Symbiodiniaceae**

a. Population-level studies evaluate the distribution of genetic variation within Symbiodiniaceae species, often across spatiotemporal gradients or among host taxa, to understand the influence of evolutionary processes such as gene flow, genetic drift, and natural selection.

b. When multiple Symbiodiniaceae lineages are present within host colonies, population-level questions are more challenging to address.

c. Pre-screening to determine which lineages are present within samples is necessary to determine the marker(s) needed to address population-level questions in Symbiodiniaceae.

d. Microsatellite loci can be effective at addressing population-level questions in Symbiodiniaceae if used appropriately.

e. The ITS2 region of Symbiodiniaceae rDNA may be an effective marker for distinguishing between different populations, but requires thorough validation to distinguish intra- and inter-genomic variation.

**3. Community-Level Assessment of Symbiodiniaceae**

a. Symbiodiniaceae communities can be conceptualized at different scales. The presence of two or more Symbiodiniaceae species within a host individual constitutes a "local community." Symbiodiniaceae diversity at larger scales (*e.g.*, among conspecific host colonies, multiple host species, or across environmental pools such as sediments and the water column including free-living Symbiodiniaceae) constitutes a "macroscale community." The total diversity of both local and macroscale communities is likely underestimated.

b. Local Symbiodiniaceae communities are often composed of representatives of different genera, rather than multiple species or lineages within the same genus.

c. Marker genes that exhibit inter- and intra-genomic variation (as well as variation in copy number across lineages) make it challenging to characterize Symbiodiniaceae community composition. Quantifying this molecular variation for Symbiodiniaceae genera and species is a priority.

d. The Symbiodiniaceae ITS2 marker can be useful for describing Symbiodiniaceae communities but there are circumstances where multiple markers or other approaches may be more appropriate. The majority of researchers at the workshop reported greatest familiarity and comfort with the ITS2 marker, which may have contributed to its popularity in characterizing Symbiodiniaceae communities.

e. There is a lack of consensus regarding best practices for interpreting Symbiodiniaceae gene amplicon data to identify species, and for applying and interpreting community diversity metrics. Authors are encouraged to clearly highlight assumptions associated with their data interpretation, acknowledge that other interpretations exist, and discuss whether or not alternative interpretations change the biological or ecological findings of their study.

**Box 1** (continued)

**4. Beyond Genotype: Phenotyping Symbiodiniaceae**

a. Phenotypic diversity varies greatly within and between Symbiodiniaceae species, thus it is critical to avoid overestimating the functional significance of a given symbiont based on taxonomic assignment alone (*e.g.*, assuming that all *Durusdinium* spp. are heat-tolerant).

b. There is a need to develop technologies to functionally assess Symbiodiniaceae in culture, *in hospite*, and in the environment–and to better contextualize the resulting phenotypes–with the understanding that functional diversity will vary depending on the metrics used.

c. When attempting to understand phenotypic variability among strains and species, using cultures of Symbiodiniaceae can help control confounding variables. However, because cultures are, by nature, artificial environments, performance *in vitro* may differ from performance *in hospite*, and many species are difficult to culture.

**5. Integrating Multiomic Technologies to Study Symbiodiniaceae**

a. Various 'omics techniques have been used to address Symbiodiniaceae taxonomic, functional, and physiological research questions. Because each technique has unique considerations, leveraging these novel tools requires stringent ground-truthing and the development of quality standards.

b. Genome projects have improved tremendously over the past decade, but there are unique biological obstacles that have restricted Symbiodiniaceae genome assembly quality. Examples include large genome sizes, high repeat content, and difficulty annotating gene functions.

c. Integrating multiple techniques, such as transcriptomics and proteomics, and coupling these with phenotyping methods, can help answer outstanding questions regarding Symbiodiniaceae-host interactions. Efficient experimentation will require combining expertise across laboratories.

**6. Ensuring an Inclusive Symbiodiniaceae Research Community**

a. Critical examination is at the heart of scientific inquiry. A diversity of perspectives has always been and will continue to be needed to move the Symbiodiniaceae field forward.

b. The publication process should be equitable. Recommendations to journals and scientific societies include increasing diversity on relevant editorial boards, scaling publication costs for researchers employed in countries with lower income economies, and implementing double-blind review. Researchers should actively cite articles led by diverse colleagues.

c. Parachute science should be avoided. Recommendations include fostering long-term international collaborations and exchange programs to involve local scientists in Symbiodiniaceae research, improving sensitivity to the challenges facing colleagues in funding-limited partner institutions, and extending full collaborative benefits including authorship and grant writing opportunities to these colleagues.

d. Accessibility and collaboration should be fostered. Recommendations include expanding a recently established database of Symbiodiniaceae researchers and their research products, maintaining hybrid format options for conferences, and supporting long-term funding for international collaborations.

e. It is critical to improve recruitment, retention, and promotion of scholars of diverse backgrounds. Recommendations include working actively to increase diversity at all levels of academia and science, promoting the work of minority scientists, and providing strong multidimensional mentorship to support and retain these scientists throughout each career stage.

and that these options will inevitably evolve as our understanding of the system continues to develop.

# GUIDANCE FOR SPECIES-LEVEL ASSESSMENT OF SYMBIODINIACEAE DIVERSITY

## Why is species-level resolution important for Symbiodiniaceae?

Species are evolutionarily independent lineages and therefore represent a fundamental level of biological organization. Species-level resolution provides insight into the ecological and evolutionary mechanisms that create diversity, and forms the basis of comparative physiological investigations (*Kareiva & Levin, 2015*). The delineation of species can affect nearly all scales of inquiry, from biochemical pathways to ecosystem processes. Species-level diversity in Symbiodiniaceae has been discussed in the literature since the

description of *Symbiodinium microadriaticum* in 1962 by *Freudenthal (1962)*. As more diversity was uncovered and more species were recognized (*LaJeunesse, 2001*; *LaJeunesse & Trench, 2000*; *Rowan & Powers, 1992*, *1991*; *Schoenberg & Trench, 1976*; *Trench & Blank, 1987*), controversy arose as to where to draw species boundaries (*Apprill & Gates, 2007*; *Correa & Baker, 2009*; *Cunning, Glynn & Baker, 2013*; *LaJeunesse et al., 2014*; *LaJeunesse, Parkinson & Reimer, 2012*; *Stat et al., 2012*; *Thornhill, LaJeunesse & Santos, 2007*; *Wham & LaJeunesse, 2016*). At present, there is general consensus among Symbiodiniaceae specialists about the need for species-level resolution, as well as support for current taxonomic methodologies that are underpinned by genetic, ecological, and morphological data (*LaJeunesse et al., 2018*; *Voolstra et al., 2021b*). Such taxonomic descriptions facilitate scientific communication and are necessary for establishing legal frameworks for conservation (*IUCN, 2021*). The recent elevation of most Symbiodiniaceae "Clades" to genera has provided some clarity (*LaJeunesse et al., 2021*, *2018*; *Nitschke et al., 2020*; *Pochon & LaJeunesse, 2021*), but the small number of formal species-level descriptions for the large genetic diversity found within most Symbiodiniaceae genera constitutes a formidable barrier to progress.

Without robust species delineation, functional differences can inadvertently be ascribed to incorrect taxonomic levels or non-existent biological entities. For example, the genus level may be too coarse and lead to over-generalizations regarding the physiology or function of Symbiodiniaceae variants (see "Beyond Genotype: Phenotyping Symbiodiniaceae"). A statement such as "*the genus* Cladocopium *consists of heat-sensitive species*" overlooks the superior stress tolerance of some *Cladocopium* species, including the dominance of *Cladocopium thermophilum* in corals on some of the world's hottest reefs in the Persian/Arabian Gulf (*Abrego et al., 2008*; *Hume et al., 2015*; *Varasteh et al., 2018*). However, diversity assessments based on gene sequence variants may recover both interspecific variation (resolving distinct species) and intraspecific variation (sequence diversity within a single genome). This is a major issue for the commonly used multi-copy ITS2 gene. Consequently, a statement such as "*Symbiodiniaceae harboring the ITS2 D13 sequence variant are adapted to temperate environments*" overlooks the fact that ITS2 sequence variants D8, D8–12, D12–13, and D13 are all characteristic of the same species, *Durusdinium eurythalpos* (*LaJeunesse et al., 2014*). The statement could give a false impression that entities harboring the D8 variant are phylogenetically and ecologically distinct from those harboring D13. In this scenario, because we know the ITS2 profile of *D. eurythalpos*, we can clarify that the four sequence variants belong to the same species. However, for many undescribed species, the profiles are not yet resolved. Such issues are problematic because they may confuse ecological interpretations of sequence data, particularly in datasets composed of communities of different symbiont species where some consist of overlapping ITS2 intragenomic variants.

## What types of data can identify Symbiodiniaceae species?

Although taxonomic descriptions are fundamental, *describing* a new species is not the same as *recognizing* a new species or *identifying* a known species. *Describing* should be based on multiple lines of evidence, whereas *recognizing* or *identifying* may require

generating and interpreting data from only one or two diagnostic methods. At minimum, there are six major components of a valid Symbiodiniaceae species description: (1) information on at least two congruent genes (see our recommendations below in "How can we Resolve Symbiodiniaceae Species with Genetic Markers?"), (2) comparison of genetic data against that from other Symbiodiniaceae, (3) morphological description (*e.g.*, comparison of cell size measurements against that from other Symbiodiniaceae), (4) a holotype or name-bearing type specimen (at minimum, an image of cells under light microscopy, but preserved cells are preferable), (5) deposition of the type specimen in a permanent archive (*e.g.*, a museum or herbarium for preserved cells, but if only images are available, their publication in a peer-reviewed journal is sufficient), and (6) proposition of a valid name (according to the International Code of Nomenclature for Algae, Fungi, and Plants; *Turland et al., 2018*). Where possible, ecological descriptions such as host associations and biogeographic ranges are also encouraged, although sometimes such information is not available.

The Biological Species Concept dictates that if two organisms cannot reproduce and create viable offspring, they should be considered different species (*Mayr, 1942*). Unfortunately, it has been impossible to apply this criterion to Symbiodiniaceae, as no direct observation of sexual reproduction has been made to date (but see *Figueroa, Howe-Kerr & Correa, 2021*; *Shah et al., 2020*). Fortunately, many other species concepts exist, each placing emphasis on different criteria (*De Queiroz, 2007*; *Leliaert et al., 2014*). Robust species descriptions satisfy multiple species concepts using independent lines of evidence. For Symbiodiniaceae, the field has largely applied three key types of data: morphological (cell size and cell wall features), ecological (host specificity and biogeographic distribution), and phylogenetic (divergence across multiple DNA markers), along with the assignment of type material (Fig. 2). The taxonomic framework for describing species has matured since the earliest effort by *Freudenthal (1962)*. For example, in line with the Morphological Species Concept, *Trench & Blank (1987)* proposed three new species based on Symbiodiniaceae cell ultrastructure. They used transmission electron microscopy (TEM) to reveal features such as the nucleus, chromosomes, pyrenoid, chloroplast thylakoid membranes, and cell size; additionally, they used scanning electron microscopy (SEM) to observe thecal plates and the arrangement of the two flagella. Technological advancements in SEM resolution now enable complete morphological characterization of amphiesmal vesicles in the cell wall (*Jeong et al., 2014*; *LaJeunesse, Lee & Gil-Agudelo, 2015*; *Lee, Jeong & LaJeunesse, 2020*; *Lee et al., 2015*; *Nitschke et al., 2020*), though such plate tabulations tend to be variable within species (*LaJeunesse et al., 2018*).

As an increasing number of host species are sampled, it has become clear that the Ecological Species Concept can also be used to support Symbiodiniaceae species descriptions. Although not diagnostic in all cases (*Cunning, Glynn & Baker, 2013*), symbiosis ecology can be particularly useful for Symbiodiniaceae species that exhibit host-specificity or coadaptation with their hosts (*Davies et al., 2020*; *Finney et al., 2010*; *Howells et al., 2020*; *Santos et al., 2004*; *Smith, Ketchum & Burt, 2017*; *Thornhill et al., 2014*). For example, *Cladocopium pacificum* and *Cladocopium latusorum* are found exclusively within corals of the genus *Pocillopora* (*Turnham et al., 2021*). Ultimately,

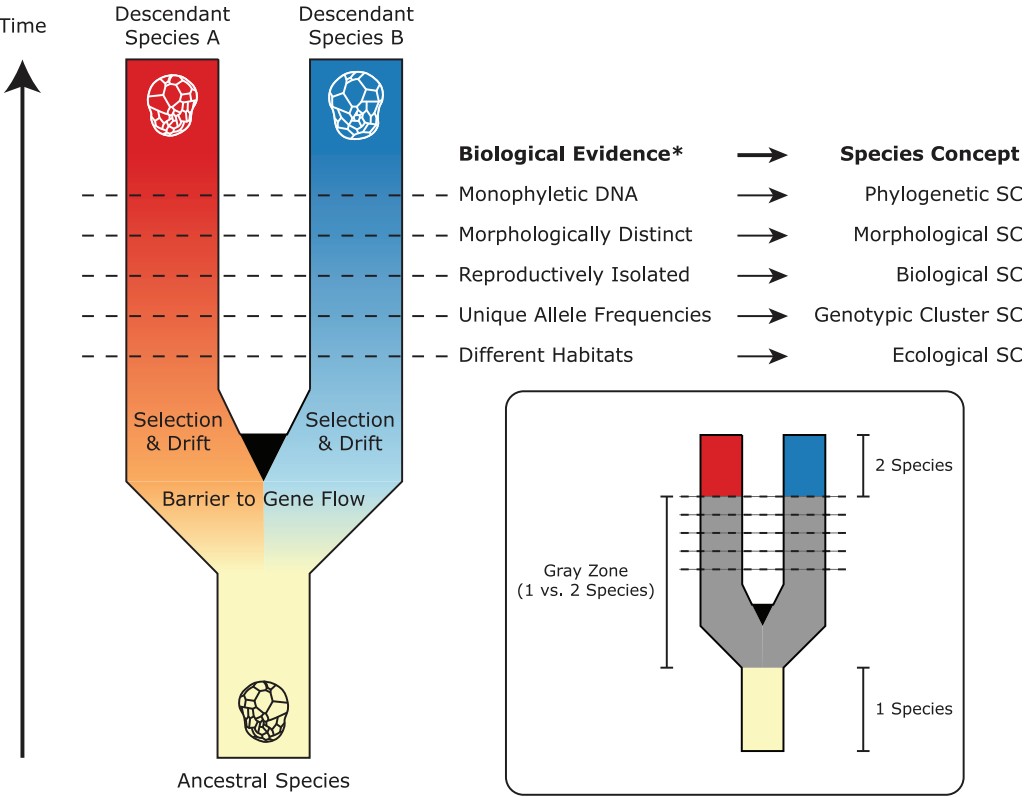

*Biological evidence in the Gray Zone may arise with different timing and order than depicted.

**Figure 2 A simplified representation of Symbiodiniaceae speciation, species concepts (SC), and associated biological evidence.** In this example, one ancestral species splits and diverges to become two descendant lineages after barriers to gene flow are established. Through selection and drift, these lineages evolve different properties, which satisfy the criteria of different species concepts (represented by horizontal lines). Because these properties may arise at different times and in different orders, there is a "gray zone" where conflict among species concepts may result in controversy about whether one or two species exist. Modified from *De Queiroz (2007)* and *Leliaert et al. (2014)*.

because Symbiodiniaceae do not always have distinct morphological characteristics, nor do they always exhibit host specificity, the collection of genetic data to satisfy the Phylogenetic Species Concept has also become a necessity in the description of species (see "How can we Resolve Symbiodiniaceae Species with Genetic Markers?").

Researchers who do not endeavor to describe Symbiodiniaceae species can encourage and incentivize those who do by accurately treating taxa names as hypotheses and citing the work of taxonomists at the first mention of previously described taxa within manuscripts. We encourage incorporating existing taxonomy (*i.e.*, species names) into current research whenever possible. Due to a general lack of funding for taxonomic descriptions, formal species names are not always available for a given entity, and therefore accommodating sequence variant terminology in the literature will continue to be important. Providing synonyms (*e.g.*, the ITS2 sequence variant *and* its species name) when a species is first mentioned will improve clarity. Ensuring that community resources consolidate current and past taxonomic assignments will be challenging, but it is critical

for connecting historical and future research. Guidance for vouchering Symbiodiniaceae genomic data sets from cultures or holobiont tissues has recently been put forth (*Voolstra et al., 2021b*). Minimum recommendations include (but are not limited to) high quality DNA voucher material, comprehensive metadata, and common phylogenetic marker sequences. Work is underway to develop a robust 'Rosetta Stone' that can translate between different marker designations and species names. Such efforts should be expanded in the future (*e.g.*, through incorporation into analysis pipelines for molecular data) to better facilitate efficient Symbiodiniaceae identification in complex samples.

### How can we resolve Symbiodiniaceae species with genetic markers?

No single marker is likely able to distinguish species across all Symbiodiniaceae genera reliably (Table 1). Instead, ecological and physiological studies will benefit from adopting a multi-gene approach where possible, given funding and resource limitations (see "Guidance for Community-Level Assessment of Symbiodiniaceae"). Congruence among sequence data from different cellular compartments (nuclear, chloroplast, and mitochondrial; Table 1) indicates that classifying Symbiodiniaceae using a lineage-based species concept is achievable (*De Queiroz, 2007*; *LaJeunesse & Thornhill, 2011*; *Sampayo, Dove & LaJeunesse, 2009*). This multi-gene approach, supported with ecological, morphological, and sometimes physiological data, has led to the formal description (or re-validation) of 39 Symbiodiniaceae species in 11 genera thus far (Table 1; *Hume et al., 2015*; *Jeong et al., 2014*; *LaJeunesse, 2017*; *LaJeunesse et al., 2021*, *2018*; *LaJeunesse, Lee & Gil-Agudelo, 2015*; *LaJeunesse et al., 2014*; *LaJeunesse, Parkinson & Reimer, 2012*; *Lee, Jeong & LaJeunesse, 2020*; *Lewis, Chan & LaJeunesse, 2019*; *Nitschke et al., 2020*; *Parkinson, Coffroth & LaJeunesse, 2015*; *Pochon & LaJeunesse, 2021*; *Ramsby et al., 2017*; *Turnham et al., 2021*; *Wham, Ning & LaJeunesse, 2017*; *Xiang et al., 2013*). This taxonomic list will continue to grow, and recent whole-genome data already point toward the potential need for further revision of some genera and species (*Dougan et al., 2022*; *González-Pech et al., 2021*).

The rate of evolution of gene markers dictates their respective power to resolve distinct genetic entities and whether these entities are likely to represent distinct species (Table 1). In addition, genetic differentiation may vary among genera for the same marker region (*Pochon, Putnam & Gates, 2014*). Efforts are underway to develop a taxonomic key for Symbiodiniaceae species based on genetic and ecological data. We envision a dynamic dichotomous key that would guide users to the appropriate markers and characteristics for a particular host organism of interest, or alternatively, suggest combinations of markers and characteristics most likely to provide species-level resolution within specific sets of closely related Symbiodiniaceae. Such a key would also reduce project costs by identifying the most informative minimal set of markers.

### How many Symbiodiniaceae species exist?

The current best estimate for the total number of symbiotic Symbiodiniaceae species is in the range of hundreds based on phylogenetic (*e.g.*, ITS2) sequence variants (*Thornhill et al., 2014*). However, these species numbers are likely a significant underestimate because

**Table 1 Current list of formally described Symbiodiniaceae species and associated diagnostic information.**

| Genus (11 total) | Species (39 total) | ITS2 Type (majority sequence) | Symbiotic? | Cultured? | Host-specificity | Cell Size | Plate tabulation | Nuclear | | | Mito | | Chloro | |
|---|---|---|---|---|---|---|---|---|---|---|---|---|---|---|
| | | | | | | | | ITS2 | LSU | msat flankers | mtCOB | mtCOX1 | cp23S | psbA$^{ncr}$ |
| Breviolum | B. aenigmaticum | – | U | Y | U | ND | X | D | D | D | ND | X | D | X |
| Breviolum | B. antillogorgium | B1 | Y | Y | D | ND | X | ND | ND | D | ND | X | D | X |
| Breviolum | B. dendrogyrum | B1k, B1 | Y | N | D | ND | X | D** | ND | D | ND | X | ND | D |
| Breviolum | B. endomadracis | B7 | Y | N | D | ND | X | D | ND | D | D | X | ND | D |
| Breviolum | B. faviinorum | B14, B14a, B1 | Y | N | ND | ND | X | D** | ND | D | ND | X | ND | D |
| Breviolum | B. meandrinium | B20, B1 | Y | N | ND | ND | X | D** | ND | D | ND | X | ND | D |
| Breviolum | B. minutum | B1 | Y | Y | ND | ND | X | ND | ND | D | ND | X | ND | D |
| Breviolum | B. pseudominutum | B1 | U | Y | U | ND | X | ND | ND | D | D | X | ND | X |
| Breviolum | B. psygmophilum | B2 | Y | Y | ND | ND | X | D | D | D | ND | X | D | X |
| Cladocopium | C. goreaui | C1 | Y | Y | ND | Y | ND* | ND | ND | X | D | D | ND | D |
| Cladocopium | C. infistulum | C2 | Y | Y | D | Y | ND* | D | D | X | D | D | D | X |
| Cladocopium | C. latusorum | C1b-c, C42a, C42a-b, C1c-ff | Y | N | D | Y | X | D | D | D | ND | X | X | D |
| Cladocopium | C. pacificum | C1d, C1d-t | Y | N | D | Y | X | D | D | D | ND | X | X | D |
| Cladocopium | C. thermophilum | C3 | Y | N | ND | U | X | D | ND | X | D | D | ND | D |
| Durusdinium | D. glynnii | D1, D1-4-6 | Y | N | ND | ND | X | D** | ND | D | ND | X | ND | D |
| Durusdinium | D. boreum | D15 | Y | N | ND | ND | X | D | D | D | D | X | ND | X |
| Durusdinium | D. eurythalpos | D8, D12-13, D13 | Y | N | ND | ND | X | D | D | D | ND | X | ND | X |
| Durusdinium | D. trenchii | D1a, D1-4, D1-4-6 | Y | Y | ND | ND | X | D** | ND | D | ND | X | D | D |
| Effrenium | E. voratum | E1 | N | Y | NA | M | M | M | M | X | M | X | M | M |
| Freudenthalidium | Fr. endolithicum | F3.8 | U | Y | X | D | ND | R | R | X | R | X | R | X |

# Table 1 (continued)

| Genus (11 total) | Species (39 total) | ITS2 Type (majority sequence) | Symbiotic? | Cultured? | Host-specificity | Cell Size | Plate tabulation | Nuclear | | | Mito | | Chloro | |
|---|---|---|---|---|---|---|---|---|---|---|---|---|---|---|
| | | | | | | | | ITS2 | LSU | msat flankers | mtCOB | mtCOX1 | cp23S | psbA$^{ncr}$ |
| Freudenthalidium | Fr. heronense | F3.7 | U | Y | X | D | ND | R | R | X | R | X | R | X |
| Fugacium | Fu. kawagutii | F1 | U | Y | U | M | X | M | M | X | X | X | X | X |
| Gerakladium | G. endoclionum | – | Y | N | D | ND | X | X | D | X | ND | X | R | R |
| Gerakladium | G. spongiolum | – | Y | N | D | ND | X | X | ND | X | ND | X | R | R |
| Halluxium | H. pauxillum | H7 | U | Y | X | M | M | R | R | X | ND | X | R | X |
| Miliolidium | M. leei | D1.1 | U | Y | U | M | X | R | R | X | R | R | X | X |
| Philozoon | P. actiniarum | A19 | Y | N | D | D | X | D | ND | X | D | ND | ND | R |
| Philozoon | P. adriaticum | – | Y | N | D | D | X | ND | D | X | ND | ND | ND | R |
| Philozoon | P. anthopleurum | – | Y | N | D | D | X | ND | ND | X | ND | ND | ND | R |
| Philozoon | P. balanophyllum | – | Y | N | D | D | X | ND | ND | X | ND | D | ND | R |
| Philozoon | P. colossum | – | Y | N | D | D | X | ND | D | X | ND | D | D | R |
| Philozoon | P. geddesianum | – | Y | N | D | D | X | ND | ND | X | ND | D | D | R |
| Philozoon | P. medusarum | – | Y | N | D | D | X | ND | ND | X | ND | ND | ND | R |
| Philozoon | P. paranemonium | – | Y | N | D | D | X | ND | ND | X | ND | D | ND | R |
| Symbiodinium | S. microadriaticum | A1 | Y | Y | ND | ND | ND | D | D | X | D | X | ND | D |
| Symbiodinium | S. natans | – | N | Y | U | ND | ND | D | D | X | D | X | D | X |
| Symbiodinium | S. necroappetens | A13 | Y*** | N | ND | ND | ND | D | D | X | D | X | ND | D |
| Symbiodinium | S. pilosum | A2 | N | Y | U | ND | X | D | D | X | D | X | D | X |
| Symbiodinium | S. tridacnidorum | A6, A3a, A3* | Y | Y | ND | ND | ND | D** | D | X | D | X | D | X |

**Notes:**

Mito, Mitochondrial; Chloro, Chloroplast; R,Resolves all species within the genus; D, Diagnostic (uniquely differentiates a particular species of the genus); ND, Not diagnostic (sequence/trait identical in two or more species); M, Measured but lacking congenerics or reference material for comparison; X, Not used in species description; U, Unknown (*e.g.*, sampled from a symbiotic habitat but not necessarily likely to be the numerically dominant symbiont); Y, Yes; N, No; NA, Not Applicable; ND*, Not diagnostic of species, but lack of elongated amphiesmal vesicles is diagnostic of Cladocopium; D**, Some ITS2 sequences may be diagnostic, but others in the in the same genome may not be; Y***, Opportunistic and occurring at background levels unless host health is compromised.

For an extended version of the table that includes authentic cultured strains, synonyms, and key references for each species, see Table S1.

sampling efforts have mainly focused on scleractinian coral hosts living at shallow depths in tropical and subtropical waters. It will be important to continue describing Symbiodiniaceae species in non-scleractinian hosts, including other cnidarians; *e.g.*, octocorals (*Goulet et al., 2017*; *Ramsby et al., 2014*), zoantharians (*Fujiwara et al., 2021*; *Mizuyama et al., 2020*), actiniarians (*Grajales, Rodríguez & Thornhill, 2016*), corallimorpharians (*Kuguru et al., 2008*; *Jacobs et al., 2022*), hydrocorals (*Rodríguez et al., 2019*), jellyfish (*Vega de Luna et al., 2019*); as well as sponges (*Hill et al., 2011*; *Ramsby et al., 2017*), acoelomorph flatworms (*Kunihiro & Reimer, 2018*), molluscs (*Baillie, Belda-Baillie & Maruyama, 2000*; *Banaszak, García Ramos & Goulet, 2013*; *Lim et al., 2019*), ciliates (*Mordret et al., 2016*), and foraminifera (*Pochon et al., 2007*). Further collections from undersampled habitats and sources such as benthic sediment and rubble (*Fujise et al., 2021*; *Nitschke et al., 2020*; *Sweet, 2014*; *Takabayashi et al., 2012*), seagrasses and macroalgae (*Porto et al., 2008*; *Yamashita & Koike, 2013*), mesophotic depths (*Frade et al., 2008*; *Goulet, Lucas & Schizas, 2019*), the water column (*Manning & Gates, 2008*; *Pochon et al., 2010*; *Sweet, 2014*), and predator feces (*Castro-Sanguino & Sánchez, 2012*; *Grupstra et al., 2021*; *Parker, 1984*) will likely yield many undiscovered species and possibly even novel genera (*Yorifuji et al., 2021*). These efforts should not be limited to subtropical and tropical waters, as Symbiodiniaceae have been reported in more temperate locations (*LaJeunesse et al., 2021*; *Lien, Fukami & Yamashita, 2012*). Systematic and wide-ranging effort to better describe the genetic diversity of Symbiodiniaceae (such as the Tara Oceans expedition; *Sunagawa et al., 2020*) will lead to a better understanding of the drivers of taxonomic and functional diversity of Symbiodiniaceae.

## What steps can be taken to enhance our understanding of Symbiodiniaceae species?

Expanding publicly accessible Symbiodiniaceae culture collections can drive not only taxonomic but also ecological, physiological, and genomic research (*LaJeunesse et al., 2018*; *Voolstra et al., 2021b*; *Xiang et al., 2013*). Most of the diversity in culture constitutes just a handful of species, predominantly from the *Symbiodinium* and *Breviolum* genera. More targeted and consistent funding to support further development, maintenance, and sharing of culture collections is critical to the field. Progress toward protocols for Symbiodiniaceae cryopreservation can help conserve biodiversity through the generation of cryogenic archives (*Di Genio et al., 2021*) and support research in laboratories that cannot maintain continuous cultures. Depositing live specimens in national and organizational archives can alleviate the burden on individual research groups. Examples of national archives include the Provasoli-Guillard National Center for Marine Algae and Microbiota at Bigelow Laboratory in the USA, (https://ncma.bigelow.org/), the Symbiont Culture Facility at the Australian Institute of Marine Science in Australia (https://www.aims.gov.au/), the National Institute for Environmental Studies, (https://mcc.nies.go.jp/) and Biological Resource Center at National Institute of Technology and Evaluation (https://www.nite.go.jp/nbrc/catalogue/) in Japan, the Central Collection of Algal Cultures in Germany (https://www.uni-due.de/biology/ccac/), the Roscoff Culture Collection in France (https://roscoff-

culture-collection.org/), and the Culture Collection of Algae and Protozoa in the United Kingdom (https://www.ccap.ac.uk/).

Live cultures established from single cells can benefit taxonomic studies by providing relatively homogeneous strains to establish baselines of diversity and morphology. Monocultures can be confirmed molecularly through fragment analysis of microsatellites. As they are haploid, Symbiodiniaceae monocultures should only show single microsatellite peaks, except in taxa with evidence for broad duplications, such as *Durusdinium trenchii* (*LaJeunesse et al., 2014*). Molecular data from cultured isoclonal strains are less noisy than data from host tissues, since such tissues may contain multiple Symbiodiniaceae genera, species, or strains (Fig. 1; *Voolstra et al., 2021b*). Cultures are also superior for holotype depositions, and they facilitate morphometric analysis, for example, on swimming behavior (motility). However, live culture is not a prerequisite for formal species description, especially because many Symbiodiniaceae are currently difficult to culture (*Krueger & Gates, 2012*). Furthermore, many strains cultured from host tissue do not represent the dominant Symbiodiniaceae in a host species (*Santos, Taylor & Coffroth, 2001*). We encourage efforts toward testing new media and bringing new species into culture (*Nitschke et al., 2020*), as well as documenting and sharing successful and failed attempts. "Culturability" itself may be a useful phenotype to track, as it may reflect the degree of host-specificity, and influence media or antibiotic choice (*Ishikura et al., 2004*; *Nitschke et al., 2020*; *Reimer et al., 2010*; *Yorifuji et al., 2021*). Motility, cell division rates (growth), bacterial communities (microbiomes) and viral consortia (viromes) are also informative characteristics that can vary within and among symbiont species (*Grupstra et al., 2022a*; *Lawson et al., 2018*; *Levin et al., 2017*; *Parkinson & Baums, 2014*; *Yamashita & Koike, 2016*). Constructing a global phenotypic database for cultures, much like the Coral Trait Database (*Madin et al., 2016*) is another priority for Symbiodiniaceae research, as is exploring the culturable fraction of coral-associated bacteria that may interact directly with Symbiodiniaceae and impact their performance (*Frommlet et al., 2015*; *Lawson et al., 2018*; *Matthews et al., 2020*; *Sweet et al., 2021*; *Li et al., 2022*).

Finally, it would be advantageous to identify and culture model Symbiodiniaceae lineages to test species boundaries. For example, measuring DNA sequence differences between sibling species separated by a geological barrier (*e.g.*, the Isthmus of Panama; *LaJeunesse et al., 2018*; *Pochon et al., 2006*) would provide molecular-divergence cutoffs that could then be applied to better resolve sympatric lineages. Additionally, cultures of closely related, putative sibling species could be used to explore cytological evidence for sexual recombination (*Figueroa, Howe-Kerr & Correa, 2021*), evaluate potential hybridization (*Brian, Davy & Wilkinson, 2019*), and characterize the role symbiotic interactions play in genome evolution (*González-Pech et al., 2019*).

# GUIDANCE FOR POPULATION-LEVEL ASSESSMENT OF SYMBIODINIACEAE

## How can we design population-level studies?

Studies evaluating the distribution of genetic variation within species, often across spatiotemporal gradients or among host taxa, seek to understand how populations are influenced by evolutionary processes such as gene flow, genetic drift, and selection (*Aichelman & Barshis, 2020*; *Davies et al., 2020*; *Forsman et al., 2020*; *Prada et al., 2014*; *Reich et al., 2021*; *Thornhill et al., 2017*; *Turnham et al., 2021*). Here, we define a population as a group of individuals belonging to the same species that live and interbreed with each other in a given space and time. The study of Symbiodiniaceae populations is fundamental to improving the resolution at which phenotypes of interest are differentiated. Thus, here we focus on allele-based identification and quantification of genetic variation.

Because a single host can contain a mixture of multiple species and/or genera, a first step in experimental design should include assessing sample sets for the presence of multiple distinct Symbiodiniaceae that may confound the interpretation of population-level genetic variation (see "Guidance for Community-Level Assessment of Symbiodiniaceae"). Such assessment can be done pre- and post-population-level analysis with established genetic markers (*e.g.*, ITS2, *cp23S*) and may be guided by published literature for some regions or host species. Pre-screening is especially advantageous where information on the community composition of Symbiodiniaceae is also sought and especially for hosts which tend to associate with multiple genera or species. Quantitative PCR (qPCR) is one potential technique to pre-screen Symbiodiniaceae samples for the presence of particular lineages (*Correa, McDonald & Baker, 2009*; *Mieog et al., 2007*; *Saad et al., 2020*). After pre-screening, population-level studies typically target genetic variation from the numerically dominant symbiont associating with a particular host or set of hosts (*Baums, Devlin-Durante & LaJeunesse, 2014*), while excluding any confounding genetic variation from additional species that may be present within host samples (*Baums et al., 2010*; *Thornhill et al., 2006*). Post-screening of samples is also possible using tests of assignment to genetic clusters (*Davies et al., 2020*) or identifying and excluding samples with outlier allelic profiles. Post-screening may be more time- and cost-effective as verification can be performed on a subset of the total sample set.

The ideal number of samples to collect and analyze will depend on the particular aim(s) of the study (*e.g.*, delineating populations *vs.* characterizing the degree of admixture among them), the scale of comparison (*e.g.*, reef, habitat, colony, intra-colony, *etc.*), and the markers being employed. However, studies leveraging more traditional markers, such as microsatellites, tend to benefit from robust sample sizes with minimum ranges of 20–30 individual hosts *per* level of interest (*e.g.*, habitat and location) (*Hale, Burg & Steeves, 2012*). Although this is a good target, studies limited by permit authorizations, budgets, and other constraints are still informative in some contexts.

## How can we best use microsatellite loci?

Microsatellite loci (or simple sequence repeats; SSRs) are segments of DNA where 1–6 base pairs are repeated in a tandem array; these loci are distributed abundantly across genomes of nearly all eukaryotic organisms (*Tautz, 1989*). Variations in the length of repeats are generated by polymerase slippage during DNA replication, resulting in homologous regions (*i.e.*, loci) of differing lengths (*i.e.*, alleles) among individuals. Microsatellites are generally thought to represent neutral loci with high mutation rates. Their single-locus, multiallelic, and codominant properties can yield valuable information regarding ploidy and reveal genetic structure among populations within and between species. Furthermore, microsatellite analyses are generally a PCR-based technique, making them cost-effective relative to other methods (*Sweet et al., 2012*). With the advent of high-throughput sequencing and transcriptomics, the generation of hundreds of potential microsatellite loci is now comparatively straightforward (*e.g.*, *Abdelkrim et al., 2009*). For species or lineages where numerous loci are available, costs and effort can remain low by multiplexing primer sets (*Davies et al., 2013*). Taken together, these features make microsatellites attractive for studying Symbiodiniaceae populations. These markers have been used to address questions related to overall diversity, population structure within and between reefs, gene flow, dispersal, and relatedness between symbionts (see Table 1 in *Thornhill et al., 2017*).

Once the target Symbiodiniaceae species or lineage has been identified within a dataset, these samples can be tested for variability using previously developed microsatellite loci *via* PCR amplification (Fig. 3). Primers for such loci have been developed for Symbiodiniaceae species across at least five genera: *Symbiodinium* (*Pinzón et al., 2011*), *Breviolum* (*Andras, Kirk & Drew Harvell, 2011*; *Grupstra et al., 2017*; *Pettay & LaJeunesse, 2007*; *Santos, Taylor & Coffroth, 2001*; *Santos, Gutierrez-Rodriguez & Coffroth, 2003*; *Wirshing, Feldheim & Baker, 2013*), *Cladocopium* (*Bay, Howells & van Oppen, 2009*; *Davies et al., 2020*; *Howells, van Oppen & Willis, 2009*; *Magalon et al., 2006*; *Wham & LaJeunesse, 2016*), *Durusdinium* (*Pettay & LaJeunesse, 2009*; *Wham, Pettay & LaJeunesse, 2011*), and *Philozoon* (*Molecular Ecology Resources Primer Development Consortium et al., 2010*). Importantly, these loci tend to have narrow phylogenetic ranges, with primers developed for a given species typically working only on other closely-related species within the same genus. Therefore, it is necessary to screen existing primers for utility with a given target species, to ensure that allelic variability among the chosen suite of microsatellite loci is sufficient, and to develop novel primer sets if existing primers fail or prove insufficiently specific. Ideally, new Symbiodiniaceae primers should be tested against monoclonal cultures of species within the same genus (positive controls) as well as against symbiont-free sperm or apo-symbiotic larvae (negative controls) to rule out off-target PCR amplification of host DNA. Although more loci will generally increase discriminatory power in population-level studies, as few as 2–3 loci have provided sufficient discriminatory power for some questions (*Santos, Gutierrez-Rodriguez & Coffroth, 2003*; *Thornhill et al., 2009*).

### Analyses of microsatellite data

Given that Symbiodiniaceae are haploid in their vegetative life stage (*Santos & Coffroth, 2003*), a single allele *per* microsatellite locus is expected when a host harbors a single clonal

**Sample Collection**

Considerations for Experimental Design:
- Scale of comparison
- Allelic diversity
- Environmental impacts of sampling
- Presence of cryptic host species

Standardize Sampling:
- Position within colony
- (Micro)habitat variation among colonies
- Environmental metadata (*e.g.*, depth)

**Marker Choice**

Pre-Screening:
- Test for the presence of multiple genera (*e.g.*, ITS2, qPCR, gel-based approaches) or use prior information on association specificity

Loci Selection and Verification:
- 6-12 loci, but as few as 2-3 may be sufficient
- More loci = higher discriminatory power
- Verify primer specificity for target species
- Develop and test new loci as required

**Analytical Pipeline**

I. Test for Multiple Species
- Identify strong assignments to different genetic clusters (*e.g.*, STRUCTURE assignment values >0.75)
- Split data by species for subsequent analyses

II. Multi-Locus Genotypes (MLGs)
- Create MLGs where single alleles per locus imply individual genotypes
- Test for genetic structure (*e.g.*, AMOVA, PCoA, STRUCTURE)

III. Presence/Absence of Alleles:
- Create binary allelic data where multiple alleles per locus prevent assignment of MLGs
- Explore patterns of genetic structure (*e.g.*, DAPC, NMDS, PCoA, STRUCTURE)
- Caveat: genetic diversity overestimated if origin is intra- rather than inter-genomic

**Figure 3 Recommendations for designing microsatellite-based Symbiodiniaceae population genetics experiments.** Sample collection, marker choice, and analytical pipeline should be considered from the outset.

strain of Symbiodiniaceae (represented by a single multi-locus genotype; MLG). When a single allele is recovered from nearly all loci, establishing MLGs is straightforward. However, recovery of multiple alleles at a given locus from a single sample is not uncommon (Fig. 1). Instances of multiple alleles *per* locus can be interpreted as detection of cells from multiple genetic strains (multiple MLGs) within host tissues (*Andras et al., 2009*; *Grupstra et al., 2017*; *Santos, Gutierrez-Rodriguez & Coffroth, 2003*; *Santos & Coffroth, 2003*; *Thornhill et al., 2017, 2009*). Examples of multiple MLGs tend to be more common within Indo-Pacific corals hosting *Cladocopium* species (*Bay, Howells & van Oppen, 2009*; *Davies et al., 2020*; *Wham, Carmichael & LaJeunesse, 2014*), whereas they are less common in Caribbean corals hosting *Cladocopium* and other genera (*Andras et al., 2009*; *Grupstra et al., 2017*; *Pettay et al., 2015*; *Santos, Gutierrez-Rodriguez & Coffroth, 2003*; *Santos & Coffroth, 2003*; *Thornhill et al., 2017, 2014, 2009*). Consistent patterns of multiple alleles for certain loci among a subset of monoclonal cultures has led to the proposal of whole or segmental genome duplication within certain Symbiodiniaceae. This scenario would make overestimation of symbiont genotype diversity within samples likely (*Wham, Carmichael & LaJeunesse, 2014*), and make the assignment of MLGs difficult, raising challenges for data analyses and interpretations.

Several approaches have been developed to accommodate instances of multiple MLGs within a sample (Fig. 3; *Andras, Kirk & Drew Harvell, 2011*; *Davies et al., 2020*; *Howells et al., 2013b*; *Kirk et al., 2009*; *Magalon et al., 2006*; *Wham & LaJeunesse, 2016*), including the exclusion of some samples and/or genotypes in certain cases. When multiple alleles for a given locus occur infrequently among samples, two data sets can be created: (1) a set where all microsatellite alleles within each sample are used and scored for presence or

absence (*i.e.*, binary) within each sample, and (2) a curtailed data set omitting samples with multiple alleles at one or more loci, allowing MLGs to be assigned. Notably, studies using this approach have come to similar conclusions across the two data sets (*e.g.*, *Andras, Kirk & Drew Harvell, 2011*; *Davies et al., 2020*; *Howells et al., 2013b*; *Kirk et al., 2009*; *Magalon et al., 2006*; *Wham & LaJeunesse, 2016*). In general, given that reported scales of genetic divergence are similar across studies using binary and MLG-based approaches, and excluding many samples can lead to underestimating genetic diversity (*Howells et al., 2016*), we suggest that the binary approach should be used when possible (*e.g.*, a high proportion of samples exhibit multiple alleles *per* locus).

### Caveats

While microsatellite analyses have proven informative and valuable in population genetic studies of Symbiodiniaceae, they present challenges in data acquisition and interpretation. For example, the long repetitive regions of microsatellites are often difficult to reliably amplify, making it arduous to verify repeat length *via* fragment analysis. Microsatellites can suffer from allele dropout, and low specificity of PCR primers, which can potentially lead to diversity underestimates within a sample. Microsatellites themselves are subject to more general criticisms including unclear mutation models and the potential for homoplasy (*Putman & Carbone, 2014*). Additionally, many analytical pipelines used to assess population genetic patterns make basic assumptions that Symbiodiniaceae do not follow (*e.g.*, that organisms are diploid and exhibit predominantly sexual reproduction). In light of this, researchers should be cautious about interpreting results from pipelines developed for organisms that exhibit more traditional population biology.

## What other markers can resolve Symbiodiniaceae populations?

The ITS2 region of rDNA is repeated in tandem arrays within all known Symbiodiniaceae genomes. For population-level assessments, this universality presents an advantage over microsatellites, but the multi-copy nature of this marker poses unique challenges. As long as appropriate analytical frameworks are applied (see "Guidance for Community-Level Assessment of Symbiodiniaceae"), ITS2 data can be used to resolve strains within species. Such assessments require consideration of similarities in the assemblages of ITS2 sequences and their relative abundances within each genome. For example, genetic structure among *Cladocopium thermophilum* strains in the Persian/Arabian Gulf has been characterized (*Hume et al., 2019*; *Smith, Ketchum & Burt, 2017*) and patterns of IGV obtained from amplicon sequencing data show fine-scale spatial structure among *C. thermophilum* populations separated by tens to hundreds of kilometers (*Howells et al., 2020*). However, recombination (*i.e.*, whether two populations are interbreeding) is often considered sufficient for operational recognition that those entities are members of the same species (*Andras et al., 2009*; *Grupstra et al., 2017*; *Santos, Gutierrez-Rodriguez & Coffroth, 2003*; *Santos & Coffroth, 2003*; *Thornhill et al., 2017*, *2009*). Therefore, it is difficult to determine whether ITS2-based genotypes correspond to distinct populations of the same species or different species. Other markers are also able to resolve at the population level, but their application to Symbiodiniaceae population biology is limited.
Examples include the chloroplast *psbA* minicircle noncoding region (*psbA^{ncr}*; *Moore et al., 2003*) and the chloroplast *23S* ribosomal region (*cp23S*; *Santos, Gutierrez-Rodriguez & Coffroth, 2003*).

## What are the next steps for understanding Symbiodiniaceae population biology?

Advancing our understanding of Symbiodiniaceae population biology will be greatly informed by leveraging samples that have single Symbiodiniaceae MLGs (*Prada et al., 2014*). For example, available monoclonal cultures of Symbiodiniaceae from several species could be used to develop and test new technologies and markers (including validation of copy number, see "Accounting for Copy Number Variation") and these technologies could then be extended to more complex associations *in hospite* (within a host organism). To overcome the challenges of widespread gene duplication in Symbiodiniaceae genomes (*González-Pech et al., 2021*; *Pochon et al., 2012*; *Prada et al., 2014*), efforts should be directed toward identifying new low copy markers (or preferably single copy markers). Discovery of single copy loci may be informed by screening for universal single copy markers collated in the Benchmarking Universal Single-Copy Orthologs (BUSCO) database (*Seppey, Manni & Zdobnov, 2019*; *Simão et al., 2015*), although many BUSCOs are undetected in Symbiodiniaceae genomes (*González-Pech et al., 2021*). Restriction-associated DNA sequencing may serve as a low-cost method for generating single copy markers for population-level assessments in Symbiodiniaceae (*Kitchen et al., 2020*; *Suyama & Matsuki, 2015*); however, these methods require further development.

Whole-genome sequencing (WGS) is also becoming more affordable, especially at low coverage (<5X), opening the possibility of evaluating genome-wide variation in Symbiodiniaceae (*González-Pech et al., 2021*; *Reich et al., 2021*), although Symbiodiniaceae genomes are large (>Gbp) and few chromosome-scale assemblies exist (*Marinov et al., 2021*; *Nand et al., 2021*). We suggest that WGS first be applied to isoclonal cultures, where possible, to ensure reads derive from one genetic entity (*Voolstra et al., 2021a*; *McKenna et al., 2021*). Subsequently, this approach can be applied to multispecies assemblages where different Symbiodiniaceae lineages within the same genus could be mapped to these reference genomes. These types of analyses would allow for simultaneous quantification of gene flow and divergence among Symbiodiniaceae populations of co-occurring species and improve estimates of effective population sizes and clonality within and among species, hosts, and reefs. Another major advantage of genome-wide data is the potential to evaluate adaptive (non-neutral) genetic variation and signatures of selection across the genome (*Ladner, Barshis & Palumbi, 2012*; *Liu et al., 2018*; *Voolstra et al., 2009*). For example, identifying associations between traditional markers, genomic regions, Symbiodiniaceae functional traits, and/or environmental variables–including those that are important for the survivorship of corals under warmer, more acidic, and more eutrophic oceans–remains a research priority (see "Beyond Genotype: Phenotyping Symbiodiniaceae" and "Integrating Multiomic Technologies to Study Symbiodiniaceae").

# GUIDANCE FOR COMMUNITY-LEVEL ASSESSMENT OF SYMBIODINIACEAE

## What is a Symbiodiniaceae community?

Generally defined, ecological communities are composed of more than one species that live together and interact. However, what is meant by terms such as "together" and "interact" can vary (*Konopka, 2009*), particularly when considering free-living *vs.* symbiotic Symbiodiniaceae. Typically one to two (but up to 10) Symbiodiniaceae cells reside in a coral gastrodermal cell (*Davy Simon, Allemand & Weis Virginia, 2012*; *Muscatine et al., 1998*), potentially restricting direct interactions between the endosymbiont cells within a coral host. Here, we use the term "local Symbiodiniaceae community" to refer to two or more Symbiodiniaceae species *within a single host*, whereas "macroscale Symbiodiniaceae community" (see "phenomenological community" in *Konopka (2009)*) describes the diversity of Symbiodiniaceae across some larger scale (*e.g.*, conspecific hosts or multiple host species). Environments that include multiple free-living Symbiodiniaceae species also constitute macroscale communities; *e.g.*, benthic sediments (*Nitschke, Davy & Ward, 2016*; *Quigley, Bay & Willis, 2017*; *Sweet, 2014*), the water column (*Fujise et al., 2021*; *Porto et al., 2008*), and macro-algal surfaces (*Fujise et al., 2021*; *Porto et al., 2008*).

Macroscale Symbiodiniaceae communities contain more species and encompass higher genetic diversity than local Symbiodiniaceae communities because symbiotic diversity accumulates with increased host colony and habitat sampling (*Swain et al., 2020*). Environmental samples include cells of symbiotic Symbiodiniaceae expelled from hosts as well as non-symbiotic, free-living species. In contrast, a given adult host typically harbors only one or two dominant Symbiodiniaceae species (*Goulet, 2006*), often from distinct genera, as well as other species at low relative abundances (*Hume et al., 2020*; *Silverstein, Correa & Baker, 2012*). *In hospite* Symbiodiniaceae communities can be transmitted vertically (promoting higher fidelity), reassembled horizontally (allowing for greater flexibility), or some combination of both (mixed-mode transmission) with each host generation (*Quigley, Willis & Bay, 2017*). The diversity of the free-living component of macroscale Symbiodiniaceae communities and the symbiotic component of local Symbiodiniaceae communities are each likely to be underestimated (*e.g.*, *Baker & Romanski, 2007*), but for different reasons. Free-living communities are relatively diffuse and are therefore more difficult to exhaustively sample. In contrast, local Symbiodiniaceae community assessments are prone to sampling bias (but see, *e.g.*, *Goulet & Coffroth, 2003*). Characterizations of local communities are often based on a single sample from a well-lit, "top" surface of a colony. Sampling across a host's surface has revealed heterogeneous distributions of dominant Symbiodiniaceae within colonies of Caribbean stony corals such as *Colpophyllia*, *Montastraea*, *Orbicella*, *Porites*, and *Siderastrea* (*e.g.*, *Correa et al., 2009*; *Kemp, Fitt & Schmidt, 2008*; *Rowan et al., 1997*; *Ulstrup & van Oppen, 2003*), as well as some Pacific stony corals (*e.g.*, *Fifer et al., 2022*; *Innis et al., 2018*; *Kemp, Fitt & Schmidt, 2008*; *Rowan et al., 1997*; *Ulstrup & van Oppen, 2003*) and zoantharians such as *Zoanthus* (*Fujiwara et al., 2021*) and *Palythoa* (*Wee, Kobayashi & Reimer, 2021*). Whether local Symbiodiniaceae communities exhibit structure over smaller spatial scales *in hospite* (*e.g.*,

oral *vs.* aboral host surfaces) is unknown, but could be resolved with single-cell techniques (see "Integrating Multiomic Technologies to Study Symbiodiniaceae").

## Why study Symbiodiniaceae community diversity?

Studying macroscale communities can provide insights into cnidarian-Symbiodiniaceae dynamics along environmental gradients (*Cunning et al., 2015*; *Rossbach et al., 2021*; *Silverstein et al., 2011*; *Terraneo et al., 2019*). Regional macroscale Symbiodiniaceae community structure (*i.e.*, beta diversity) may also reflect chronic disturbance from anthropogenic activity (*Claar et al., 2020a*) and help identify more resilient or resistant reefs (*Ziegler et al., 2015*). Additionally, macroscale communities in reef seawater, sediments, feces, and on macro-algal surfaces may be important sources of symbiotic Symbiodiniaceae that can be acquired horizontally by prospective hosts (*Adams, Cumbo & Takabayashi, 2009*; *Ali et al., 2019*; *Castro-Sanguino & Sánchez, 2012*; *Coffroth et al., 2006*; *Cumbo, Baird & van Oppen, 2013*; *Fujise et al., 2021*; *Granados-Cifuentes et al., 2015*; *Grupstra et al., 2022b, 2021*; *Nitschke, Davy & Ward, 2016*; *Porto et al., 2008*; *Quigley et al., 2018*; *Quigley, Bay & Willis, 2017*; *Sweet, 2014*; *Umeki et al., 2020*; *Venera-Ponton et al., 2010*). Symbiodiniaceae in a free-living mode may influence important processes, such as sexual reproduction, hybridization, and gene flow within Symbiodiniaceae (*Figueroa, Howe-Kerr & Correa, 2021*).

Positive and negative species interactions can occur within local Symbiodiniaceae communities resulting in resource and niche partitioning (*Davy Simon, Allemand & Weis Virginia, 2012*; *Howe-Kerr et al., 2020*; *Matthews et al., 2020*). Quantifying these interactions may help disentangle the factors and processes governing Symbiodiniaceae community assembly in early host life history stages (*McIlroy et al., 2019*; *Quigley, Willis & Bay, 2016*), as well as successional dynamics (or stability) in adult hosts. Studying local Symbiodiniaceae communities can also identify conditions that trigger symbiotic breakdown (*i.e.*, dysbiosis). Dysbiosis has frequently been documented in the bacterial communities of stressed hosts (*e.g.*, *Zaneveld, McMinds & Vega Thurber, 2017*; *Ziegler et al., 2017*; *Boilard et al., 2020*), and may also be evident in local Symbiodiniaceae communities. Generally speaking, dysbiosis can manifest itself in the host as: (1) an increase in symbiont richness (invasion or proliferation of low abundance symbionts), (2) a decrease in symbiont richness (loss of symbionts), or (3) more complex changes in community structure or beta diversity (*Egan & Gardiner, 2016*). For example, *Symbiodinium necroappetens* (*LaJeunesse, Lee & Gil-Agudelo, 2015*; *Stat, Morris & Gates, 2008*) and some symbionts in the genera *Durusdinium* (*Bay et al., 2016*; *Manzello et al., 2018*), *Breviolum* (*LaJeunesse et al., 2010b*), and *Cladocopium* (*Wee, Kobayashi & Reimer, 2021*) can opportunistically increase or decrease their abundance in bleached or stressed hosts. Stony coral juveniles in the field (*Quigley, Willis & Bay, 2016*) and adults in tank-based experiments (*Howe-Kerr et al., 2020*) have exhibited decreased survival in conjunction with more diverse local Symbiodiniaceae communities. Additional experiments to assess how frequently different types of dysbiosis occur in local Symbiodiniaceae communities are needed, including in non-scleractinian hosts, some of which can harbor up to 60 symbionts *per* host cell (*Fitt, 2000*). Testing the extent to which

different types of dysbiosis are associated with specific cnidarian hosts, as well as specific environmental contexts, should also be prioritized.

Current challenges in understanding local Symbiodiniaceae community diversity and dynamics include: (1) determining actual and relative abundances of Symbiodiniaceae species given IGV and copy number issues (see "Accounting for Copy Number Variation"); and (2) understanding the roles (if any) that low abundance Symbiodiniaceae play in holobiont survival and fitness (see *Arif et al., 2014*; *Bay et al., 2016*; *Lee et al., 2016*). This knowledge is key to connecting Symbiodiniaceae genotypes to phenotypes (see "Beyond Genotype: Phenotyping Symbiodiniaceae"). Low abundance Symbiodiniaceae may serve as a reservoir of *in hospite* algal genotypes that may increase to dominance (at least ephemerally) during or following a change in environmental conditions (*Bay et al., 2016*; *Berkelmans & Van Oppen, 2006*; *Boulotte et al., 2016*; *Buddemeier & Fautin, 1993*; *Claar et al., 2020a*; *Jones et al., 2008*; *Lewis, Neely & Rodriguez-Lanetty, 2019*; *Thornhill et al., 2006*; *Ziegler et al., 2018*). The mechanisms controlling this turnover *in hospite* remain poorly understood, but involve host rewards and sanctions (*Kiers et al., 2011*, *2003*) and competitive interactions among symbionts (*Palmer, Stanton & Young, 2003*). Competition among Symbiodiniaceae affects the initial uptake of symbionts in early coral ontogeny (*McIlroy et al., 2019*) and influences longer-term persistence in experimentally-generated symbioses (*Gabay et al., 2019*), but the relative importance of competition in shaping *in hospite* communities once they are established remains poorly understood. Beyond their potential to shift *in hospite* following bleaching events (*Jones et al., 2008*; *Thornhill et al., 2006*), low abundance Symbiodiniaceae could also contribute to emergent holobiont properties (*Howe-Kerr et al., 2020*; *Ziegler et al., 2018*). Quantification of holobiont traits with and without the addition of low abundance homologous Symbiodiniaceae (*i.e.*, lineages that typically enter into a symbiotic relationship with a given host taxon) from a range of inoculation sources constitutes a critical next step to understanding the functional role these symbionts play in the host.

## How can we optimize the study of Symbiodiniaceae community diversity?

Improving our understanding of the processes shaping Symbiodiniaceae communities is critical to predicting their distributions and potentially mitigating coral reef decline driven by global change. The methods below constitute suggested approaches for analyzing the diversity of macroscale and local Symbiodiniaceae communities. In some circumstances, identifying numerically dominant Symbiodiniaceae lineages (as opposed to the total diversity of a Symbiodiniaceae community) may be sufficient for the question at hand because hosts are generally selective in the symbionts they harbor, and some are highly specific to particular symbiont lineages (*e.g.*, *Hume et al., 2020*; *Thornhill et al., 2014*). Whether quantifying numerically dominant lineages or total Symbiodiniaceae community, the selection of molecular marker(s) and the approach(es) to data generation and analysis have implications for the interpretation of diversity. Molecular markers available for assessing Symbiodiniaceae community diversity are multicopy, and thus, present the challenge of distinguishing intragenomic from intergenomic variation. Inclusion of

symbiont taxa above or below the species level in the calculation of alpha and beta community diversity is problematic as these metrics are designed for species-level input. Including anything but species-level data in the calculation of these metrics can obscure patterns and lead to under- or over-estimation of diversity.

### Markers that behave as if single copy

The Symbiodiniaceae *SSU* (*i.e.*, *Murugesan et al., 2022*) and *LSU* rDNA markers as well as the *cob* mitochondrial marker are multicopy but are considered to behave like single copy loci because the vast majority of copies present are a single sequence. The few intragenomic sequence differences that do occur tend to be relatively straightforward to resolve in the context of identifying the dominant Symbiodiniaceae lineage within each genus. Many analysis algorithms produce amplicon sequence variants (ASVs), which are statistically inferred based on sequence variation within and among samples; the degree to which ASVs represent distinct genotypes may vary by marker and Symbiodiniaceae genus.
For example, *LSU* consistently resolves species within *Symbiodinium* (*Lee et al., 2015*), but not for all of *Breviolum* (Table 1; *Parkinson, Coffroth & LaJeunesse, 2015*). Thus, it is important to keep in mind that when assessing total community diversity (across multiple Symbiodiniaceae genera) with *LSU*, the number of species within certain genera may be under-represented. Despite this, markers that behave as if single copy are arguably the best option currently available for assessing total community diversity in Symbiodiniaceae as they avoid the many complications associated with interpreting variation from multi-copy markers (*LaJeunesse et al. 2022*).

### Multicopy markers

Among the commonly used markers, the hypervariable chloroplast *psbA* non-coding region ($psbA^{ncr}$) can resolve below the species level in Symbiodiniaceae (*LaJeunesse, Lee & Gil-Agudelo, 2015*; *LaJeunesse & Thornhill, 2011*; *Lewis, Chan & LaJeunesse, 2019*; *Turnham et al., 2021*; *Wham, Ning & LaJeunesse, 2017*), while the ITS2 region can resolve at, below, or above the species level depending on the lineage. Higher resolution comes at a considerable cost in terms of complexity of analyses.

$psbA^{ncr}$: The $psbA^{ncr}$ region can assess relatedness only among closely related Symbiodiniaceae lineages within the genus (*LaJeunesse & Thornhill, 2011*; *Thornhill et al., 2014*). It is helpful to have *a priori* knowledge of the genera being amplified when using this marker (see "When should Researchers use Multiple Symbiodiniaceae Genetic Markers for Community-level Analyses?") as available primers have known biases for specific genera. For example, the Symbiodiniaceae $psbA^{ncr}$ primers 7.4-Forw and 7.8-Rev (*Moore et al., 2003*) preferentially amplify *Cladocopium* in samples of mixed communities, whereas the more recent psbAFor_1 and psbARev_1 do not (*LaJeunesse & Thornhill, 2011*). Although $psbA^{ncr}$ is multi-copy and can exhibit IGV in some species, drawing inferences from these sequence datasets is still relatively straightforward because large genetic distances exist even between sequences from closely related species (*LaJeunesse et al., 2021*), similar to markers that do not present IGV. However, because the $psbA^{ncr}$ region cannot be amplified across Symbiodiniaceae using a single set of primers, this marker is suboptimal for some

**PeerJ** _________________________________________________

types of community-level analyses, such as assessing total community diversity or beta diversity metrics. Nevertheless, it would be appropriate to pair $psbA^{ncr}$ with other markers; *i.e.*, to resolve additional diversity within established ITS2 lineages (*Noda et al., 2017*; *Reimer et al., 2017*); and also to use this marker to verify ITS2 sequence variants generated *via* amplicon sequencing (*Hume et al., 2019*; *Smith et al., 2020*).

*ITS2:* The ITS2 region of Symbiodiniaceae rDNA resolves many species and some subspecies (*Hume et al., 2019*). ITS2 has a broader application for defining lineages because one set of primers amplifies all known Symbiodiniaceae ITS2 sequences (note, however, that sequence variants only align well within-genus). These two favorable characteristics, in concert with its history of use within the field, make ITS2 a popular choice among researchers, even in situations when greater resolution might be achieved with alternative marker(s). Intragenomic sequence diversity is relatively high within Symbiodiniaceae ITS2 (*Arif et al., 2014*; *Gong, Zhang & Li, 2018*; *LaJeunesse et al., 2022*) and along with copy number, varies considerably across genera (*Saad et al., 2020*) and likely species (though no data are currently available at this resolution). This IGV severely restricts the inferences that can be made regarding the relative abundance of community members in cases of multiple Symbiodiniaceae lineages *per* host (see "Accounting for Copy Number Variation"). The central issue in using ITS2 to characterize symbiont diversity *in hospite* is differentiating intragenomic sequence variants (those that reflect differences within one genetic entity) from intergenomic sequence variants (those that reflect differences between two or more genetic entities). This is of particular importance because, unlike with $psbA^{ncr}$, Symbiodiniaceae ITS2 intragenomic distances can be larger than intergenomic distances. Practically, it can be challenging to determine if sequence variation comes from one species or multiple species. Varied awareness and treatment of this issue among Symbiodiniaceae researchers has generated significant debate, which has often played out in peer review, rather than being articulated, addressed, and resolved as a research community (see "Ensuring an Inclusive Symbiodiniaceae Research Community").

One technique to differentiate between intra- and inter-genomic sequence variants involves analyzing co-occurrence patterns. Sets of different sequences that co-occur across multiple biological replicates are more likely to be from the same genotype than to derive from multiple co-occurring lineages, with each lineage contributing a subset of the sequences. This is particularly true in cases where the relative abundances of each of the sequences of the set are similar across biological replicates. There are gel-based (Denaturing Gradient Gel Electrophoresis, DGGE; *LaJeunesse, 2001*) and high-throughput sequencing methods that require downstream bioinformatic analysis (*e.g.*, *Frøslev et al., 2017*; *Green et al., 2014*; *Hume et al., 2019*) to detect these co-occurring sequences in both dominant and low abundance taxa. Gel-based and *in silico* approaches each have their advantages and disadvantages, which have been discussed elsewhere (*Saad et al., 2020*). Because these techniques rely on identifying banding profiles that correspond to references (gel-based) or other biological replicates (gel- and bioinformatic-based), their power to resolve diversity generally increases with access to references or further biological replicates. For this purpose, reference sets of DGGE profiles as published in the literature (*e.g.*, *LaJeunesse et al., 2010a*; *LaJeunesse & Thornhill, 2011*; *Silverstein et al., 2011*), or

online reference databases of *in silico* profiles (*e.g.*, at symportal.org) are available to researchers. However, strong inferences can still often be made from relatively small datasets for Symbiodiniaceae taxa that are sampled multiple times in the dataset. Both techniques rely on the same biological assumption: that coral hosts commonly associate with one numerically dominant Symbiodiniaceae taxon *per* genus. In cases where this assumption does not hold–when congeneric Symbiodiniaceae co-occur in multiple biological replicates–diversity may be underestimated with multiple taxa being considered one. Identifying intergenomic and intragenomic variation is necessary for making conclusions about diversity when using multi-copy markers like ITS2. Differentiating between this variation can be challenging, particularly when dealing with less common genotypes, smaller numbers of biological replicates, lower sequencing depths, and complex communities; in these situations, sequencing of the samples in question with an additional marker may be necessary. Critically, such an additional marker must be able to resolve between the putative taxa. For example, if attempting to ascertain whether 'two closely related *Cladocopium* taxa (*e.g.*, within the C3-radiation) are present in a sample, $psbA^{ncr}$ would be more appropriate than $cp23S$ as the former is highly likely to resolve between such taxa (*Thornhill et al., 2014*), whereas the latter may or may not (*Pochon et al., 2019*).

### Assessing total Symbiodiniaceae diversity

When characterizing both dominant and low abundance Symbiodiniaceae *in hospite*, three general considerations need to be made. First, Symbiodiniaceae communities can exhibit spatial structure within an individual host (*e.g.*, *Correa et al., 2009*; *Fifer et al., 2022*; *Kemp, Fitt & Schmidt, 2008*; *Rowan et al., 1997*). Second, assessment of total Symbiodiniaceae diversity is recommended with high-throughput sequencing or qPCR (genera/species present must be known *a priori* and primers specific to these must be available or designed) as these approaches provide the resolution to detect both dominant and low abundance Symbiodiniaceae. Gel- or Sanger sequencing-based methods can provide qualitative information on diversity, but lack the resolution to detect Symbiodiniaceae present at very low abundances (*i.e.*, <2–11% for restriction fragment length polymorphism-based (RFLP-based) methods, (*Correa, 2009*); <5–30% for denaturing gradient gel electrophoresis-based (DGGE-based) methods, (*LaJeunesse, Loh & Trench, 2009*; *Lien et al., 2007*; *Loram et al., 2007*)). Third, all caveats for specific markers from above still apply (*e.g.*, only diversity that can be resolved can be detected, and PCR biases may occur). Markers that behave as if single copy (*e.g.*, SSU, LSU, cob) are putatively well suited to characterizing total Symbiodiniaceae diversity due to their taxonomic breadth; analyses of total diversity using these markers will often be more straightforward than with $psbA^{ncr}$ (or ITS2). Despite this, $psbA^{ncr}$ is also a reasonable choice when investigating total Symbiodiniaceae diversity due to its apparent low(er) copy number and intragenomic richness, as long as the community diversity in question does not exceed the taxonomic range of this marker. In these limited circumstances, $psbA^{ncr}$ may resolve lineages well because genetic distances among taxa are relatively high with this marker.

When assessing the total diversity of macroscale Symbiodiniaceae communities, it is important to consider how molecular techniques and approaches apply to 'free-living

**Box 2 Characterizing free-living Symbiodiniaceae community diversity.**

Markers that behave as if they are single copy, as well as multicopy markers, may be applied to answering questions related to free-living Symbiodiniaceae communities. Although commonly used, primer sets for ITS2 are problematic because they result in non-target amplification of other species (*e.g.*, fungi, other dinoflagellates) present in the reef environments (*Sweet, 2014*; *Hume et al., 2018*; *Nitschke et al., 2020*). Despite this drawback of ITS2, other markers pose greater challenges to assessing free-living Symbiodiniaceae community diversity. Specifically, the *cp23S* marker frequently amplifies non-Symbiodiniaceae plastid-containing taxa when used in free-living systems (Nitschke unpublished data). Additionally, *cp23S*'s relatively coarse taxonomic resolution in some lineages (*e.g.*, *Breviolum*; *Parkinson, Coffroth & LaJeunesse, 2015*) may not be suited to some research questions. In contrast, *psbA^{ncr}* operates on a narrow taxonomic breadth (see "Multicopy Markers"). Thus, although there are no issues with non-target amplifications by *psbA^{ncr}*, multiple primer pairs would be required to amplify across all species of Symbiodiniaceae likely to be of interest; some of these primer pairs have yet to be developed.

ITS2 has its own challenges for assessing free-living Symbiodiniaceae communities because the process of looking for sets of sequences that co-occur among samples as a proxy for collapsing intragenomic variants (*e.g.*, *Hume et al., 2019*) is not a valid approach for free-living Symbiodiniaceae. This is because in the free-living environment, multiple Symbiodiniaceae species per genus are likely to be present in a single sample. A number of strategies exist to alleviate this problem. First, free-living Symbiodiniaceae communities, while interesting for their novel diversity, are likely to be studied alongside symbiotic Symbiodiniaceae on the same reef, allowing for recognition of symbionts in the water column that are likely derived from host expulsion. For example, *Fujise et al. (2021)* studied coral symbionts from the C15 and C3 radiations of *Cladocopium* and generated ITS2 defining intragenomic variant (DIV) profiles, or informative assemblies of within-sample intragenomic sequences (see *Hume et al., 2019* for details). These sets of ITS2 DIV sequences were then searched for in water, macroalgae, and sediment samples from the same reef. Complete sets of sequences from the C15 and C3 profiles were successfully retrieved from water and macroalgae, however in sediments only partial DIV profiles were retrieved alongside a greater representation of sequences from additional genera (*e.g.*, *Symbiodinium*, *Freudenthalidium*, *Gerakladium*, and *Halluxium*). It is not possible to differentiate whether these partial profiles in sediments represent novel *Cladocopium* diversity not present in corals or other hosts, or if sequencing depth was exhausted due to the greater representation of diversity across the family. A second approach, analogous to the first, leverages the high culturability of Symbiodiniaceae from free-living environments (*Hirose et al., 2008*; *Nitschke et al., 2020*; *Yamashita & Koike, 2013*). Of 263 Symbiodiniaceae-like single cells isolated from sands of the same reef examined by *Fujise et al. (2021)*, 114 successfully established as novel cultures belonging to the family Symbiodiniaceae (*Nitschke et al., 2020*). ITS2 sequences of these isoclonal cultures were later used by *Fujise et al. (2021)* as reference sequences and exact matches were found within the free-living communities. Again, both of these strategies rely upon building definitive sets of ITS2 sequences from Symbiodiniaceae cells of (ideally) a clonal population of a single strain within a single species, and then querying for these ITS2 sequence sets within communities of greater complexity.

Prior to the advent of high-throughput sequencing techniques, multiple markers were PCR amplified, cloned, and Sanger sequenced when examining free-living Symbiodiniaceae communities (a method which leads to issues of interpreting inter- *vs.* intra-genomic variation). For example, ITS2 and the short hypervariable region of *cp23S* (*cp23S-HVR*) have been used to study Symbiodiniaceae communities in the water column, sediments, and in stony corals in Hawaii and the Caribbean (*Manning & Gates, 2008*; *Pochon et al., 2010*). The *cp23S-HVR* primers were selected for their high specificity for Symbiodiniaceae; although the amplicons produced by these primers are of a size amenable to high-throughput sequencing workflows (~140 bp), this sequencing approach is not cost effective for these primers because the gene region appears to have less resolving power than ITS2 (*Pochon et al., 2010*; *Santos, Gutierrez-Rodriguez & Coffroth, 2003*). New, low copy number markers that resolve diversity at or below the level of ITS2 are needed to study the diversity of free-living Symbiodiniaceae communities. *psbA^{ncr}* has yet to be applied to free-living communities in a high-throughput approach, but this gene region is an obvious candidate.

Symbiodiniaceae'. In the broadest sense, this term refers to all cells external to metazoan (*e.g.*, coral, mollusc) or protistan (*e.g.*, ciliate, foraminifera) hosts. These cells may be found in the water column or associated with benthic substrates. 'Transiently free-living' refers to Symbiodiniaceae cells that are recently released from nearby hosts but that are not adapted to proliferate outside of hosts (*Yamashita & Koike, 2013*). In contrast, 'exclusively free-living' refers to Symbiodiniaceae species with lifestyles entirely external to hosts (*Jeong et al., 2014*). Although exclusively free-living Symbiodiniaceae may be detected occasionally "within" host samples, such detections can be interpreted as contamination resulting from host ingestion (rather than symbiosis establishment) or adherence to mucus (*Baker & Romanski, 2007*; *Lee et al., 2016*; *Silverstein, Correa & Baker, 2012*). Because 'free-living' (*sensu lato*) Symbiodiniaceae communities on reefs are complex mixtures of these two categories, resolving this diversity presents specific challenges (Box 2).
### Assessing beta diversity

Beta diversity can be useful for measuring changes to Symbiodiniaceae community structure over space and time (*Eckert et al., 2020*; *Epstein, Torda & van Oppen, 2019*). Although beta diversity encompasses a range of metrics including dissimilarity, turnover, nestedness, and dispersion, it is dispersion that is most commonly used to assess Symbiodiniaceae communities (*Arif et al., 2014*; *Claar et al., 2020b*; *Cunning, Gates & Edmunds, 2017*; *Green et al., 2014*; *Howe-Kerr et al., 2020*; *Hume et al., 2019*; *Quigley et al., 2014*). An important consideration when analyzing Symbiodiniaceae beta diversity data is establishing whether the analysis focuses on *sequence* beta diversity (*e.g.*, amplicon sequence variant data, which typically encompass copy number and intragenomic variability below the species level), or whether the analysis focuses on *ecological* beta diversity (*e.g.*, species data). Either approach may be viable, but it is important to explicitly state which is being used, and to frame interpretations based on the potential pitfalls relevant to that approach.

### Accounting for copy number variation

Copy number variation (CNV) is any genetic trait involving the number of copies of a gene in the genome of an individual. Efforts to quantify the absolute and relative abundances of different species in a local Symbiodiniaceae community are complicated by the presence of high CNV across taxa for key markers such as ITS2 (*Correa, McDonald & Baker, 2009*; *Mieog et al., 2007*; *Saad et al., 2020*; *Stat, Carter & Hoegh-Guldberg, 2006*; *Thornhill, LaJeunesse & Santos, 2007*). Thus, relative rDNA-PCR amplicon abundance does not necessarily equate to actual abundance of Symbiodiniaceae cells in a sample (especially when inter-genus comparisons are being made; *Arif et al., 2014*; *Correa, McDonald & Baker, 2009*; *Quigley et al., 2014*; *LaJeunesse et al., 2022*). For instance, some symbiont taxa that have been reported to possess a considerably higher rDNA copy number than others (*e.g.*, *Cladocopium* spp.; *Saad et al., 2020*); these high copy number taxa can appear to be abundant in mixed communities even though they might represent a low fraction of cells *in hospite*, leading to inaccurate estimation of actual symbiont abundances (Fig. 4).

The incorrect classification of low abundance *vs.* dominant taxa can impact interpretations related to biogeography and ecology. Such errors could be avoided if a correction factor is applied (*e.g.*, dividing the abundance value by the number of copies present in the genome of the relevant species, Fig. 4; *Correa, McDonald & Baker, 2009*; *Mieog et al., 2007*; *Rubin et al., 2021*; *Saad et al., 2020*), but such corrections rely on accurate copy number reference values, which are not currently available for the majority of Symbiodiniaceae taxa. Most studies that have quantified CNV have been limited to comparisons between genera, and there is considerable variation in the values reported across studies (*e.g.*, *Gong & Marchetti, 2019*; *Loram et al., 2007*; *Mieog et al., 2007*; *Quigley et al., 2014*; *Saad et al., 2020*; *Thornhill, LaJeunesse & Santos, 2007*). Inconsistencies in reported CNV values may be attributed to variation among strains or species (within-genus differences can be as large as

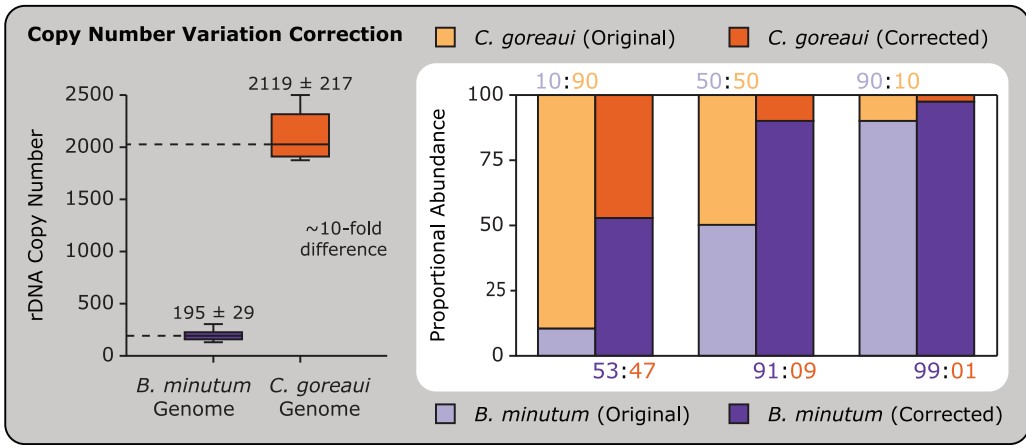

**Figure 4 An example of ITS2 rDNA copy number variation (CNV) between the genomes of two Symbiodiniaceae species from different genera (*Breviolum minutum* and *Cladocopium goreaui*).** Bar graphs demonstrate how original, uncorrected values (lighter bars) can lead to inaccurate perceptions regarding the proportional representation and numerical dominance of a species. In this case, raw *C. goreaui* ITS2 counts need to be divided by ~10 to correct for CNV (darker bars). Modified from *Saad et al. (2020).*

between-genus differences) or to methodological differences between studies. Therefore, the extent of CNV within Symbiodiniaceae genera or across populations largely remains to be established.

Lineage-specific qPCR assays have helped to quantitatively characterize mixed communities at the genus level (*Correa, McDonald & Baker, 2009*; *Cunning, Silverstein & Baker, 2018*; *Cunning & Baker, 2013*) and species level (*Fujiwara et al., 2021*). These targeted qPCR assays are more quantitative than sequencing approaches, but still require correction for CNV. When applied to systems with known symbiont diversity, qPCR can accurately and cost-effectively quantify local symbiont community structure and dynamics; however, these primer sets must be developed on a *per*-taxon basis. Another approach is the use of flow cytometry to quantify and/or physically separate cells of interest. While the natural variability in cell characteristics (*e.g.*, size, shape, fluorescence) cannot distinguish taxa (*Apprill, Bidigare & Gates, 2007*), the use of fluorescent probes to tag taxa of interest has successfully quantified the absolute and relative abundance of co-occurring taxa (*McIlroy, Wong & Baker, 2020*; *McIlroy, Smith & Geller, 2014*). Importantly, these methods are also conducive to subsequent genetic and physiological analyses of sorted cells. The development of further resources to account for CNV is an important priority within the field (see "Integrating Multiomic Technologies to Study Symbiodiniaceae").

## When should researchers use multiple Symbiodiniaceae genetic markers for community-level analyses?

As each marker has its own evolutionary history and methodological bias (*e.g.*, primer bias, CNV, *etc.*), congruence among multiple independent markers should enable more robust characterization of Symbiodiniaceae community ecology (*Fujiwara et al., 2021*; *Kavousi*

 *Smith, Ketchum & Burt, 2017*; *LaJeunesse et al., 2022*). Where possible, when multiple markers are used, the markers should complement each other's taxonomic scope and power to resolve. For example, it may be productive to pair the *cp23S* (larger taxonomic scope, lower resolving power) with the *psbA$^{ncr}$* (smaller taxonomic scope, higher resolving power). Instances where multiple markers produce conflicting results may help identify a Symbiodiniaceae lineage that cannot be accurately characterized with a single broad taxonomic marker. Additionally, combining multiple markers can provide greater resolution and improved interpretability compared to single-marker approaches. For example, the use of *psbA$^{ncr}$* can help overcome issues associated with interpretation of IGV and high copy number in ITS2 and provide support for ITS2-type profiles (*Smith et al., 2020*) by confirming which ITS2 IGVs are most likely part of a single lineage (*LaJeunesse & Thornhill, 2011*; *Smith, Ketchum & Burt, 2017*). Budget and logistics permitting, it is recommended to use multiple markers in some situations. For some ITS2 lineages (*e.g., Breviolum* B1 or *Cladocopium* C15; *Hoadley et al., 2021*; *Parkinson, Coffroth & LaJeunesse, 2015*), available markers that behave as if single copy do not distinguish ecologically relevant variants, but universal primers have yet to be designed for gene regions that capture this variation (*e.g., psbA$^{ncr}$*). Therefore, a marker that behaves as if single copy can first be applied to determine the dominant Symbiodiniaceae genera present to assist in the selection of the correct primer set for a higher resolution marker. We recognize that each additional marker can greatly increase high-throughput sequencing project costs, so such designs are only recommended when resources are available. Single marker studies can still provide great insight into symbiont community diversity as long as they are interpreted carefully.

## How can we interpret Symbiodiniaceae diversity while acknowledging the pitfalls of common markers?

Given the complexities associated with common methodological approaches and how they influence ecological interpretations of Symbiodiniaceae genetic information in community-level studies, it is critical to provide sufficient methodological details when reporting and interpreting results. We encourage the field to follow reproducible research standards, which include making analysis pipelines and raw data available after publication (see *Lowndes et al., 2017* for a comprehensive guide to open science tools). At minimum, commented code (including filtering thresholds, analysis decision points, and processing steps) should be deposited in each article's Supplemental Materials or in a publicly accessible repository (*e.g.,* GitHub) with a DOI (*e.g.,* procured through GitHub and Zenodo). Raw sequencing data must be deposited in a dedicated archive such as NCBI SRA for amplicon sequencing data, or NCBI Genbank for single sequence data. Alongside the code and sequences, additional metadata (*e.g.,* environmental and physiological parameters, as well as trackable information regarding the hosts' ID, if applicable; *Voolstra et al., 2021b*), should be deposited either with the publishing journal or with a data repository (*e.g.,* Dryad, Zenodo, the National Science Foundation's BCO-DMO). Finally, all of these deposition options can be integrated. For example, a Zenodo deposition can be

linked to a GitHub repository so that new code releases are automatically updated in the Zenodo repository. By making published data widely available, we can accelerate our understanding of cnidarian-dinoflagellate symbioses and their responses to a changing environment.

There continues to be dialogue amongst members in the Symbiodiniaceae field regarding the interpretation of gene amplicon data produced by metabarcoding (*e.g.*, Illumina MiSeq). Specifically, there is an ongoing debate about if and how to incorporate Symbiodiniaceae taxa present at low abundances, and how certain parameters are built into existing analytical pipelines. For example, SymPortal will not attempt to predict profiles for a Symbiodiniaceae genus in a given sample if there are less than 200 reads for that genus/sample combination (*Hume et al., 2019*); this can potentially contribute to systematic underestimation of total Symbiodiniaceae community diversity. As such, we encourage authors to consider carefully what their data can and cannot discern (*e.g.*, Table 1), report assumptions associated with their data interpretation, acknowledge that other interpretations exist, and discuss whether or not these other interpretations change the biological or ecological conclusions of their study. Diversity metrics merit careful attention and gene sequence diversity should not be conflated with species diversity. Authors (and reviewers and editors) can weigh what results, tables, or figures might be included in the Supplemental Material to acknowledge and address these additional interpretations in order to facilitate the inclusion of diverse perspectives (see "Ensuring an Inclusive Symbiodiniaceae Research Community").

## BEYOND GENOTYPE: PHENOTYPING SYMBIODINIACEAE

### Why do we need to characterize Symbiodiniaceae phenotypic diversity?

Not all genetically distinct Symbiodiniaceae taxa exhibit physiological differences (*e.g.*, through functional convergence; *Goyen et al., 2017*; *Suggett et al., 2015*), whereas unique isolates of the same taxon may be functionally divergent (*e.g.*, *Beltrán et al., 2021*; *Díaz-Almeyda et al., 2017*; *Hawkins, Hagemeyer & Warner, 2016*; *Howells et al., 2011*; *Mansour et al., 2018*; *Parkinson et al., 2016*; *Parkinson & Baums, 2014*; *Russnak, Rodriguez-Lanetty & Karsten, 2021*). This is not surprising given the effects of strong local selection within reef habitats (*Howells et al., 2011*; *Kriefall et al., 2022*; *Marhoefer et al., 2021*; *Suggett, Warner & Leggat, 2017*; *van Oppen et al., 2018*) and the role of acclimatization (*Torda et al., 2017*). Stringent functional interrogation is therefore critical to determining how healthy cnidarian-symbiont associations will survive the climate crisis. This goal rests on advancing physiological descriptions, increasing the number of cultured isolates from diverse hosts, and extending the methodological toolbox to characterize Symbiodiniaceae differences. Thus, a more comprehensive functional characterization can accompany taxonomic assignment, helping to build greater community consensus on methods and standards for describing phenotypes of interest (Fig. 5).

## Axenic Monoclonal Cultures (excluding bacteria)

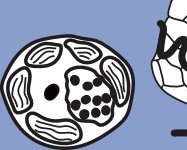

**Accessible Topics:**
- Prototrophy vs. auxotrophy (prototrophs are wild types that can grow on minimal growth media; auxotrophs are mutants that cannot grow on minimal growth media without supplements)

**Phenotype Influenced By:**
- Co-occuring microbes (*e.g.*, viruses)
- Potentially impacted by the absence of beneficial prokaryotes

**Benefits:**
- Control over co-ocurring prokaryotes and their bioactive products
- Host-symbiont recognition dynamics that are free from co-ocurring prokaryote interaction

**Caveats:**
- Effect of removing co-ocurring prokaryotes is unknown (antibiotics tend to reduce Symbiodiniaceae growth rates in culture)

**Need For Development:**
- Methodologies to produce and maintain axenic cultures
- Studies on mixotrophy and heterotrophy

## Xenic Monoclonal Cultures (including bacteria)

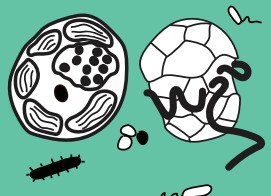

**Accessible Topics:**
- Zoospore morphology ~
- Formal taxonomy ~ (including morphological characteristics and genetics)
- Positive and negative taxis ~ (*e.g.*, chemotaxis, phototaxis)
- Circadian rhythms ~
- Heterotrophy ~

**Phenotype Influenced By:**
- Time in isolation ~
- Physical culture conditions ~ (*e.g.*, nutrients, light, trace metals)
- Co-occurring microbes (*e.g.*, prokaryotes, viruses)

**Benefits:**
- Easy and inexpensive to manipulate ~
- Removes any influence of holobiont environment ~
- Host-symbiont recognition dynamics

**Caveats:**
- Sampling bias for genera that are easy to culture ~
- Long-term mutations and adaptations influenced by culture history ~
- Artificial selection of co-occurring microbes

**Need For Development:**
- Methodologies to increase culture success
- Modified culture conditions that more closely reflect the endosymbiotic and free-living environments

## *In Hospite* (within host cells)

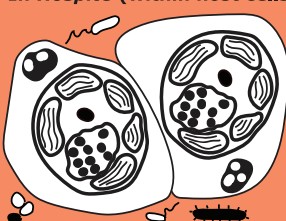

**Accessible Topics:**
- Photophysiology ^*
- Population growth ^*
- Cell cycle dynamics ^*
- Nutrient dynamics ^*
- Organelle-specific function ^*
- Population size *
- Multi-omics * (*e.g.*, transcriptomics, proteomics, lipidomics, *etc.*)

**Phenotype Influenced By:**
- Endosymbiotic environment
- Host genotype
- Host phenotype (*e.g.*, skeletal morphology, tissue biomass)
- Co-occurring microbes (*e.g.*, Symbiodiniaceae, prokaryotes, viruses)

**Benefits:**
- Provides the most accurate ecological and environmental context for the role of algal physiology and influence on (and reaction to) holobiont physiology

**Caveats:**
- Multiple partners in the holobiont (*e.g.*, possibly more than one alga, and a range of bacteria, viruses, archaea, fungi, *etc.*)

**Need For Development:**
- Establishing model systems that allow for:
  - Phenomic platforms *
  - High resolution functional imaging *
  - Control of host genetic background
  - Control of mixed colonization dynamics

~ Shared across axenic and xenic culture states
^ Complexity enhanced by the *in hospite* state compared to isolated culture
* Shared across all three states (axenic culture, xenic culture, and *in hospite*)

**Figure 5 Considerations for efforts to measure Symbiodiniaceae phenotypes across three states (axenic monoclonal culture, xenic monoclonal culture, and *in hospite*).**

## What do we need to consider when assessing Symbiodiniaceae phenotypes *in hospite*?

An overwhelming interest among Symbiodiniaceae researchers to date has been identifying thermal threshold phenotypes based on bio-optics (*e.g.*, *Goyen et al., 2017*; *Hennige et al., 2009*; *Voolstra et al., 2021d*), targeted biochemistry (*e.g.*, *Tchernov et al., 2004*) or "-omics" metrics (*e.g.*, *Olander et al., 2021*; *Roach et al., 2021*). These varied foci illustrate that phenotypes are operationally defined. Consequently, the detected functional diversity (the extent and range of phenotypes resolved) may appear different depending on the metrics used. For example, descriptions of phenotype diversity for heat stress sensitivity based on photobiological properties may not align with those based on metabolic indicators (*Goyen et al., 2017*) or light adaptation (*Suggett et al., 2015*, *2022*). Thus, reconciling genetic diversity with functional diversity must be carefully contextualized based on the measurement criteria and scientific questions at hand.

While it is valuable to confirm symbiont traits when in symbiosis, the presence of local Symbiodiniaceae communities and "secondary" symbionts in cnidarian holobionts complicate this effort. In local Symbiodiniaceae communities, it can be difficult to determine the relative abundance of each lineage present. Although accounting for copy number can help determine symbiont cell number and density in such cases (see "Accounting for Copy Number Variation"), other algal-centric physiological metrics, which reflect the combined average of all symbionts present within the host (*Cunning, Silverstein & Baker, 2018*), will be difficult to interpret. Single-cell sorting techniques may help assess unique phenotypic distinctions across different symbiont species from the same host (*Snyder et al., 2020*), but these techniques constitute additional effort and cost. Moreover, cnidarians host a variety of other microeukaryotic, prokaryotic, and viral symbionts (*Ainsworth, Fordyce & Camp, 2017*; *Hernandez-Agreda, Gates & Ainsworth, 2017*; *Thurber et al., 2017*), some of which are associated with colony health and resilience to environmental stress (*Bourne, Morrow & Webster, 2016*; *Voolstra et al., 2021c*; *Ziegler et al., 2017*). Some viruses even infect Symbiodiniaceae cells themselves (*Grupstra et al., 2022a*; *Levin et al., 2017*), with diverse potential impacts on Symbiodiniaceae phenotypes (*Correa et al., 2021*; *van Oppen, Leong & Gates, 2009*). The degree to which these "secondary" symbionts impact the observed phenotype of Symbiodiniaceae cells *in hospite* is an active area of research (*Maire et al., 2021*; *Matthews et al., 2020*). Finally, coral tissue thickness, pigmentation, skeletal reflectance, or other coral-associated microorganisms can affect irradiance levels reaching Symbiodiniaceae *in hospite* (*Dimond et al., 2013*; *Dimond, Holzman & Bingham, 2012*; *Enríquez, Méndez & Iglesias-Prieto, 2005*; *Marcelino et al., 2013*; *Smith et al., 2013*; *Titlyanov et al., 2009*; *Wangpraseurt et al., 2014*, *2012*). Variation in these physiological metrics can therefore affect symbiont phenotype, and lead to variable responses to climate stress (*Hoadley et al., 2019*).

Traits where variability across species exceeds that within populations are ideally suited for phenotypic analysis, but are presently unknown to the field or are challenging to measure in consistent and ecologically meaningful ways. Consequently, high-throughput approaches for assessing Symbiodiniaceae phenotypes need to consider tradeoffs that are

constrained by end goals. For example, recent high-throughput approaches for assessing thermal tolerance at the whole coral level—such as coral bleaching automated stress systems (CBASS; *Voolstra et al., 2020*)—and the single cell level (*Behrendt et al., 2020*) have incorporated short thermal challenges followed by stress characterization through the measurement of 1–2 physiological variables such as maximum PSII photochemical efficiency ($F_v/F_m$) and cell density. While single-phenotype assays can be informative within the context of ecosystem service values (*e.g.*, identifying thermally tolerant corals for nursery propagation; *Cunning et al., 2021*), identification of functionally distinct Symbiodiniaceae phenotypes will benefit from measuring a broader spectrum of physiological metrics (*Hoadley et al., 2021*). Phenotypic characterization using multiple photosynthetic metrics can provide some species-specific resolution (*Suggett et al., 2015*), and the non-invasive nature of chlorophyll *a* fluorometry lends itself to high-throughput approaches. However, poor contextualization of photosynthetic parameters with respect to cnidarian resilience currently limits the use of these techniques alone for large-scale phenomic studies, and may ultimately require integration of fitness metrics influenced by resource availability such as elemental composition *via* nutrient acquisition. While specific consensus on measurement protocols is beyond the scope of this perspective, taking a multidisciplinary approach and transparently documenting important methodological choices will help move the field forward.

## What do we need to consider when assessing Symbiodiniaceae phenotypes in culture?

Axenic or bacteria-depleted cultures are promising tools for connecting Symbiodiniaceae genotypes to phenotypes because their genetic identity is readily determined (see "Guidance for Species-Level Assessment of Symbiodiniaceae Diversity") and morphological, physiological, and behavioral diversity are readily discernible among such algal isolates (*Costa et al., 2019*; *Xiang, 2018*; *Xiang et al., 2013*). In terms of photo-physiology, fluorometry has become a convenient and accessible tool to gauge "culture health" (*Hennige et al., 2009*; *Robison & Warner, 2006*; *Suggett et al., 2009*). Fluorometry is also used in studies examining phenotypic variation focused on photosynthetic traits and how they are affected by resource availability (light, nutrients) and temperature (*Díaz-Almeyda et al., 2017*; *Suggett et al., 2015*), and has been inferred to reflect holobiont health (*Voolstra et al., 2020*). While photosynthetic traits are informative of cellular functioning, they are insufficient in isolation of other measurements to explain phenotypic variation in growth (*e.g.*, *Brading et al., 2011*; *Hennige et al., 2009*; *Suggett et al., 2015*). Recent data point to variable photo-physiological tolerance and thermal plasticity of genetically divergent Symbiodiniaceae grown in monoculture, which has contributed to a deeper understanding of the algal symbiont response to increasing sea surface temperatures (*Grégoire et al., 2017*; *Klueter et al., 2017*; *Russnak, Rodriguez-Lanetty & Karsten, 2021*; *Suggett et al., 2015*). However, a large number of genetically distinct algal symbionts identified *in hospite* have resisted sustained growth in culture (*e.g. Krueger & Gates, 2012*; *Santos, Taylor & Coffroth, 2001*). Furthermore, physiological and functional 'omics data indicate that when in culture or freshly isolated, Symbiodiniaceae exhibit
responses to thermal stress that differ from those of the same population *in hospite* (*Bellantuono et al., 2019*; *Gabay et al., 2019*; *Goulet, Cook & Goulet, 2005*). These data suggest that some physiological traits measured from culture-based studies may not be easily extrapolated to the symbiotic state. Such issues are particularly pronounced when measuring nutrient-associated phenotypes, as most culture media are nutrient-replete while Symbiodiniaceae *in hospite* appear to be nutrient-limited (*Maruyama & Weis, 2021*).

Emergent properties are novel characteristics that smaller units of organization gain when they become part of a larger complex system. Research focusing on core emergent properties expressed in culture, that can also be easily assessed in nature (*in hospite*), is logical given that phenotypes will consistently be the result of specific environmental conditions operating on the underlying molecular machinery. However, in decades of studies on Symbiodiniaceae cultures, the environmental conditions imposed have not consistently been reported at the time of, or prior to, sampling. Examples of such metadata include the growth phase (steady state *vs.* non-steady state; *Tivey, Parkinson & Weis, 2020*) or cell cycle phase (*Fujise et al., 2018*; *Tivey, Parkinson & Weis, 2020*), as well as the actual environments in the cultures (light quality/quantity, nutrients) as opposed to those measured in the incubators or assumed from the recipe of the medium used (*e.g.*, *Camp et al., 2020*; *Reich et al., 2020*), and the extent of bacterial loading. Consequently, developing guidelines for rigorous reporting of environmental (experimental) conditions when phenotypes are quantified is a key priority. Ensuring inter-comparability among studies in the future will similarly depend on operating under a more consistent set of measurement protocols for phenotypic traits.

## INTEGRATING MULTIOMIC TECHNOLOGIES TO STUDY SYMBIODINIACEAE

### How can genomics and high-throughput sequencing be leveraged?

With continued cost reductions and increases in computational power and accessibility, advanced "-omic" technologies, including genomics, transcriptomics, proteomics, and metabolomics, are rapidly enhancing our ability to understand biological mechanisms (*Krassowski et al., 2020*). Coupled with powerful, multivariate statistical techniques and machine learning, omics technologies have greatly refined our understanding of Symbiodiniaceae biology.

Whole-genome sequencing (WGS) remains the gold standard for capturing genetic diversity (*Aranda et al., 2016*; *González-Pech et al., 2021*; *Lin et al., 2015*; *Liu et al., 2018*; *Reich et al., 2021*; *Shoguchi et al., 2018, 2013*). While individual and concatenated genes can help resolve phylogenetic relationships and define taxonomic lineages (*LaJeunesse et al., 2018*; *Parkinson, Coffroth & LaJeunesse, 2015*), WGS data provide more comprehensive phylogenomic signals, which can be used to investigate divergent selection. For example, while a Symbiodiniaceae phylogeny reconstructed using *k*-mers (short, sub-sequences of defined length *k*) derived from whole-genome sequences is largely consistent with the phylogeny reconstructed with *LSU* rDNA data (*González-Pech et al., 2021*), different genomic regions exhibit distinct phylogenetic signals (*Lo et al., 2022*). Further comparison

of WGS data indicate that the similarity shared between different species within the genus *Symbiodinium* is comparable to that between different Symbiodiniaceae genera, revealing more extensive divergence than anticipated and suggesting a need for future revision (*Dougan et al., 2022*).

WGS efforts employ short- or long-read technologies, or a combination of both strategies. Short-read sequencing technologies (*e.g.*, Illumina) have typically offered a cost-effective approach for deep sequencing with low error rates (<1%) and have provided valuable insights into Symbiodiniaceae diversity, including gene family expansions across different Symbiodiniaceae lineages (*Aranda et al., 2016*; *Lin et al., 2015*; *Liu et al., 2018*). Though short-read sequencing can identify lineage-specific divergence, short reads are difficult to assemble, especially with highly repetitive genomic regions. Long-read sequencing technologies such as Pacific Biosciences (PacBio) or Oxford Nanopore Technologies offer a viable alternative to improve the contiguity of highly fragmented short-read genomes. However, long-read sequencing technologies are more expensive and error-prone relative to short-read platforms, though these technologies are rapidly advancing (*Karst et al., 2021*). Long-read data allow us to observe chromosome structure, a greater number of genomic elements, such as DNA transposons, long terminal repeats, or chromosomal enrichment for genes with similar biological functions (*González-Pech et al., 2021*; *Li et al., 2020*; *Nand et al., 2021*). Chromosome-level assemblies represent a major milestone in dinoflagellate genomics as they confirm that many genes are encoded in unidirectional clusters which correspond to large topological domains (*Marinov et al., 2021*; *Nand et al., 2021*). Not only does this discovery provide insights into the structure of genes in Symbiodiniaceae, but the structure can be correlated with the encoded genes and their expression patterns to observe their interactions, elucidating novel insights into the evolution of diverse Symbiodiniaceae lineages at the chromosome level (*Lin, Song & Morse, 2021*).

Efforts are underway to expand the number of high-quality short- and long-read assemblies for cnidarian-associated and free-living Symbiodiniaceae and incorporate these data into taxonomic descriptions (*Dougan et al., 2022*; *McKenna et al., 2021*; *Voolstra et al., 2021b*). Additionally, the two read types can be coupled (*e.g.*, Illumina with PacBio HiFi) to incorporate both the contiguous sequences (>20 kb) of long reads with the low error rate of short reads, allowing robust comparisons of sequence divergence within and across genomes (*Ebert et al., 2021*). Once more data become available, it may be feasible to incorporate whole-genome information into future taxonomic and systematic revisions to the family (*Dougan et al., 2022*). However, it will be crucial to achieve consensus on how to use these data to study Symbiodiniaceae diversity and taxonomy. A lack of consistency in methodology and quality standards persists, making cross-study analyses difficult (*e.g.*, *Chen et al., 2020*). First steps in such standardization have been taken (*Voolstra et al., 2021b*; *McKenna et al., 2021*), but will need to be expanded as more genome data become available and the community using these data grows. Additionally, the current costs of completely sequencing the genomes of hundreds of potential Symbiodiniaceae species remains prohibitive. For the near future, feasible alternatives for WGS include using reduced representation phylogenomic approaches (such as those targeting ultraconserved

elements; *Cowman et al., 2020*; *Quattrini et al., 2020*), full-length rDNA gene amplicons (*Tedersoo, Tooming-Klunderud & Anslan, 2018*), or entire organellar genome sequences (*Liu et al., 2020*). These alternatives may represent a compromise between WGS and phylogenetic marker studies by providing an intermediate amount of sequence information for taxonomic accuracy.

## How can transcriptomics, proteomics, and single-cell techniques advance our knowledge?

Researchers are keen to make functional inferences about Symbiodiniaceae, which requires focusing on coding regions. To do so requires sequencing the collection of RNAs within the cells (*i.e.*, transcriptomics). Transcriptome sequencing characterizes molecular phenotypes, such as transient responses to the environment, and can reveal differential gene expression among taxa that could reflect selective pressures driving Symbiodiniaceae diversification at the functional level (*Avila-Magaña et al., 2021*; *Bayer et al., 2012*; *Parkinson et al., 2016*). However, the extent of gene expression changes among Symbiodiniaceae is often surprisingly subtle (*e.g.*, *Barshis et al., 2014*; *Davies et al., 2018*; *Parkinson et al., 2016*), although this is a matter of current debate (*e.g.*, *Bellantuono et al., 2019*; *Voolstra et al., 2021d*). Furthermore, transcription can be influenced through alternatively spliced transcripts (*Lin, 2011*; *Méndez, Ahlenstiel & Kelleher, 2015*), RNA editing (*Liew et al., 2017*; *Mungpakdee et al., 2014*; *Shoguchi et al., 2020*), microRNA interactions (*Baumgarten et al., 2018*), and methylation of mRNAs (*de Mendoza et al., 2018*; *Lohuis & Miller, 1998*; *Yang, Li & Lin, 2020*). These post-transcriptional modifications create variation in the transcriptome, which can complicate transcriptomic interpretation, but tracking the conservation and divergence of these variations across the Symbiodiniaceae phylogeny may elucidate novel insights into the evolution of diverse lineages.

The primary concern with bulk transcriptomic analysis is that methods often pool transcriptomes from all Symbiodiniaceae cells within a host sample, so only "average" expression profiles can be generated (*Traylor-Knowles, 2021*). This approach may obscure nuances in the interactions between specific symbiont and host cells, especially for less abundant symbionts. Single-cell transcriptomics (*i.e.*, isolating individual cells and sequencing their transcriptomes) could solve this issue as gene expression could be explored within and among each symbiont cell *in hospite*. The generation of a cell atlas for the coral *Stylophora pistillata* has enabled the characterization of fine-scale metabolic interactions between symbionts and host gastrodermal cells (*Levy et al., 2021*). Single-cell sequencing can also enable high-resolution interrogations of how Symbiodiniaceae and host cells interact during symbiosis establishment, maintenance, and breakdown, particularly when Symbiodiniaceae cells can be isolated from different parts of the host coral that exhibit contrasting physiologies. By comparing expression from symbiont cells derived from different positions in the coral colony, the location and ecological role of Symbiodiniaceae can be characterized, which is a major priority for improving our understanding of symbiont communities within cnidarians.

Proteomic analyses provide an alternative mechanism to explore Symbiodiniaceae physiology and the functional impacts of different symbionts on cnidarian-algal associations, such as metabolic mismatches that occur when hosts associate with atypical (heterologous) symbionts (*Sproles et al., 2019*). Proteomic analyses use liquid chromatography-mass spectrometry to identify and quantify proteins, which are more directly linked to phenotype than transcript abundance (*Feder & Walser, 2005*). "Bottom-up" or "shotgun" approaches are commonly employed to identify and quantify as many proteins as possible in a sample in an untargeted manner, and with modern instrumentation, 3,000–4,000 proteins are commonly quantified in established model systems as well as Symbiodiniaceae (*Camp et al., 2022*; *Richards, Merrill & Coon, 2015*). Two features of proteomic analyses are particularly powerful. One is the characterization of post-translational modifications, such as protein phosphorylation, oxidation, acetylation, or ubiquitination (*Witze et al., 2007*). Post-translational modifications help regulate protein activity and are crucial in many biological processes, yet they cannot be detected by pre-translational analyses. The second is the ability to localize proteins to particular cellular compartments by either selectively enriching such compartments (*Tortorelli et al., 2022*) or fractionating the global cell content. Spatial proteomics aims to resolve the compositional architecture of cells by mapping the subcellular location of thousands of proteins simultaneously (*Lundberg & Borner, 2019*). Spatial resolution is achieved by separation and differential enrichment of cell content *via* density gradients or step-wise centrifugation and the subsequent quantitation of relative protein abundance across these fractions (*Dunkley et al., 2004*; *Geladaki et al., 2019*). Protein populations from differentially enriched organelles, membranes, and molecular complexes will have consistent but distinct distribution profiles and can thus be classified and mapped accordingly. Global proteome maps are powerful blueprints of the cell and can provide functional context for many uncharacterized proteins. This may be of particular use with Symbiodiniaceae, where sequence homology-based gene annotation is less effective due to their phylogenetic distance from well-studied organisms, and due to the highly derived genomes of dinoflagellates. However, these methods still require high-quality protein model search databases derived from genomic or transcriptomic sequences as they cannot identify proteins from complex samples *de novo*. Protein expression studies in combination with spatial proteomes will be powerful tools that can, for example, provide insights into the known architectural and physiological changes that accompany the symbiotic engagement of Symbiodiniaceae with their hosts. In the broader context, proteomics as a means to study the functional phenotype of Symbiodiniaceae under various conditions will likely overcome many of the current limitations of gene expression studies in dinoflagellates.

## What other omic technologies are promising?

Epigenomics, genome editing, metabolomics, and volatilomics are emerging areas within Symbiodiniaceae research. Epigenomic mechanisms, such as DNA methylation or chromatin modification, can modulate gene expression *via* gene suppression, gene enhancement, alternative mRNA splicing, or the regulation of spurious transcription

without requiring any changes to genomic sequences (*Bossdorf, Richards & Pigliucci, 2008*; *Feil & Fraga, 2012*; *Foret et al., 2012*). DNA methylation is one epigenetic modification that occurs when methyl groups are added to DNA nucleotides, altering how transcriptional proteins bind to promoter regions thereby altering gene expression (*Suzuki & Bird, 2008*). Symbiodiniaceae have unusually high levels of genome methylation (*Lohuis & Miller, 1998*). Originally, the high level of methylation raised uncertainty about whether methylation actually played a role in gene regulation, but methylation has been linked to differential gene expression with varying irradiance (*Yang, Li & Lin, 2020*). Thus far, epigenomic analyses have largely been focused on the host animal, and questions are often centered around how methylation contributes to environmental tolerance (*Dixon et al., 2018*; *Dixon, Bay & Matz, 2014*; *Dixon & Matz, 2021*; *Durante et al., 2019*; *Liew et al., 2018*; *Putnam, Davidson & Gates, 2016*; *Putnam & Gates, 2015*; *Rodriguez-Casariego et al., 2021*; *Rodríguez-Casariego et al., 2020*; *Rodriguez-Casariego et al., 2018*). Therefore, determining how methylation contributes to Symbiodiniaceae functional diversity requires further exploration.

Overall, Symbiodiniaceae genomes are very difficult to annotate. At present, dinoflagellate genome and transcriptome projects rarely manage to annotate >50% of putative coding sequences *via* homology searches against genes that have been functionally characterized in other organisms (*González-Pech et al., 2021*; *Stephens et al., 2018*). In the future, genome editing could be better developed to knock out Symbiodiniaceae genes with unknown functions, making it easier to determine their biological roles. UV mutagenesis is a classic method for introducing mutations; it was used recently to create photosynthesis mutants *via* screening of colored mutants (*Jinkerson et al., 2022*), but its random nature is less than ideal for reverse genetics. RNA silencing is a more targeted approach that could potentially be exploited in Symbiodiniaceae studies (*Zhang & Lin, 2019*), but the rapidly advancing CRISPR/Cas9 technology is most desirable for its ability to knock out specific genes. Although genome editing efforts for protists have made encouraging progress (*Faktorová et al., 2020*), success with Symbiodiniaceae remains elusive (*Chen et al., 2019*; but see *Gornick et al., 2022*).

Biochemical analyses of Symbiodiniaceae are also in the early stages. Characterization of metabolic products (metabolomics) and volatile organic compounds (volatilomics) can provide insights into molecular cross-talk between partners. Among Symbiodiniaceae, both metabolomic and volatilomic profiles are species-specific, but they also fluctuate with environmental conditions (*Klueter et al., 2015*; *Lawson et al., 2019*; *Roach et al., 2021*). Distinct biochemical profiles reflect the interactions and coadaptation (or lack thereof) between the host and symbiont (*Matthews et al., 2017*), so biochemical assays can lead to a greater understanding of the drivers of Symbiodiniaceae evolution. As with all the omics methods mentioned so far, if metabolomics and volatilomics are to be used to understand Symbiodiniaceae divergence, more data spanning the phylogeny will be required.

## How can we integrate omic technologies?
Integrative approaches that use more than one type of technology may be required to answer intricate research questions about Symbiodiniaceae biology. For example,

transcriptomics informs us of gene expression patterns, but cannot reveal protein end-products and how they are used for symbiosis, particularly because transcript abundance does not correlate well with protein levels (*Cziesielski et al., 2018*; *Liang et al., 2021*). However, when transcriptomics is integrated with proteomics or metabolomics, phenotypes can be directly observed. Then, tools that make use of multivariate statistics to combine these different types of biological data across studies, such as mixOmics (*Rohart et al., 2017*) or weighted gene coexpression network analysis (WGCNA; *Langfelder & Horvath, 2008*), can improve our ability to elucidate molecular mechanisms associated with phenotypes of interest. Thus, integration across several omic technologies ("multiomics") holds great promise for advancing our understanding of cnidarian-algal symbiosis (*e.g.*, *Camp et al., 2022*), yet the financial costs associated with using multiple technologies in parallel remains a limiting factor. Additionally, the expertise of one laboratory may be restricted to one major type of analysis or instrument. Therefore, collaborations among different research groups are essential (see "Ensuring an Inclusive Symbiodiniaceae Research Community"). Logistical hurdles to collaborations include how to house and share samples and data, along with the major financial burden. Integrative approaches will remain constrained until these issues are resolved or facilitated through funding agencies.

# ENSURING AN INCLUSIVE SYMBIODINIACEAE RESEARCH COMMUNITY

## How can we improve inclusivity in Symbiodiniaceae research?

Despite an increased recognition of the benefits of and need for more diverse representation in science, systemic biases continue to persist in science, limiting our creativity and innovation potential (*Ahmadia et al., 2021*). In coral reef science, where reefs are mostly found in non-industrial nations, capacity-building through "leveling the playing field" is required to facilitate a more inclusive research community and advance novel and important discoveries (*O'Brien, Bart & Garcia, 2020*). Marginalized groups within science have been and continue to be excluded from access to many opportunities, including funding, publishing, resources, collaborations, and networking. This exclusion is driven by limited resource availability and systemic racism, sexism, and ableism (*Davies et al., 2021*; *Dzirasa, 2020*; *Ginther et al., 2011*; *Hoppe et al., 2019*; *Taffe & Gilpin, 2021*; *Brown & Leigh, 2018*; *Yerbury & Yerbury, 2021*). While some progress has been made in the scientific community more broadly, there are still many deeply entrenched biases and critical gender, race, and ethnicity gaps that exist with respect to resource access; these need to be addressed by researchers, including those whose work focuses on Symbiodiniaceae, to ensure a more inclusive scientific community.

Research institutions, hiring committees, and organizers of panels, seminars, and conferences must actively work to change the demographics of scientists by increasing diversity at all career levels–from trainees to senior research scientists in positions of power. Gender, race, and ethnicity biases are rampant in the scientific-hiring process (*e.g.*, *Barber et al., 2020*; *Bennett et al., 2020*; *Huang et al., 2020*); for example, in the United

States these biases are particularly strong against Black and Latin scholars (*Eaton et al., 2020*) and in New Zealand biases are stronger against people of Māori and Pasifika descent (*Naepi et al., 2020*). In addition to recruitment difficulties, if scholars from these backgrounds are hired, they often face continued challenges that hinder their retention. Recognizing this, people in positions of power in the Symbiodiniaceae scientific community should (1) invest in retaining a diverse workforce by promoting the academic work of minority scientists; (2) provide spaces where researchers can safely report aggressions and other challenges (*Valenzuela-Toro & Viglino, 2021*); and (3) create programs that provide strong multidimensional mentorship, which serve to support and retain these scholars throughout each career stage (*Davies et al., 2021*; *Montgomery, 2017a*, *2017b*; *Montgomery, Dodson & Johnson, 2014*). Moving forward, Symbiodiniaceae researchers need to understand and implement the strategies and proposals that already exist and continue to be put forward regarding increasing recruitment and retention of historically marginalized scholars (*e.g.*, *Barber et al., 2020*; *Chaudhary & Berhe, 2020*; *Greider et al., 2019*). Increasing the diversity of perspectives at the decision-making table leads to more innovative discoveries (*Hofstra et al., 2020*; *Nielsen, Bloch & Schiebinger, 2018*), which are desperately needed to meet the formidable challenges of the coral reef crisis.

## How can we ensure an equitable publication process for everyone?

Equity and diversity issues exist in the scholarly publication process at multiple levels and across different areas of research. Men are first authors more often than women (*Casadevall et al., 2019*), notably even when both authors are identified as having contributed equally to the work (*Broderick & Casadevall, 2019*). Such systematic and implicit gender biases are also evident in the peer-review processes (*Calaza et al., 2021*). Manuscript authors, irrespective of gender, are also less likely to suggest women reviewers (*Fox et al., 2017*). Unprofessional reviews disproportionately impact members of underrepresented groups, who report greater self-doubt after receiving such reviews, ultimately reducing scientific productivity overall (*Silbiger & Stubler, 2019*). Beyond peer evaluation, more men serve in editorial roles than women (*Fox et al., 2019*; *Grinnell et al., 2020*; *Hafeez et al., 2019*; *Palser, Lazerwitz & Fotopoulou, 2022*; *Pinho-Gomes et al., 2021*), and editors tend to invite men more often than women to write invited reviews or perspectives. For example, the journal *Molecular Ecology*, which often publishes research from the Symbiodiniaceae community, found significant gender bias in authorship of invited 'perspective' articles, with women only authoring between 17.2–28.6% of these pieces (*Baucom, Geraldes & Rieseberg, 2019*).

Language biases are also pervasive. English is currently the default language of science, which disadvantages scientists who do not consider English as their primary language (*Gordin, 2015*). Non-native English speakers spend on average 97 more writing hours than native English speakers on preparation for each manuscript (*Ramírez-Castañeda, 2020*). In addition, ideas may be lost in translation or are often challenging to explain in a secondary language (*Flowerdew, 2001*). There are also costs associated with publishing in a non-native language: for example, paying for translation and editorial services.

Conversely, fluent English speakers publish more research articles at higher rates than non-English speakers (*Taubert et al., 2021*). To address English-centric journals, regional journals publish in their native languages (*Bordons, 2004*), but these publications are read by a smaller readership and are cited less (*Di Bitetti & Ferreras, 2017*), and thereby viewed as less impactful, and are less likely to be shared widely in the Symbiodiniaceae community.

General actions that can be taken within our community to ensure a more equitable publication process include: (1) increasing diversity on editorial boards, (2) increasing the diversity of invited reviewers, as well as the authors we review for; (3) promoting cost reduction strategies (as implemented in journals like *Frontiers, PeerJ*, and *PLoS One*) whereby publication fees are prorated by country or institution type, as well as other strategies that reduce editorial costs for non-native speakers (*Taubert et al., 2021*); (4) promoting double-blind review processes (*Budden et al., 2008*); (5) intentionally citing articles led by diverse colleagues (*e.g.*, from diverse gender identities and geographical areas), thereby increasing the diversity of perspectives in the field that contribute to discussion; and (6) gathering data about where and how these inequities exist and working together as a community to take actionable steps for equity in science.

### How can we avoid parachute science?

Parachute science, sometimes referred to as "helicopter" or "colonial" science, is a common practice whereby members of the scientific community from higher-income countries fail to involve local/indigenous/native people in an equitable fashion when performing research in lower-income countries (*Haelewaters, Hofmann & Romero-Olivares, 2021*; *Stefanoudis et al., 2021*). These practices tend to be more common in ecology and conservation research (*de Vos, 2020*), including coral reef studies. As a community, we need to understand the largely exploitative history of our discipline and avoid perpetuating it. We should stay informed regarding the history of the lands and peoples who live in the areas in which we conduct our studies. Our specific recommendations include: (1) developing laboratory manuals that include sections outlining values and best practices (including ethics approval and necessary permits); (2) adequately training students to conduct transparent research and develop equitable relationships with members of the host region; (3) supporting the establishment of long-term collaborations and exchange programs to involve local students in research; and (4) including the development of these relationships as important components of the tenure and promotion process in departments and institutions.

Importantly, researchers from institutions in high-income regions (*e.g.*, North America, Australia, Western Europe) should: (1) be sensitive to the many challenges their colleagues in lower-income regions experience, such as a lack of funding, infrastructure, and institutional support; (2) be respectful of these collaborators by treating them as peers and not as assistants, involving them in all steps of the science, and acknowledging their intellectual contributions during discussions, and; (3) be fair to these researchers and their

contributions through continued involvement in planning, manuscripts, projects, and grants generated through these collaborations. Together, these strategies can facilitate a more inclusive and collaborative Symbiodiniaceae community (*Armenteras, 2021*; *Belhabib, 2021*). Importantly, integration of members of the local community provides a long-term context to the collected scientific data, such as anecdotal observations (*e.g.*, episodes of bleaching) that may be critically informative to the research.

## How might we increase accessibility and collaboration?

There is an urgent need to increase the accessibility of Symbiodiniaceae science and foster collaboration, as innovation is necessary to address the coral reef crisis. Toward this goal, we have developed a living, global database of Symbiodiniaceae researchers and their key research expertise (http://symcollab.reefgenomics.org/; Supplemental Information). The database can be queried based on topic and methodology to aid scientists in diversifying their networks. Researchers joining the database are then also invited to participate in an open Slack channel ('Sym Slack', https://symslack.slack.com). These resources connect and promote diverse researchers and facilitate discussions of science from different perspectives. The COVID-19 pandemic also showcased the effectiveness of virtual conferences (case in point: this perspective is the product of a virtual workshop). Maintaining hybrid conferences with reduced costs for virtual attendance along with staggered schedules and access to presentation recordings to accommodate different time zones would ensure that the sharing of scientific information is more inclusive, alongside continued efforts to drop conference charges for lower-income countries (*e.g.*, the 15th International Coral Reef Symposium in 2022). Therefore, we encourage conference organizers to facilitate virtual attendance and funding sources for those who have difficulties traveling to foster equitable networking opportunities across people from diverse backgrounds and academic stages. Additionally, incentivizing consortia of Symbiodiniaceae researchers across diverse career stages and locations and explicitly engaging researchers from marginalized backgrounds would lead to stronger capacity building and greater transfer of knowledge. Lastly, we encourage sponsors to continue expanding their funding schemes to support international collaborations. Several examples of these efforts exist, including a new grant solicitation from the United States National Science Foundation, which calls for collaborations with Brazilian scientists through the São Paulo Research Foundation. Similar schemes (*e.g.*, Deutsche Forschungsgemeinschaft) also exist to foster collaboration between researchers in developing countries. The United States Fulbright Program and the European Union Marie Sklodowska-Curie Program are other examples that support collaborations across countries. The Japanese Society for the Promotion of Science provides funding for international exchange and research for graduate students, postdoctoral scholars, and early career scientists. These types of funding mechanisms are important because they promote wealth sharing across countries, encourage collaboration while thwarting parachute science, help with international challenges including research permitting, and ultimately lead to more open sharing of data and ideas within our Symbiodiniaceae research community and beyond.

## CONCLUSIONS

Addressing (and ultimately solving) the challenges associated with the coral reef crisis is increasingly urgent as climate change accelerates. Microalgae in the family Symbiodiniaceae play a critical role in determining coral bleaching outcomes. Advancing our knowledge of the genetic diversity of these organisms, how their diversity functionally impacts coral bleaching, and how we can apply such knowledge to mitigate climate change consequences is vital. We have identified consensus approaches for studies of Symbiodiniaceae genetic diversity at the species and population levels, while recognizing several outstanding issues regarding the characterization of community diversity. We highlight key paths forward for research including exploration of the phenotypic landscape and leveraging new technologies that are broadly applied in model systems. We also emphasize the need for increased collaboration and inclusivity among Symbiodiniaceae researchers. Overall, we acknowledge the dire need for advancing our understanding of Symbiodiniaceae ecology, physiology, and evolution, which will have the potential to expedite restoration practices and facilitate management decisions as we continue to push for political action on climate change.

## ACKNOWLEDGEMENTS

We would like to thank Dan Thornhill, Allison Lewis, and Joanna Shisler at NSF for attending and facilitating the workshop, and Andrea Grottoli, Jody Harwood, Krista Liguori, Amy Pruden, and Rebecca Vega Thurber for advice on running workshops. We thank Julia Baum, David Baker, Ramkumar Balakrishnan, Arthur Grossman, Sridhar Jayavel, Thomas Krueger, Mikhail Matz, Rajesh Kannan Murugesan, Ramakritinan Chockalingam Muthiah, Sivakumar Natesan, and Michael Sweet for feedback throughout the development of this work. We appreciate reviewer feedback we received on the manuscript from Tom Oliver, an anonymous reviewer, and Rob Toonen (handling editor at *PeerJ*). Finally, we thank all Symbiodiniaceae researchers, past and present, whose research has helped shape our understanding of Symbiodiniaceae diversity and biology.

### Funding

Funding for the virtual workshop during which the content of this manuscript was developed was provided by the National Science Foundation Division of Biological Oceanography (OCE 2127506; 2127514; and 2127508 to Sarah W. Davies, Adrienne M. S. Correa, and John Everett Parkinson, respectively). The funders had no role in study design, data collection and analysis, decision to publish, or preparation of the manuscript.

### Grant Disclosures

The following grant information was disclosed by the authors:
National Science Foundation Division of Biological Oceanography: OCE 2127506, 2127514 and 2127508.

## Competing Interests

Anastazia T. Banaszak and James Davis Reimer are Academic Editors for PeerJ.

## Author Contributions

- Sarah W. Davies conceived and designed the experiments, performed the experiments, prepared figures and/or tables, authored or reviewed drafts of the article, and approved the final draft.
- Matthew H. Gamache conceived and designed the experiments, performed the experiments, prepared figures and/or tables, authored or reviewed drafts of the article, and approved the final draft.
- Lauren I. Howe-Kerr conceived and designed the experiments, performed the experiments, prepared figures and/or tables, authored or reviewed drafts of the article, and approved the final draft.
- Nicola G. Kriefall conceived and designed the experiments, performed the experiments, prepared figures and/or tables, authored or reviewed drafts of the article, and approved the final draft.
- Andrew C. Baker performed the experiments, authored or reviewed drafts of the article, and approved the final draft.
- Anastazia T. Banaszak performed the experiments, authored or reviewed drafts of the article, and approved the final draft.
- Line Kolind Bay performed the experiments, authored or reviewed drafts of the article, and approved the final draft.
- Anthony J. Bellantuono performed the experiments, authored or reviewed drafts of the article, and approved the final draft.
- Debashish Bhattacharya performed the experiments, authored or reviewed drafts of the article, and approved the final draft.
- Cheong Xin Chan performed the experiments, authored or reviewed drafts of the article, and approved the final draft.
- Danielle C. Claar performed the experiments, authored or reviewed drafts of the article, and approved the final draft.
- Mary Alice Coffroth performed the experiments, authored or reviewed drafts of the article, and approved the final draft.
- Ross Cunning performed the experiments, authored or reviewed drafts of the article, and approved the final draft.
- Simon K. Davy performed the experiments, authored or reviewed drafts of the article, and approved the final draft.
- Javier del Campo performed the experiments, authored or reviewed drafts of the article, and approved the final draft.
- Erika M. Díaz-Almeyda performed the experiments, authored or reviewed drafts of the article, and approved the final draft.
- Jörg C. Frommlet performed the experiments, authored or reviewed drafts of the article, and approved the final draft.

- Lauren E. Fuess performed the experiments, authored or reviewed drafts of the article, and approved the final draft.
- Raúl A. González-Pech performed the experiments, authored or reviewed drafts of the article, and approved the final draft.
- Tamar L. Goulet performed the experiments, authored or reviewed drafts of the article, and approved the final draft.
- Kenneth D. Hoadley performed the experiments, authored or reviewed drafts of the article, and approved the final draft.
- Emily J. Howells performed the experiments, authored or reviewed drafts of the article, and approved the final draft.
- Benjamin C. C. Hume performed the experiments, authored or reviewed drafts of the article, and approved the final draft.
- Dustin W. Kemp performed the experiments, prepared figures and/or tables, authored or reviewed drafts of the article, and approved the final draft.
- Carly D. Kenkel performed the experiments, authored or reviewed drafts of the article, and approved the final draft.
- Sheila A. Kitchen performed the experiments, authored or reviewed drafts of the article, and approved the final draft.
- Todd C. LaJeunesse performed the experiments, authored or reviewed drafts of the article, and approved the final draft.
- Senjie Lin performed the experiments, authored or reviewed drafts of the article, and approved the final draft.
- Shelby E. McIlroy performed the experiments, authored or reviewed drafts of the article, and approved the final draft.
- Ryan McMinds performed the experiments, authored or reviewed drafts of the article, and approved the final draft.
- Matthew R. Nitschke performed the experiments, authored or reviewed drafts of the article, and approved the final draft.
- Clinton A. Oakley performed the experiments, authored or reviewed drafts of the article, and approved the final draft.
- Raquel S. Peixoto performed the experiments, authored or reviewed drafts of the article, and approved the final draft.
- Carlos Prada performed the experiments, authored or reviewed drafts of the article, and approved the final draft.
- Hollie M. Putnam performed the experiments, authored or reviewed drafts of the article, and approved the final draft.
- Kate Quigley performed the experiments, authored or reviewed drafts of the article, and approved the final draft.
- Hannah G. Reich performed the experiments, authored or reviewed drafts of the article, and approved the final draft.

- James Davis Reimer performed the experiments, authored or reviewed drafts of the article, and approved the final draft.
- Mauricio Rodriguez-Lanetty performed the experiments, authored or reviewed drafts of the article, and approved the final draft.
- Stephanie M. Rosales performed the experiments, authored or reviewed drafts of the article, and approved the final draft.
- Osama S. Saad performed the experiments, authored or reviewed drafts of the article, and approved the final draft.
- Eugenia M. Sampayo performed the experiments, authored or reviewed drafts of the article, and approved the final draft.
- Scott R. Santos performed the experiments, prepared figures and/or tables, authored or reviewed drafts of the article, and approved the final draft.
- Eiichi Shoguchi performed the experiments, authored or reviewed drafts of the article, and approved the final draft.
- Edward G. Smith performed the experiments, authored or reviewed drafts of the article, and approved the final draft.
- Michael Stat performed the experiments, authored or reviewed drafts of the article, and approved the final draft.
- Timothy G. Stephens performed the experiments, authored or reviewed drafts of the article, and approved the final draft.
- Marie E. Strader performed the experiments, authored or reviewed drafts of the article, and approved the final draft.
- David J. Suggett performed the experiments, prepared figures and/or tables, authored or reviewed drafts of the article, and approved the final draft.
- Timothy D. Swain performed the experiments, authored or reviewed drafts of the article, and approved the final draft.
- Cawa Tran performed the experiments, authored or reviewed drafts of the article, and approved the final draft.
- Nikki Traylor-Knowles performed the experiments, authored or reviewed drafts of the article, and approved the final draft.
- Christian R. Voolstra performed the experiments, authored or reviewed drafts of the article, and approved the final draft.
- Mark E. Warner performed the experiments, prepared figures and/or tables, authored or reviewed drafts of the article, and approved the final draft.
- Virginia M. Weis performed the experiments, authored or reviewed drafts of the article, and approved the final draft.
- Rachel M. Wright performed the experiments, authored or reviewed drafts of the article, and approved the final draft.
- Tingting Xiang performed the experiments, authored or reviewed drafts of the article, and approved the final draft.
- Hiroshi Yamashita performed the experiments, authored or reviewed drafts of the article, and approved the final draft.

- Maren Ziegler performed the experiments, authored or reviewed drafts of the article, and approved the final draft.
- Adrienne M. S. Correa conceived and designed the experiments, performed the experiments, prepared figures and/or tables, authored or reviewed drafts of the article, and approved the final draft.
- John Everett Parkinson conceived and designed the experiments, performed the experiments, prepared figures and/or tables, authored or reviewed drafts of the article, and approved the final draft.

## Data Availability

This is a literature review.

## Supplemental Information

Supplemental information for this article can be found online at http://dx.doi.org/10.7717/peerj.15023#supplemental-information.

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

# PeerJ

**Cunning R, Silverstein RN, Baker AC. 2018.** Symbiont shuffling linked to differential photochemical dynamics of *Symbiodinium* in three Caribbean reef corals. *Coral Reefs* **37**:145–152 DOI 10.1007/s00338-017-1640-3.

**Cunning R, Yost DM, Guarinello ML, Putnam HM, Gates RD. 2015.** Variability of *Symbiodinium* communities in waters, sediments, and corals of thermally distinct reef pools in American Samoa. *PLOS ONE* **10**:e0145099 DOI 10.1371/journal.pone.0145099.

**Cziesielski MJ, Liew YJ, Cui G, Schmidt-Roach S, Campana S, Marondedze C, Aranda M. 2018.** Multi-omics analysis of thermal stress response in a zooxanthellate cnidarian reveals the importance of associating with thermotolerant symbionts. *Proceedings of the Royal Society B: Biological Sciences* **285**:20172654 DOI 10.1098/rspb.2017.2654.

**Davies SW, Moreland KN, Wham DC, Kanke MR, Matz MV. 2020.** *Cladocopium* community divergence in two *Acropora* coral hosts across multiple spatial scales. *Molecular Ecology* **29**:4559–4572 DOI 10.1111/mec.15668.

**Davies SW, Putnam HM, Ainsworth T, Baum JK, Bove CB, Crosby SC, Côté IM, Duplouy A, Fulweiler RW, Griffin AJ, Hanley TC, Hill T, Humanes A, Mangubhai S, Metaxas A, Parker LM, Rivera HE, Silbiger NJ, Smith NS, Spalding AK, Traylor-Knowles N, Weigel BL, Wright RM, Bates AE. 2021.** Promoting inclusive metrics of success and impact to dismantle a discriminatory reward system in science. *PLOS Biology* **19**:e3001282 DOI 10.1371/journal.pbio.3001282.

**Davies SW, Rahman M, Meyer E, Green EA, Buschiazzo E, Medina M, Matz MV. 2013.** Novel polymorphic microsatellite markers for population genetics of the endangered Caribbean star coral, *Montastraea faveolata*. *Marine Biodiversity* **43**:167–172 DOI 10.1007/s12526-012-0133-4.

**Davies SW, Ries JB, Marchetti A, Castillo KD. 2018.** *Symbiodinium* functional diversity in the coral *Siderastrea siderea* is influenced by thermal stress and reef environment, but not ocean acidification. *Frontiers in Marine Science* **5**:150 DOI 10.3389/fmars.2018.00150.

**Davy Simon K, Allemand D, Weis Virginia M. 2012.** Cell biology of cnidarian-dinoflagellate symbiosis. *Microbiology and Molecular Biology Reviews* **76**:229–261 DOI 10.1128/MMBR.05014-11.

**de Mendoza A, Bonnet A, Vargas-Landin DB, Ji N, Li H, Yang F, Li L, Hori K, Pflueger J, Buckberry S, Ohta H, Rosic N, Lesage P, Lin S, Lister R. 2018.** Recurrent acquisition of cytosine methyltransferases into eukaryotic retrotransposons. *Nature Communications* **9**:1341 DOI 10.1038/s41467-018-03724-9.

**De Queiroz K. 2007.** Species concepts and species delimitation. *Systematic Biology* **56**:879–886 DOI 10.1080/10635150701701083.

**de Vos A. 2020.** *The problem of "colonial science"*. New York: Scientific American.

**Díaz-Almeyda EM, Prada C, Ohdera AH, Moran H, Civitello DJ, Iglesias-Prieto R, Carlo TA, LaJeunesse TC, Medina M. 2017.** Intraspecific and interspecific variation in thermotolerance and photoacclimation in *Symbiodinium* dinoflagellates. *Proceedings of the Royal Society B: Biological Sciences* **284**:20171767 DOI 10.1098/rspb.2017.1767.

**Di Bitetti MS, Ferreras JA. 2017.** Publish (in English) or perish: the effect on citation rate of using languages other than English in scientific publications. *AMBIO a Journal of the Human Environment* **46**:121–127 DOI 10.1007/s13280-016-0820-7.

**Di Genio S, Wang LH, Meng PJ, Tsai S, Lin C. 2021.** "Symbio-Cryobank": toward the development of a cryogenic archive for the coral reef dinoflagellate symbiont Symbiodiniaceae. *Biopreservation and Biobanking* **19**:91–93 DOI 10.1089/bio.2020.0071.

**Dimond JL, Holzman BJ, Bingham BL. 2012.** Thicker host tissues moderate light stress in a cnidarian endosymbiont. *Journal of Experimental Biology* **215**:2247–2254 DOI 10.1242/jeb.067991.

**Dimond JL, Kerwin AH, Rotjan R, Sharp K, Stewart FJ, Thornhill DJ. 2013.** A simple temperature-based model predicts the upper latitudinal limit of the temperate coral *Astrangia poculata*. *Coral Reefs* **32**:401–409 DOI 10.1007/s00338-012-0983-z.

**Dixon GB, Bay LK, Matz MV. 2014.** Bimodal signatures of germline methylation are linked with gene expression plasticity in the coral *Acropora millepora*. *BMC Genomics* **15**:1109 DOI 10.1186/1471-2164-15-1109.

**Dixon G, Liao Y, Bay LK, Matz MV. 2018.** Role of gene body methylation in acclimatization and adaptation in a basal metazoan. *Proceedings of the National Academy of Sciences of the United States of America* **115**:13342–13346 DOI 10.1073/pnas.1813749115.

**Dixon G, Matz M. 2021.** Benchmarking DNA methylation assays in a reef-building coral. *Molecular Ecology Resources* **21**:464–477 DOI 10.1111/1755-0998.13282.

**Dougan KE, González-Pech RA, Stephens TG, Shah S, Chen Y, Ragan MA, Bhattacharya D, Chan CX. 2022.** Genome-powered classification of microbial eukaryotes: focus on coral algal symbionts. *Trends in Microbiology* **30(9)**:831–840 DOI 10.1016/j.tim.2022.02.001.

**Dunkley TP, Watson R, Griffin JL, Dupree P, Lilley KS. 2004.** Localization of organelle proteins by isotope tagging (LOPIT). *Molecular & Cellular Proteomics* **3(11)**:1128–1134 DOI 10.1074/mcp.T400009-MCP200.

**Durante MK, Baums IB, Williams DE, Kemp DW, Vohsen S. 2019.** What drives phenotypic divergence among coral clonemates of *Acropora palmata*? *Molecular Ecology* **28**:3208–3224 DOI 10.1111/mec.15140.

**Dzirasa K. 2020.** Revising the *a priori* hypothesis: systemic racism has penetrated scientific funding. *Cell* **183**:576–579 DOI 10.1016/j.cell.2020.09.026.

**Eaton AA, Saunders JF, Jacobson RK, West K. 2020.** How gender and race stereotypes impact the advancement of scholars in STEM: professors' biased evaluations of physics and biology post-doctoral candidates. *Sex Roles* **82**:127–141 DOI 10.1007/s11199-019-01052-w.

**Ebert P, Audano PA, Zhu Q, Rodriguez-Martin B, Porubsky D, Bonder MJ, Sulovari A, Ebler J, Zhou W, Serra Mari R, Yilmaz F, Zhao X, Hsieh P, Lee J, Kumar S, Lin J, Rausch T, Chen Y, Ren J, Santamarina M, Höps W, Ashraf H, Chuang NT, Yang X, Munson KM, Lewis AP, Fairley S, Tallon LJ, Clarke WE, Basile AO, Byrska-Bishop M, Corvelo A, Evani US, Lu TY, Chaisson MJP, Chen J, Li C, Brand H, Wenger AM, Ghareghani M, Harvey WT, Raeder B, Hasenfeld P, Regier AA, Abel HJ, Hall IM, Flicek P, Stegle O, Gerstein MB, Tubio JMC, Mu Z, Li YI, Shi X, Hastie AR, Ye K, Chong Z, Sanders AD, Zody MC, Talkowski ME, Mills RE, Devine SE, Lee C, Korbel JO, Marschall T, Eichler EE. 2021.** Haplotype-resolved diverse human genomes and integrated analysis of structural variation. *Science* **372**:eabf7117 DOI 10.1126/science.abf7117.

**Eckert RJ, Reaume AM, Sturm AB, Studivan MS, Voss JD. 2020.** Depth influences Symbiodiniaceae associations among *Montastraea cavernosa* corals on the Belize Barrier Reef. *Frontiers in Microbiology* **11**:518 DOI 10.3389/fmicb.2020.00518.

**Egan S, Gardiner M. 2016.** Microbial dysbiosis: rethinking disease in marine ecosystems. *Frontiers in Microbiology* **7**:991 DOI 10.3389/fmicb.2016.00991.

**Enríquez S, Méndez ER, Iglesias-Prieto R. 2005.** Multiple scattering on coral skeletons enhances light absorption by symbiotic algae. *Limnology and Oceanography* **50**:1025–1032 DOI 10.4319/lo.2005.50.4.1025.

**Epstein HE, Torda G, van Oppen MJH. 2019.** Relative stability of the *Pocillopora acuta* microbiome throughout a thermal stress event. *Coral Reefs* **38**:373–386 DOI 10.1007/s00338-019-01783-y.

**Faktorová D, Nisbet RER, Fernández Robledo JA, Casacuberta E, Sudek L, Allen AE, Ares M Jr, Aresté C, Balestreri C, Barbrook AC, Beardslee P, Bender S, Booth DS, Bouget FY, Bowler C, Breglia SA, Brownlee C, Burger G, Cerutti H, Cesaroni R, Chiurillo MA, Clemente T, Coles DB, Collier JL, Cooney EC, Coyne K, Docampo R, Dupont CL, Edgcomb V, Einarsson E, Elustondo PA, Federici F, Freire-Beneitez V, Freyria NJ, Fukuda K, García PA, Girguis PR, Gomaa F, Gornik SG, Guo J, Hampl V, Hanawa Y, Haro-Contreras ER, Hehenberger E, Highfield A, Hirakawa Y, Hopes A, Howe CJ, Hu I, Ibañez J, Irwin NAT, Ishii Y, Janowicz NE, Jones AC, Kachale A, Fujimura-Kamada K, Kaur B, Kaye JZ, Kazana E, Keeling PJ, King N, Klobutcher LA, Lander N, Lassadi I, Li Z, Lin S, Lozano J-C, Luan F, Maruyama S, Matute T, Miceli C, Minagawa J, Moosburner M, Najle SR, Nanjappa D, Nimmo IC, Noble L, Novák Vanclová AMG, Nowacki M, Nuñez I, Pain A, Piersanti A, Pucciarelli S, Pyrih J, Rest JS, Rius M, Robertson D, Ruaud A, Ruiz-Trillo I, Sigg MA, Silver PA, Slamovits CH, Jason Smith G, Sprecher BN, Stern R, Swart EC, Tsaousis AD, Tsypin L, Turkewitz A, Turnšek J, Valach M, Vergé V, von Dassow P, von der Haar T, Waller RF, Wang L, Wen X, Wheeler G, Woods A, Zhang H, Mock T, Worden AZ, Lukeš J. 2020.** Genetic tool development in marine protists: emerging model organisms for experimental cell biology. *Nature Methods* **17**:481–494 DOI 10.1038/s41592-020-0796-x.

**Feder ME, Walser JC. 2005.** The biological limitations of transcriptomics in elucidating stress and stress responses. *Journal of Evolutionary Biology* **18**:901–910 DOI 10.1111/j.1420-9101.2005.00921.x.

**Feil R, Fraga MF. 2012.** Epigenetics and the environment: emerging patterns and implications. *Nature Reviews Genetics* **13**:97–109 DOI 10.1038/nrg3142.

**Fifer JE, Bui V, Berg JT, Kriefall N, Klepac C, Bentlage B, Davies SW. 2022.** Microbiome structuring within a coral colony and along a sedimentation gradient. *Frontiers in Marine Science* **8**:2071 DOI 10.3389/fmars.2021.805202.

**Figueroa RI, Howe-Kerr LI, Correa AMS. 2021.** Direct evidence of sex and a hypothesis about meiosis in Symbiodiniaceae. *Scientific Reports* **11**:18838 DOI 10.1038/s41598-021-98148-9.

**Finney JC, Pettay DT, Sampayo EM, Warner ME, Oxenford HA, LaJeunesse TC. 2010.** The relative significance of host–habitat, depth, and geography on the ecology, endemism, and speciation of coral endosymbionts in the genus *Symbiodinium*. *Microbial Ecology* **60**:250–263 DOI 10.1007/s00248-010-9681-y.

**Fitt WK. 2000.** Cellular growth of host and symbiont in a cnidarian-zooxanthellar symbiosis. *Biological Bulletin* **198**:110–120 DOI 10.2307/1542809.

**Fitt WK, McFarland FK, Warner ME, Chilcoat GC. 2000.** Seasonal patterns of tissue biomass and densities of symbiotic dinoflagellates in reef corals and relation to coral bleaching. *Limnology and Oceanography* **45(3)**:677–685 DOI 10.4319/lo.2000.45.3.0677.

**Flowerdew J. 2001.** Attitudes of journal editors to nonnative speaker contributions. *TESOL Quarterly* **35(1)**:121–150 DOI 10.2307/3587862.

**Foret S, Kucharski R, Pellegrini M, Feng S, Jacobsen SE, Robinson GE, Maleszka R. 2012.** DNA methylation dynamics, metabolic fluxes, gene splicing, and alternative phenotypes in honey bees. *Proceedings of the National Academy of Sciences of the United States of America* **109(13)**:4968–4973 DOI 10.1073/pnas.1202392109.

**Forsman ZH, Ritson-Williams R, Tisthammer KH, Knapp ISS, Toonen RJ. 2020.** Host-symbiont coevolution, cryptic structure, and bleaching susceptibility, in a coral species complex (Scleractinia; Poritidae). *Scientific Reports* **10(1)**:1–12 DOI 10.1038/s41598-020-73501-6.

**Fox CW, Burns CS, Muncy AD, Meyer JA. 2017.** Author-suggested reviewers: gender differences and influences on the peer review process at an ecology journal. *Functional Ecology* **31(1)**:270–280 DOI 10.1111/1365-2435.12665.

**Fox CW, Duffy MA, Fairbairn DJ, Meyer JA. 2019.** Gender diversity of editorial boards and gender differences in the peer review process at six journals of ecology and evolution. *Ecology and Evolution* **9**:13636–13649 DOI 10.1002/ece3.5794.

**Frade PR, De Jongh F, Vermeulen F, van Bleijswijk J, Bak RPM. 2008.** Variation in symbiont distribution between closely related coral species over large depth ranges. *Molecular Ecology* **17**:691–703 DOI 10.1111/j.1365-294X.2007.03612.x.

**Freudenthal HD. 1962.** *Symbiodinium* gen. nov. and *Symbiodinium microadriaticum* sp. nov., a Zooxanthella: taxonomy, life cycle, and morphology. *Journal of Protozoology* **9**:45–52 DOI 10.1111/j.1550-7408.1962.tb02579.x.

**Frommlet JC, Sousa ML, Alves A, Vieira SI, Suggett DJ, Serôdio J. 2015.** Coral symbiotic algae calcify ex hospite in partnership with bacteria. *Proceedings of the National Academy of Sciences of the United States of America* **112**:6158–6163 DOI 10.1073/pnas.1420991112.

**Frøslev TG, Kjøller R, Bruun HH, Ejrnæs R, Brunbjerg AK, Pietroni C, Hansen AJ. 2017.** Algorithm for post-clustering curation of DNA amplicon data yields reliable biodiversity estimates. *Nature Communications* **8**:1188 DOI 10.1038/s41467-017-01312-x.

**Fujise L, Nitschke MR, Frommlet JC, Serôdio J, Woodcock S, Ralph PJ, Suggett DJ. 2018.** Cell cycle dynamics of cultured coral endosymbiotic microalgae (*Symbiodinium*) across different types (species) under alternate light and temperature conditions. *Journal of Eukaryotic Microbiology* **65**:505–517 DOI 10.1111/jeu.12497.

**Fujise L, Suggett DJ, Stat M, Kahlke T, Bunce M, Gardner SG, Goyen S, Woodcock S, Ralph PJ, Seymour JR, Siboni N, Nitschke MR. 2021.** Unlocking the phylogenetic diversity, primary habitats, and abundances of free-living Symbiodiniaceae on a coral reef. *Molecular Ecology* **30**:343–360 DOI 10.1111/mec.15719.

**Fujiwara Y, Kawamura I, Reimer JD, Parkinson JE. 2021.** Zoantharian endosymbiont community dynamics during a stress event. *Frontiers in Microbiology* **12**:1282 DOI 10.3389/fmicb.2021.674026.

**Gabay Y, Parkinson JE, Wilkinson SP, Weis VM, Davy SK. 2019.** Inter-partner specificity limits the acquisition of thermotolerant symbionts in a model cnidarian-dinoflagellate symbiosis. *The ISME Journal* **13**:2489–2499 DOI 10.1038/s41396-019-0429-5.

**Geladaki A, Kočevar Britovšek N, Breckels LM, Smith TS, Vennard OL, Mulvey CM, Crook OM, Gatto L, Lilley KS. 2019.** Combining LOPIT with differential ultracentrifugation for high-resolution spatial proteomics. *Nature Communications* **10(1)**:1–15 DOI 10.1038/s41467-018-08191-w.

**Ginther DK, Schaffer WT, Schnell J, Masimore B, Liu F, Haak LL, Kington R. 2011.** Race, ethnicity, and NIH research awards. *Science* **333**:1015–1019 DOI 10.1126/science.1196783.

**Glynn PW. 1996.** Coral reef bleaching: facts, hypotheses and implications. *Global Change Biology* **2**:495–509 DOI 10.1111/j.1365-2486.1996.tb00063.x.

**Gong S, Zhang F, Li Z. 2018.** Genotype and phylogenetic diversity of *Symbiodinium* ITS2 sequences within Clade C in three typical coral species from Luhuitou Fringing Reef of the South China Sea. *Journal of Ocean University of China* **17**:1411–1417 DOI 10.1007/s11802-018-3628-1.

**Gong W, Marchetti A. 2019.** Estimation of 18S gene copy number in marine eukaryotic plankton using a next-generation sequencing approach. *Frontiers in Marine Science* **6**:219 DOI 10.3389/fmars.2019.00219.

**González-Pech RA, Bhattacharya D, Ragan MA, Chan CX. 2019.** Genome evolution of coral reef symbionts as intracellular residents. *Trends in Ecology & Evolution* **34**:799–806 DOI 10.1016/j.tree.2019.04.010.

**González-Pech RA, Stephens TG, Chen Y, Mohamed AR, Cheng Y, Shah S, Dougan KE, Fortuin MDA, Lagorce R, Burt DW, Bhattacharya D, Ragan MA, Chan CX. 2021.** Comparison of 15 dinoflagellate genomes reveals extensive sequence and structural divergence in family Symbiodiniaceae and genus *Symbiodinium*. *BMC Biology* **19**:73 DOI 10.1186/s12915-021-00994-6.

**Gordin MD. 2015.** *Scientific babel: how science was done before and after global english.* Chicago: University of Chicago Press.

**Gornick SG, Maegele I, Hambleton EA, Voss PA, Waller RF, Guse A. 2022.** Nuclear transformation of a dinoflagellate symbiont of corals. *Frontiers in Marine Science* **9**:1035413 DOI 10.3389/fmars.2022.1035413.

**Goulet T, Coffroth M. 2003.** Genetic composition of zooxanthellae between and within colonies of the octocoral *Plexaura kuna*, based on small subunit rDNA and multilocus DNA fingerprinting. *Marine Biology* **142**:233–239 DOI 10.1007/s00227-002-0936-0.

**Goulet TL. 2006.** Most corals may not change their symbionts. *Marine Ecology Progress Series* **321**:1–7 DOI 10.3354/meps321001.

**Goulet TL, Cook CB, Goulet D. 2005.** Effect of short-term exposure to elevated temperatures and light levels on photosynthesis of different host-symbiont combinations in the *Aiptasia pallida/Symbiodinium* symbiosis. *Limnology and Oceanography* **50**:1490–1498 DOI 10.4319/lo.2005.50.5.1490.

**Goulet TL, Lucas MQ, Schizas NV. 2019.** Symbiodiniaceae genetic diversity and symbioses with hosts from shallow to mesophotic coral ecosystems. In: Loya Y, Puglise KA, Bridge TCL, eds. *Mesophotic Coral Ecosystems.* Cham: Springer International Publishing, 537–551.

**Goulet TL, Shirur KP, Ramsby BD, Iglesias-Prieto R. 2017.** The effects of elevated seawater temperatures on Caribbean gorgonian corals and their algal symbionts, *Symbiodinium* spp. *PLOS ONE* **12**:e0171032 DOI 10.1371/journal.pone.0171032.

**Goyen S, Pernice M, Szabó M, Warner ME, Ralph PJ, Suggett DJ. 2017.** A molecular physiology basis for functional diversity of hydrogen peroxide production amongst *Symbiodinium* spp. (Dinophyceae). *Marine Biology* **164**:1–12 DOI 10.1007/s00227-017-3073-5.

**Grajales A, Rodríguez E, Thornhill DJ. 2016.** Patterns of *Symbiodinium* spp. associations within the family Aiptasiidae, a monophyletic lineage of symbiotic of sea anemones (Cnidaria, Actiniaria). *Coral Reefs* **35**:345–355 DOI 10.1007/s00338-015-1352-5.

**Granados-Cifuentes C, Neigel J, Leberg P, Rodriguez-Lanetty M. 2015.** Genetic diversity of free-living *Symbiodinium* in the Caribbean: the importance of habitats and seasons. *Coral Reefs* **34**:927–939 DOI 10.1007/s00338-015-1291-1.

**Green EA, Davies SW, Matz MV, Medina M. 2014.** Quantifying cryptic *Symbiodinium* diversity within *Orbicella faveolata* and Orbicella franksi at the Flower Garden Banks, Gulf of Mexico. *PeerJ* **2**:e386 DOI 10.7717/peerj.386.

**Grégoire V, Schmacka F, Coffroth MA, Karsten U. 2017.** Photophysiological and thermal tolerance of various genotypes of the coral endosymbiont *Symbiodinium* sp. (Dinophyceae). *Journal of Applied Phycology* **29**:1893–1905 DOI 10.1007/s10811-017-1127-1.

Greider CW, Sheltzer JM, Cantalupo NC, Copeland WB, Dasgupta N, Hopkins N, Jansen JM, Joshua-Tor L, McDowell GS, Metcalf JL, McLaughlin B, Olivarius A, O'Shea EK, Raymond JL, Ruebain D, Steitz JA, Stillman B, Tilghman SM, Valian V, Villa-Komaroff L, Wong JY. 2019. Increasing gender diversity in the STEM research workforce. *Science* **366**:692–695 DOI 10.1126/science.aaz0649.

Grinnell M, Higgins S, Yost K, Ochuba O, Lobl M, Grimes P, Wysong A. 2020. The proportion of male and female editors in women's health journals: a critical analysis and review of the sex gap. *International Journal of Women's Dermatology* **6**:7–12 DOI 10.1016/j.ijwd.2019.11.005.

Grupstra CGB, Coma R, Ribes M, Leydet KP, Parkinson JE, McDonald K, Catllà M, Voolstra CR, Hellberg ME, Coffroth MA. 2017. Evidence for coral range expansion accompanied by reduced diversity of *Symbiodinium* genotypes. *Coral Reefs* **36**:981–985 DOI 10.1007/s00338-017-1589-2.

Grupstra CGB, Howe-Kerr LI, Veglia AJ, Bryant RL, Coy SR, Blackwelder PL, Correa AMS. 2022a. Thermal stress triggers productive viral infection of a key coral reef symbiont. *The ISME Journal* **16**:1430–1441 DOI 10.1038/s41396-022-01194-y.

Grupstra CGB, Lemoine NP, Cook C, Correa AMS. 2022b. Thank you for biting: dispersal of beneficial microbiota through "antagonistic" interactions. *Trends in Microbiology* **30**:930–939 DOI 10.1016/j.tim.2022.03.006.

Grupstra CGB, Rabbitt KM, Howe-Kerr LI, Correa AMS. 2021. Fish predation on corals promotes the dispersal of coral symbionts. *Animal Microbiome* **3**:25 DOI 10.1186/s42523-021-00086-4.

Haelewaters D, Hofmann TA, Romero-Olivares AL. 2021. Ten simple rules for Global North researchers to stop perpetuating helicopter research in the Global South. *PLOS Computational Biology* **17**:e1009277 DOI 10.1371/journal.pcbi.1009277.

Hafeez DM, Waqas A, Majeed S, Naveed S, Afzal KI, Aftab Z, Zeshan M, Khosa F. 2019. Gender distribution in psychiatry journals' editorial boards worldwide. *Comprehensive Psychiatry* **94**:152119 DOI 10.1016/j.comppsych.2019.152119.

Hale ML, Burg TM, Steeves TE. 2012. Sampling for microsatellite-based population genetic studies: 25 to 30 individuals per population is enough to accurately estimate allele frequencies. *PLOS ONE* **7**:e45170 DOI 10.1371/journal.pone.0045170.

Hawkins TD, Hagemeyer JCG, Warner ME. 2016. Temperature moderates the infectiousness of two conspecific *Symbiodinium* strains isolated from the same host population. *Environmental Microbiology* **18**:5204–5217 DOI 10.1111/1462-2920.13535.

Hennige SJ, Suggett DJ, Warner ME, McDougall KE, Smith DJ. 2009. Photobiology of *Symbiodinium* revisited: bio-physical and bio-optical signatures. *Coral Reefs* **28**:179–195 DOI 10.1007/s00338-008-0444-x.

Hernandez-Agreda A, Gates RD, Ainsworth TD. 2017. Defining the core microbiome in corals' microbial soup. *Trends in Microbiology* **25**:125–140 DOI 10.1016/j.tim.2016.11.003.

Hill M, Allenby A, Ramsby B, Schönberg C, Hill A. 2011. *Symbiodinium* diversity among host clionaid sponges from Caribbean and Pacific reefs: evidence of heteroplasmy and putative host-specific symbiont lineages. *Molecular Phylogenetics and Evolution* **59**:81–88 DOI 10.1016/j.ympev.2011.01.006.

Hirose M, Reimer JD, Hidaka M, Suda S. 2008. Phylogenetic analyses of potentially free-living *Symbiodinium* spp. isolated from coral reef sand in Okinawa, Japan. *Marine Biology* **155**:105–112 DOI 10.1007/s00227-008-1011-2.

Hoadley KD, Lewis AM, Wham DC, Pettay DT, Grasso C, Smith R, Kemp DW, LaJeunesse TC, Warner ME. 2019. Host-symbiont combinations dictate the photo-physiological response of

reef-building corals to thermal stress. *Scientific Reports* **9(1)**:9985
DOI 10.1038/s41598-019-46412-4.

**Hoadley KD, Pettay DT, Lewis A, Wham D, Grasso C, Smith R, Kemp DW, LaJeunesse T, Warner ME. 2021.** Different functional traits among closely related algal symbionts dictate stress endurance for vital Indo-Pacific reef-building corals. *Global Change Biology* **27(20)**:5295–5309 DOI 10.1111/gcb.15799.

**Hofstra B, Kulkarni VV, Munoz-Najar Galvez S, He B, Jurafsky D, McFarland DA. 2020.** The diversity-innovation paradox in science. *Proceedings of the National Academy of Sciences of the United States of America* **117(17)**:9284–9291 DOI 10.1073/pnas.1915378117.

**Hoppe TA, Litovitz A, Willis KA, Meseroll RA, Perkins MJ, Hutchins BI, Davis AF, Lauer MS, Valantine HA, Anderson JM, Santangelo GM. 2019.** Topic choice contributes to the lower rate of NIH awards to African-American/black scientists. *Science Advances* **5(10)**:eaaw7238 DOI 10.1126/sciadv.aaw7238.

**Howe-Kerr LI, Bachelot B, Wright RM, Kenkel CD, Bay LK, Correa AMS. 2020.** Symbiont community diversity is more variable in corals that respond poorly to stress. *Global Change Biology* **26(4)**:2220–2234 DOI 10.1111/gcb.14999.

**Howells EJ, Bauman AG, Vaughan GO, Hume BCC, Voolstra CR, Burt JA. 2020.** Corals in the hottest reefs in the world exhibit symbiont fidelity not flexibility. *Molecular Ecology* **29**:889–911 DOI 10.1111/mec.15372.

**Howells EJ, Beltran VH, Larsen NW, Bay LK, Willis BL, van Oppen MJH. 2011.** Coral thermal tolerance shaped by local adaptation of photosymbionts. *Nature Climate Change* **2**:116–120 DOI 10.1038/nclimate1330.

**Howells EJ, Berkelmans R, van Oppen MJH, Willis BL, Bay LK. 2013a.** Historical thermal regimes define limits to coral acclimatization. *Ecology* **94**:1078–1088 DOI 10.1890/12-1257.1.

**Howells EJ, van Oppen MJH, Willis BL. 2009.** High genetic differentiation and cross-shelf patterns of genetic diversity among Great Barrier Reef populations of *Symbiodinium*. *Coral Reefs* **28**:215–225 DOI 10.1007/s00338-008-0450-z.

**Howells EJ, Willis BL, Bay LK, van Oppen MJH. 2016.** Microsatellite allele sizes alone are insufficient to delineate species boundaries in *Symbiodinium*. *Molecular Ecology* **25**:2719–2723 DOI 10.1111/mec.13631.

**Howells EJ, Willis BL, Bay LK, van Oppen MJH. 2013b.** Spatial and temporal genetic structure of *Symbiodinium* populations within a common reef-building coral on the Great Barrier Reef. *Molecular Ecology* **22**:3693–3708 DOI 10.1111/mec.12342.

**Huang J, Gates AJ, Sinatra R, Barabási AL. 2020.** Historical comparison of gender inequality in scientific careers across countries and disciplines. *Proceedings of the National Academy of Sciences of the United States of America* **117**:4609–4616 DOI 10.1073/pnas.1914221117.

**Hughes TP, Barnes ML, Bellwood DR, Cinner JE, Cumming GS, Jackson JBC, Kleypas J, van de Leemput IA, Lough JM, Morrison TH, Palumbi SR, van Nes EH, Scheffer M. 2017.** Coral reefs in the Anthropocene. *Nature* **546**:82–90 DOI 10.1038/nature22901.

**Hume BCC, D'Angelo C, Smith EG, Stevens JR, Burt J, Wiedenmann J. 2015.** *Symbiodinium thermophilum* sp. nov., a thermotolerant symbiotic alga prevalent in corals of the world's hottest sea, the Persian/Arabian Gulf. *Scientific Reports* **5**:8562 DOI 10.1038/srep08562.

**Hume BCC, Mejia-Restrepo A, Voolstra CR, Berumen ML. 2020.** Fine-scale delineation of Symbiodiniaceae genotypes on a previously bleached central Red Sea reef system demonstrates a prevalence of coral host-specific associations. *Coral Reefs* **39**:583–601 DOI 10.1007/s00338-020-01917-7.

**Hume BCC, Smith EG, Ziegler M, Warrington HJM, Burt JA, LaJeunesse TC, Wiedenmann J, Voolstra CR. 2019.** SymPortal: a novel analytical framework and platform for coral algal symbiont next-generation sequencing ITS2 profiling. *Molecular Ecology Resources* **19**:1063–1080 DOI 10.1111/1755-0998.13004.

**Hume BCC, Ziegler M, Poulain J, Pochon X, Romac S, Boissin E, de Vargas C, Planes S, Wincker P, Voolstra CR. 2018.** An improved primer set and amplification protocol with increased specificity and sensitivity targeting the *Symbiodinium* ITS2 region. *PeerJ* **6**:e4816 DOI 10.7717/peerj.4816.

**Innis T, Cunning R, Ritson-Williams R, Wall CB, Gates RD. 2018.** Coral color and depth drive symbiosis ecology of *Montipora capitata* in Kāneʻohe Bay, Oʻahu, Hawaiʻi. *Coral Reefs* **37**:423–430 DOI 10.1007/s00338-018-1667-0.

**Ishikura M, Hagiwara K, Takishita K, Haga M, Iwai K, Maruyama T. 2004.** Isolation of new *Symbiodinium* strains from tridacnid giant clam (*Tridacna crocea*) and sea slug (*Pteraeolidia ianthina*) using culture medium containing giant clam tissue homogenate. *Marine Biotechnology* **6**:378–385 DOI 10.1007/s10126-004-1800-7.

**IUCN. 2021.** The IUCN red list of threatened species. *Available at* https://www.iucnredlist.org/resources/tax-sources.

**Jacobs KP, Hunder CL, Forsman ZH, Pollock AL, de Souza MR, Toonen RJ. 2022.** A phylogenomic examination of Palmyra Atoll's corallimorpharian invader. *Coral Reefs* **41**:673–685 DOI 10.1007/s00338-021-02143-5.

**Jeong HJ, Lee SY, Kang NS, Yoo YD, Lim AS, Lee MJ, Kim HS, Yih W, Yamashita H, LaJeunesse TC. 2014.** Genetics and morphology characterize the dinoflagellate *Symbiodinium voratum*, n. sp., (Dinophyceae) as the sole representative of *Symbiodinium* Clade E. *Journal of Eukaryotic Microbiology* **61**:75–94 DOI 10.1111/jeu.12088.

**Jinkerson RE, Russo JA, Newkirk CR, Kirk AL, Chi RJ, Martindale MQ, Grossman AR, Hatta M, Xiang T. 2022.** Cnidarian-Symbiodiniaceae symbiosis establishment is independent of photosynthesis. *Current Biology* **32**:2402–2415 DOI 10.1016/j.cub.2022.04.021.

**Jones AM, Berkelmans R, van Oppen MJH, Mieog JC, Sinclair W. 2008.** A community change in the algal endosymbionts of a scleractinian coral following a natural bleaching event: field evidence of acclimatization. *Proceedings of the Royal Society B: Biological Sciences* **275**:1359–1365 DOI 10.1098/rspb.2008.0069.

**Kareiva P, Levin SA. 2015.** *The importance of species.* Princeton: Princeton University Press.

**Karst SM, Ziels RM, Kirkegaard RH, Sørensen EA, McDonald D, Zhu Q, Knight R, Albertsen M. 2021.** High-accuracy long-read amplicon sequences using unique molecular identifiers with Nanopore or PacBio sequencing. *Nature Methods* **18**:165–169 DOI 10.1038/s41592-020-01041-y.

**Kavousi J, Denis V, Sharp V, Reimer JD, Nakamura T, Parkinson JE. 2020.** Unique combinations of coral host and algal symbiont genotypes reflect intraspecific variation in heat stress responses among colonies of the reef-building coral, *Montipora digitata*. *Marine Biology* **167**:23 DOI 10.1007/s00227-019-3632-z.

**Kemp DW, Fitt WK, Schmidt GW. 2008.** A microsampling method for genotyping coral symbionts. *Coral Reefs* **27**:289–293 DOI 10.1007/s00338-007-0333-8.

**Kiers ET, Denison RF, Kawakita A, Herre EA. 2011.** The biological reality of host sanctions and partner fidelity. *Proceedings of the National Academy of Sciences of the United States of America* **108**:E7 DOI 10.1073/pnas.1014546108.

**Kiers ET, Rousseau RA, West SA, Denison RF. 2003.** Host sanctions and the legume-rhizobium mutualism. *Nature* **425**:78–81 DOI 10.1038/nature01931.

**Kirk NL, Andras JP, Drew Harvell C, Santos SR, Coffroth MA. 2009.** Population structure of *Symbiodinium* sp. associated with the common sea fan, *Gorgonia ventalina*, in the Florida Keys across distance, depth, and time. *Marine Biology* **156**:1609–1623 DOI 10.1007/s00227-009-1196-z.

**Kitchen SA, Von Kuster G, Kuntz KLV, Reich HG, Miller W, Griffin S, Fogarty ND, Baums IB. 2020.** STAGdb: a 30K SNP genotyping array and Science Gateway for *Acropora* corals and their dinoflagellate symbionts. *Scientific Reports* **10**:12488 DOI 10.1038/s41598-020-69101-z.

**Klueter A, Crandall JB, Archer FI, Teece MA, Coffroth MA. 2015.** Taxonomic and environmental variation of metabolite profiles in marine dinoflagellates of the genus *Symbiodinium*. *Metabolites* **5**:74–99 DOI 10.3390/metabo5010074.

**Klueter A, Trapani J, Archer FI, McIlroy SE, Coffroth MA. 2017.** Comparative growth rates of cultured marine dinoflagellates in the genus *Symbiodinium* and the effects of temperature and light. *PLOS ONE* **12**:e0187707 DOI 10.1371/journal.pone.0187707.

**Konopka A. 2009.** What is microbial community ecology? *The ISME Journal* **3**:1223–1230 DOI 10.1038/ismej.2009.88.

**Krassowski M, Das V, Sahu SK, Misra BB. 2020.** State of the field in multi-omics research: from computational needs to data mining and sharing. *Frontiers in Genetics* **11**:610798 DOI 10.3389/fgene.2020.610798.

**Kriefall NG, Kanke MR, Aglyamova GV, Davies SW. 2022.** Reef environments shape microbial partners in a highly connected coral population. *Proceedings of the Royal Society B: Biological Sciences* **289**:20212459 DOI 10.1098/rspb.2021.2459.

**Krueger T, Gates RD. 2012.** Cultivating endosymbionts—host environmental mimics support the survival of *Symbiodinium* C15 *ex hospite*. *Journal of Experimental Marine Biology and Ecology* **413**:169–176 DOI 10.1016/j.jembe.2011.12.002.

**Kuguru B, Chadwick NE, Achituv Y, Zandbank K, Tchernov D. 2008.** Mechanisms of habitat segregation between corallimorpharians: photosynthetic parameters and *Symbiodinium* types. *Marine Ecology Progress Series* **369**:115–129 DOI 10.3354/meps07651.

**Kunihiro S, Reimer JD. 2018.** Phylogenetic analyses of *Symbiodinium* isolated from *Waminoa* and their anthozoan hosts in the Ryukyu Archipelago, southern Japan. *Symbiosis* **76**:253–264 DOI 10.1007/s13199-018-0557-0.

**Ladner JT, Barshis DJ, Palumbi SR. 2012.** Protein evolution in two co-occurring types of *Symbiodinium*: an exploration into the genetic basis of thermal tolerance in *Symbiodinium* Clade D. *BMC Evolutionary Biology* **12**:217 DOI 10.1186/1471-2148-12-217.

**LaJeunesse TC. 2017.** Validation and description of *Symbiodinium microadriaticum*, the type species of *Symbiodinium* (Dinophyta). *Journal of Phycology* **53**:1109–1114 DOI 10.1111/jpy.12570.

**LaJeunesse TC. 2001.** Investigating the biodiversity, ecology, and phylogeny of endosymbiotic dinoflagellates in the genus *Symbiodinium* using the ITS region: in search of a "species" level marker. *Journal of Phycology* **37**:866–880 DOI 10.1046/j.1529-8817.2001.01031.x.

**LaJeunesse TC, Casado-Amezúa P, Hume BCC, Butler CC, Mordret S, Piredda R, De Luca P, Pannone R, Sarno D, Wiedenmann J, D'Ambra I. 2022.** Mutualistic dinoflagellates with big disparities in ribosomal DNA variation may confound estimates of symbiont diversity and ecology in the jellyfish Cotylorhiza tuberculata. *Symbiosis* **88**:1–10 DOI 10.1007/s13199-022-00880-x.

**LaJeunesse TC, Lee SY, Gil-Agudelo DL. 2015.** *Symbiodinium necroappetens* sp. nov. (Dinophyceae): an opportunist "zooxanthella" found in bleached and diseased tissues of

Caribbean reef corals. *European Journal of Phycology* **50(2)**:223–238 DOI 10.1080/09670262.2015.1025857.

**LaJeunesse TC, Loh W, Trench RK. 2009.** Do introduced endosymbiotic dinoflagellates "take" to new hosts? *Biological Invasions* **11**:995–1003 DOI 10.1007/s10530-008-9311-5.

**LaJeunesse TC, Parkinson JE, Gabrielson PW, Jeong HJ, Reimer JD, Voolstra CR, Santos SR. 2018.** Systematic revision of Symbiodiniaceae highlights the antiquity and diversity of coral endosymbionts. *Current Biology* **28**:2570–2580 DOI 10.1016/j.cub.2018.07.008.

**LaJeunesse TC, Parkinson JE, Reimer JD. 2012.** A genetics-based description of *Symbiodinium minutum* sp. nov. and *S. psygmophilum* sp. nov. (Dinophyceae), two dinoflagellates symbiotic with cnidaria. *Journal of Phycology* **48**:1380–1391 DOI 10.1111/j.1529-8817.2012.01217.x.

**LaJeunesse TC, Pettay DT, Sampayo EM, Phongsuwan N, Brown B, Obura DO, Hoegh-Guldberg O, Fitt WK. 2010a.** Long-standing environmental conditions, geographic isolation and host-symbiont specificity influence the relative ecological dominance and genetic diversification of coral endosymbionts in the genus *Symbiodinium*. *Journal of Biogeography* **37**:785–800 DOI 10.1111/j.1365-2699.2010.02273.x.

**LaJeunesse TC, Smith RT, Walther M, Pinzón JH, Pettay DT, McGinley MP, Aschaffenburg MD, Medina-Rosas P, Cupul-Magaña AL, Pérez AL, Reyes-Bonilla H, Warner ME. 2010b.** Host-symbiont recombination versus natural selection in the response of coral-dinoflagellate symbioses to environmental disturbance. *Proceedings of the Royal Society B: Biological Sciences* **277**:2925–2934 DOI 10.1098/rspb.2010.0385.

**LaJeunesse TC, Thornhill DJ. 2011.** Improved resolution of reef-coral endosymbiont (*Symbiodinium*) species diversity, ecology, and evolution through *psbA* non-coding region genotyping. *PLOS ONE* **6**:e29013 DOI 10.1371/journal.pone.0029013.

**LaJeunesse TC, Trench RK. 2000.** Biogeography of two species of *Symbiodinium* (Freudenthal) inhabiting the intertidal sea anemone *Anthopleura elegantissima* (Brandt). *Biological Bulletin* **199**:126–134 DOI 10.2307/1542872.

**LaJeunesse TC, Wham DC, Pettay DT, Parkinson JE. 2014.** Ecologically differentiated stress-tolerant endosymbionts in the dinoflagellate genus *Symbiodinium* (Dinophyceae) Clade D are different species. *Phycologia* **53**:305–319 DOI 10.2216/13-186.1.

**LaJeunesse TC, Wiedenmann J, Casado-Amezúa P, D'Ambra I, Turnham KE, Nitschke MR, Oakley CA, Goffredo S, Spano CA, Cubillos VM, Davy SK, Suggett DJ. 2021.** Revival of *Philozoon* Geddes for host-specialized dinoflagellates, "zooxanthellae", in animals from coastal temperate zones of northern and southern hemispheres. *European Journal of Phycology* **57**:166–180 DOI 10.1080/09670262.2021.1914863.

**Langfelder P, Horvath S. 2008.** *WGCNA*: an R package for weighted correlation network analysis. *BMC Bioinformatics* **9**:559 DOI 10.1186/1471-2105-9-559.

**Lawson CA, Possell M, Seymour JR, Raina JB, Suggett DJ. 2019.** Coral endosymbionts (Symbiodiniaceae) emit species-specific volatilomes that shift when exposed to thermal stress. *Scientific Reports* **9**:17395 DOI 10.1038/s41598-019-53552-0.

**Lawson CA, Raina J-B, Kahlke T, Seymour JR, Suggett DJ. 2018.** Defining the core microbiome of the symbiotic dinoflagellate, *Symbiodinium*. *Environmental Microbiology Reports* **10**:7–11 DOI 10.1111/1758-2229.12599.

**Lee MJ, Jeong HJ, Jang SH, Lee SY, Kang NS, Lee KH, Kim HS, Wham DC, LaJeunesse TC. 2016.** Most low-abundance "background" *Symbiodinium* spp. are transitory and have minimal functional significance for symbiotic corals. *Microbial Ecology* **71**:771–783 DOI 10.1007/s00248-015-0724-2.

Lee SY, Jeong HJ, Kang NS, Jang TY. 2015. *Symbiodinium tridacnidorum* sp. nov., a dinoflagellate common to Indo-Pacific giant clams, and a revised morphological description of *Symbiodinium microadriaticum*. *European Journal of Phycology* **50**:155–172 DOI 10.1080/09670262.2015.1018336.

Lee SY, Jeong HJ, LaJeunesse TC. 2020. *Cladocopium infistulum* sp. nov. (Dinophyceae), a thermally tolerant dinoflagellate symbiotic with giant clams from the western Pacific Ocean. *Phycologia* **59**:515–526 DOI 10.1080/00318884.2020.1807741.

Leliaert F, Verbruggen H, Vanormelingen P, Steen F, López-Bautista JM, Zuccarello GC, De Clerck O. 2014. DNA-based species delimitation in algae. *European Journal of Phycology* **49**:179–196 DOI 10.1080/09670262.2014.904524.

Levin RA, Voolstra CR, Weynberg KD, van Oppen MJH. 2017. Evidence for a role of viruses in the thermal sensitivity of coral photosymbionts. *The ISME Journal* **11**:808–812 DOI 10.1038/ismej.2016.154.

Levy S, Elek A, Grau-Bové X, Menéndez-Bravo S, Iglesias M, Tanay A, Mass T, Sebé-Pedrós A. 2021. A stony coral cell atlas illuminates the molecular and cellular basis of coral symbiosis, calcification, and immunity. *Cell* **184**:2973–2987.e18 DOI 10.1016/j.cell.2021.04.005.

Lewis AM, Chan AN, LaJeunesse TC. 2019. New species of closely related endosymbiotic dinoflagellates in the greater Caribbean have niches corresponding to host coral phylogeny. *Journal of Eukaryotic Microbiology* **66**:469–482 DOI 10.1111/jeu.12692.

Lewis C, Neely K, Rodriguez-Lanetty M. 2019. Recurring episodes of thermal stress shift the balance from a dominant host-specialist to a background host-generalist zooxanthella in the threatened pillar coral, *Dendrogyra cylindrus*. *Frontiers in Marine Science* **6**:5 DOI 10.3389/fmars.2019.00005.

Li J, Zou Y, Yang J, Li Q, Bourne DG, Sweet M, Liu C, Guo A, Zhang S. 2022. Cultured bacteria provide insight into the functional potential of the coral-associated microbiome. *mSystems* **7(4)**:e0032722 DOI 10.1128/msystems.00327-22.

Li T, Yu L, Song B, Song Y, Li L, Lin X, Lin S. 2020. Genome improvement and core gene set refinement of *Fugacium kawagutii*. *Microorganisms* **8**:102 DOI 10.3390/microorganisms8010102.

Liang J, Luo W, Yu K, Xu Y, Chen J, Deng C, Ge R, Su H, Huang W, Wang G. 2021. Multi-omics revealing the response patterns of symbiotic microorganisms and host metabolism in scleractinian coral *Pavona minuta* to temperature stresses. *Metabolites* **12**:18 DOI 10.3390/metabo12010018.

Lien Y-T, Fukami H, Yamashita Y. 2012. *Symbiodinium* Clade C dominates zooxanthellate corals (Scleractinia) in the temperate region of Japan. *Zoological Science* **29**:173–180 DOI 10.2108/zsj.29.173.

Lien Y-T, Nakano Y, Plathong S, Fukami H, Wang J-T, Chen CA. 2007. Occurrence of the putatively heat-tolerant *Symbiodinium* phylotype D in high-latitudinal outlying coral communities. *Coral Reefs* **26**:35–44 DOI 10.1007/s00338-006-0185-7.

Liew YJ, Li Y, Baumgarten S, Voolstra CR, Aranda M. 2017. Condition-specific RNA editing in the coral symbiont *Symbiodinium microadriaticum*. *PLOS Genetics* **13**:e1006619 DOI 10.1371/journal.pgen.1006619.

Liew YJ, Zoccola D, Li Y, Tambutté E, Venn AA, Michell CT, Cui G, Deutekom ES, Kaandorp JA, Voolstra CR, Forêt S, Allemand D, Tambutté S, Aranda M. 2018. Epigenome-associated phenotypic acclimatization to ocean acidification in a reef-building coral. *Science Advances* **4**:eaar8028 DOI 10.1126/sciadv.aar8028.

**Lim SSQ, Huang D, Soong K, Neo ML. 2019.** Diversity of endosymbiotic Symbiodiniaceae in giant clams at Dongsha Atoll, northern South China Sea. *Symbiosis* **78**:251–262 DOI 10.1007/s13199-019-00615-5.

**Lin S. 2011.** Genomic understanding of dinoflagellates. *Research in Microbiology* **162**:551–569 DOI 10.1016/j.resmic.2011.04.006.

**Lin S, Cheng S, Song B, Zhong X, Lin X, Li W, Li L, Zhang Y, Zhang H, Ji Z, Cai M, Zhuang Y, Shi X, Lin L, Wang L, Wang Z, Liu X, Yu S, Zeng P, Hao H, Zou Q, Chen C, Li Y, Wang Y, Xu C, Meng S, Xu X, Wang J, Yang H, Campbell DA, Sturm NR, Dagenais-Bellefeuille S, Morse D. 2015.** The *Symbiodinium kawagutii* genome illuminates dinoflagellate gene expression and coral symbiosis. *Science* **350**:691–694 DOI 10.1126/science.aad0408.

**Lin S, Song B, Morse D. 2021.** Spatial organization of dinoflagellate genomes: novel insights and remaining critical questions. *Journal of Phycology* **57**:1674–1678 DOI 10.1111/jpy.13206.

**Liu B, Hu Y, Hu Z, Liu G, Zhu H. 2020.** Taxonomic scheme of the order Chaetophorales (Chlorophyceae, Chlorophyta) based on chloroplast genomes. *BMC Genomics* **21**:442 DOI 10.1186/s12864-020-06845-y.

**Liu H, Stephens TG, González-Pech RA, Beltran VH, Lapeyre B, Bongaerts P, Cooke I, Aranda M, Bourne DG, Forêt S, Miller DJ, van Oppen MJH, Voolstra CR, Ragan MA, Chan CX. 2018.** *Symbiodinium* genomes reveal adaptive evolution of functions related to coral-dinoflagellate symbiosis. *Communications Biology* **1**:95 DOI 10.1038/s42003-018-0098-3.

**Lohuis MR, Miller DJ. 1998.** Hypermethylation at CpG-motifs in the dinoflagellates *Amphidinium carterae* (Dinophyceae) and *Symbiodinium microadriaticum* (Dinophyceae): evidence from restriction analyses, 5-azacytidine and ethionine treatment. *Journal of Phycology* **34**:152–159 DOI 10.1046/j.1529-8817.1998.340152.x.

**Loram JE, Boonham N, O'Toole P, Trapido-Rosenthal HG, Douglas AE. 2007.** Molecular quantification of symbiotic dinoflagellate algae of the genus *Symbiodinium*. *Biological Bulletin* **212**:259–268 DOI 10.2307/25066608.

**Lo R, Dougan KE, Chen Y, Shah S, Bhattacharya D, Chan CX. 2022.** Alignment-free analysis of Symbiodiniaceae whole-genome sequences reveals differential phylogenetic signals in distinct regions. *Frontiers in Plant Science* **1233**:815714 DOI 10.3389/fpls.2022.815714.

**Lowndes JSS, Best BD, Scarborough C, Afflerbach JC, Frazier MR, O'Hara CC, Jiang N, Halpern BS. 2017.** Our path to better science in less time using open data science tools. *Nature Ecology & Evolution* **1**:160 DOI 10.1038/s41559-017-0160.

**Lundberg E, Borner GH. 2019.** Spatial proteomics: a powerful discovery tool for cell biology. *Nature Reviews Molecular Cell Biology* **20(5)**:285–302 DOI 10.1038/s41580-018-0094-y.

**Madin JS, Anderson KD, Andreasen MH, Bridge TCL, Cairns SD, Connolly SR, Darling ES, Diaz M, Falster DS, Franklin EC, Gates RD, Harmer A, Hoogenboom MO, Huang D, Keith SA, Kosnik MA, Kuo CY, Lough JM, Lovelock CE, Luiz O, Martinelli J, Mizerek T, Pandolfi JM, Pochon X, Pratchett MS, Putnam HM, Roberts TE, Stat M, Wallace CC, Widman E, Baird AH. 2016.** The Coral Trait Database, a curated database of trait information for coral species from the global oceans. *Scientific Data* **3**:160017 DOI 10.1038/sdata.2016.17.

**Magalon H, Baudry E, Husté A, Adjeroud M, Veuille M. 2006.** High genetic diversity of the symbiotic dinoflagellates in the coral *Pocillopora meandrina* from the South Pacific. *Marine Biology* **148**:913–922 DOI 10.1007/s00227-005-0133-z.

**Maire J, Girvan SK, Barkla SE, Perez-Gonzalez A, Suggett DJ, Blackall LL, van Oppen MJH. 2021.** Intracellular bacteria are common and taxonomically diverse in cultured and *in hospite* algal endosymbionts of coral reefs. *The ISME Journal* **15**:2028–2042 DOI 10.1038/s41396-021-00902-4.

Manning MM, Gates RD. 2008. Diversity in populations of free-living *Symbiodinium* from a Caribbean and Pacific reef. *Limnology and Oceanography* 53:1853–1861 DOI 10.4319/lo.2008.53.5.1853.

Mansour JS, Pollock FJ, Díaz-Almeyda E, Iglesias-Prieto R, Medina M. 2018. Intra- and interspecific variation and phenotypic plasticity in thylakoid membrane properties across two *Symbiodinium* Clades. *Coral Reefs* 37:841–850 DOI 10.1007/s00338-018-1710-1.

Manzello DP, Matz MV, Enochs IC, Valentino L, Carlton RD, Kolodziej G, Serrano X, Towle EK, Jankulak M. 2018. Role of host genetics and heat-tolerant algal symbionts in sustaining populations of the endangered coral *Orbicella faveolata* in the Florida Keys with ocean warming. *Global Change Biology* 25:1016–1031 DOI 10.1111/gcb.14545.

Marcelino LA, Westneat MW, Stoyneva V, Henss J, Rogers JD, Radosevich A, Turzhitsky V, Siple M, Fang A, Swain TD, Fung J, Backman V. 2013. Modulation of light-enhancement to symbiotic algae by light-scattering in corals and evolutionary trends in bleaching. *PLOS ONE* 8:e61492 DOI 10.1371/journal.pone.0061492.

Marhoefer SR, Zenger KR, Strugnell JM, Logan M, van Oppen MJH, Kenkel CD, Bay LK. 2021. Signatures of adaptation and acclimatization to reef flat and slope habitats in the coral *Pocillopora damicornis*. *Frontiers in Marine Science* 8:704709 DOI 10.3389/fmars.2021.704709.

Marinov GK, Trevino AE, Xiang T, Kundaje A, Grossman AR, Greenleaf WJ. 2021. Transcription-dependent domain-scale three-dimensional genome organization in the dinoflagellate *Breviolum minutum*. *Nature Genetics* 53(5):1–5 DOI 10.1038/s41588-021-00848-5.

Maruyama S, Weis VM. 2021. Limitations of using cultured algae to study cnidarian-algal symbioses and suggestions for future studies. *Journal of Phycology* 57:30–38 DOI 10.1111/jpy.13102.

Matthews JL, Crowder CM, Oakley CA, Lutz A, Roessner U, Meyer E, Grossman AR, Weis VM, Davy SK. 2017. Optimal nutrient exchange and immune responses operate in partner specificity in the cnidarian-dinoflagellate symbiosis. *Proceedings of the National Academy of Sciences of the United States of America* 114:13194–13199 DOI 10.1073/pnas.1710733114.

Matthews JL, Raina JB, Kahlke T, Seymour JR, van Oppen MJH, Suggett DJ. 2020. Symbiodiniaceae-bacteria interactions: rethinking metabolite exchange in reef-building corals as multi-partner metabolic networks. *Environmental Microbiology* 22:1675–1687 DOI 10.1111/1462-2920.14918.

Mayr E. 1942. *Systematics and the origin of species*. New York: Columbia University Press.

McIlroy SE, Cunning R, Baker AC, Coffroth MA. 2019. Competition and succession among coral endosymbionts. *Ecology and Evolution* 9:12767–12778 DOI 10.1002/ece3.5749.

McIlroy SE, Smith GJ, Geller JB. 2014. FISH-Flow: a quantitative molecular approach for describing mixed clade communities of *Symbiodinium*. *Coral Reefs* 33:157–167 DOI 10.1007/s00338-013-1087-0.

McIlroy SE, Wong JCY, Baker DM. 2020. Competitive traits of coral symbionts may alter the structure and function of the microbiome. *The ISME Journal* 14:2424–2432 DOI 10.1038/s41396-020-0697-0.

McKenna V, Archibald JM, Beinart R, Dawson MN, Hentschel U, Keeling PJ, Lopez JV, Martín-Durán JM, Petersen JM, Sigwart JD, Simakov O, Sutherland KR, Sweet M, Talbot N, Thompson AW, Bender S, Harrison PW, Rajan J, Cochrane G, Berriman M, Lawniczak M, Blaxter M. 2021. The aquatic symbiosis genomics project: probing the evolution of symbiosis across the tree of life. *Wellcome Open Research* 6:254 DOI 10.12688/wellcomeopenres.17222.1.

Méndez C, Ahlenstiel CL, Kelleher AD. 2015. Post-transcriptional gene silencing, transcriptional gene silencing and human immunodeficiency virus. *World Journal of Virology* **4**:219–244 DOI 10.5501/wjv.v4.i3.219.

Mieog JC, van Oppen MJH, Cantin NE, Stam WT, Olsen JL. 2007. Real-time PCR reveals a high incidence of *Symbiodinium* clade D at low levels in four scleractinian corals across the Great Barrier Reef: implications for symbiont shuffling. *Coral Reefs* **26**:449–457 DOI 10.1007/s00338-007-0244-8.

Mizuyama M, Iguchi A, Iijima M, Gibu K, Reimer JD. 2020. Comparison of Symbiodiniaceae diversities in different members of a *Palythoa* species complex (Cnidaria: Anthozoa: Zoantharia)-implications for ecological adaptations to different microhabitats. *PeerJ* **8**:e8449 DOI 10.7717/peerj.8449.

Molecular Ecology Resources Primer Development Consortium, Abdoullaye D, Acevedo I, Adebayo AA, Behrmann-Godel J, Benjamin RC, Bock DG, Born C, Brouat C, Caccone A, Cao LZ, Casado-Amezúa P, Catanéo J, Correa-Ramirez MM, Cristescu ME, Dobigny G, Egbosimba EE, Etchberger LK, Fan B, Fields PD, Forcioli D, Furla P, Garcia de Leon FJ, García-Jiménez R, Gauthier P, Gergs R, González C, Granjon L, Gutiérrez-Rodríguez C, Havill NP, Helsen P, Hether TD, Hoffman EA, Hu X, Ingvarsson PK, Ishizaki S, Ji H, Ji XS, Jimenez ML, Kapil R, Karban R, Keller SR, Kubota S, Li S, Li W, Lim DD, Lin H, Liu X, Luo Y, Machordom A, Martin AP, Matthysen E, Mazzella MN, McGeoch MA, Meng Z, Nishizawa M, O'Brien P, Ohara M, Ornelas JF, Ortu MF, Pedersen AB, Preston L, Ren Q, Rothhaupt KO, Sackett LC, Sang Q, Sawyer GM, Shiojiri K, Taylor DR, Van Dongen S, Van Vuuren BJ, Vandewoestijne S, Wang H, Wang JT, Wang LE, Xu XL, Yang G, Yang Y, Zeng YQ, Zhang QW, Zhang Y, Zhao Y, Zhou Y. 2010. Permanent genetic resources added to Molecular Ecology Resources Database 1 August 2009-30 September 2009. *Molecular Ecology Resources* **10**:232–236 DOI 10.1111/j.1755-0998.2009.02796.x.

Montgomery BL. 2017a. From deficits to possibilities: mentoring lessons from plants on cultivating individual growth through environmental assessment and optimization. *Manifold* **1**:1–11 DOI 10.25335/M5/PPJ.1.1-3.

Montgomery BL. 2017b. Mapping a mentoring roadmap and developing a supportive network for strategic career advancement. *SAGE Open* **7**:2158244017710288 DOI 10.1177/2158244017710288.

Montgomery BL, Dodson JE, Johnson SM. 2014. Guiding the way: mentoring graduate students and junior faculty for sustainable academic careers. *SAGE Open* **4**:2158244014558043 DOI 10.1177/2158244014558043.

Moore RB, Ferguson KM, Loh WKW, Hoegh-Guldberg O, Carter DA. 2003. Highly organized structure in the non-coding region of the *psbA* minicircle from Clade C *Symbiodinium*. *International Journal of Systematic and Evolutionary Microbiology* **53**:1725–1734 DOI 10.1099/ijs.0.02594-0.

Mordret S, Romac S, Henry N, Colin S, Carmichael M, Berney C, Audic S, Richter DJ, Pochon X, de Vargas C, Decelle J. 2016. The symbiotic life of *Symbiodinium* in the open ocean within a new species of calcifying ciliate (*Tiarina* sp.). *The ISME Journal* **10**:1424–1436 DOI 10.1038/ismej.2015.211.

Mungpakdee S, Shinzato C, Takeuchi T, Kawashima T, Koyanagi R, Hisata K, Tanaka M, Goto H, Fujie M, Lin S, Satoh N, Shoguchi E. 2014. Massive gene transfer and extensive RNA editing of a symbiotic dinoflagellate plastid genome. *Genome Biology and Evolution* **6**:1408–1422 DOI 10.1093/gbe/evu109.

**Murugesan RK, Balakrishan R, Natesan S, Jayavel S, Muthiah RC. 2022.** Identification of coral endosymbionts of Veedhalai and Mandapam coasts of Palk Bay, India using small subunit rDNA. *Bioinformatician* **18(4)**:318–324 DOI 10.6026/97320630018318.

**Muscatine L, Ferrier-Pagès C, Blackburn A, Gates RD, Baghdasarian G, Allemand D. 1998.** Cell-specific density of symbiotic dinoflagellates in tropical anthozoans. *Coral Reefs* **17**:329–337 DOI 10.1007/s003380050133.

**Naepi S, McAllister TG, Thomsen P, Leenen-Young M, Walker LA, McAllister AL, Theodore R, Kidman J, Suaaliia T. 2020.** The pakaru "pipeline": Māori and Pasifika pathways within the academy. *The New Zealand Annual Review of Education* **24**:142 DOI 10.26686/nzaroe.v24i0.6338.

**Nand A, Zhan Y, Salazar OR, Aranda M, Voolstra CR, Dekker J. 2021.** Genetic and spatial organization of the unusual chromosomes of the dinoflagellate *Symbiodinium microadriaticum*. *Nature Genetics* **53**:618–629 DOI 10.1038/s41588-021-00841-y.

**Nielsen MW, Bloch CW, Schiebinger L. 2018.** Making gender diversity work for scientific discovery and innovation. *Nature Human Behaviour* **2**:726–734 DOI 10.1038/s41562-018-0433-1.

**Nitschke MR, Craveiro SC, Brandão C, Fidalgo C, Serôdio J, Calado AJ, Frommlet JC. 2020.** Description of *Freudenthalidium* gen. nov. and *Halluxium* gen. nov. to formally recognize Clades Fr3 and H as genera in the family Symbiodiniaceae (Dinophyceae). *Journal of Phycology* **56**:923–940 DOI 10.1111/jpy.12999.

**Nitschke MR, Davy SK, Ward S. 2016.** Horizontal transmission of *Symbiodinium* cells between adult and juvenile corals is aided by benthic sediment. *Coral Reefs* **35**:335–344 DOI 10.1007/s00338-015-1349-0.

**Noda H, Parkinson JE, Yang SY, Reimer JD. 2017.** A preliminary survey of zoantharian endosymbionts shows high genetic variation over small geographic scales on Okinawa-jima Island, Japan. *PeerJ* **5**:e3740 DOI 10.7717/peerj.3740.

**O'Brien LT, Bart HL, Garcia DM. 2020.** Why are there so few ethnic minorities in ecology and evolutionary biology? Challenges to inclusion and the role of sense of belonging. *Social Psychology of Education* **23**:449–477 DOI 10.1007/s11218-019-09538-x.

**Olander A, Lawson CA, Possell M, Raina JB, Ueland M, Suggett DJ. 2021.** Comparative volatilomics of coral endosymbionts from one- and comprehensive two-dimensional gas chromatography approaches. *Marine Biology* **168**:76 DOI 10.1007/s00227-021-03859-2.

**Palmer TM, Stanton ML, Young TP. 2003.** Competition and coexistence: Exploring mechanisms that restrict and maintain diversity within mutualist guilds. *American Naturalist* **162**:S63–S79 DOI 10.1086/378682.

**Palser ER, Lazerwitz M, Fotopoulou A. 2022.** Gender and geographical disparity in editorial boards of journals in psychology and neuroscience. *Nature Neuroscience* **25**:272–279 DOI 10.1038/s41593-022-01012-w.

**Parker GM. 1984.** Dispersal of zooxanthellae on coral reefs by predators on cnidarians. *Biological Bulletin* **167**:159–167 DOI 10.2307/1541344.

**Parkinson JE, Baumgarten S, Michell CT, Baums IB, LaJeunesse TC, Voolstra CR. 2016.** Gene expression variation resolves species and individual strains among coral-associated dinoflagellates within the genus *Symbiodinium*. *Genome Biology and Evolution* **8**:665–680 DOI 10.1093/gbe/evw019.

**Parkinson JE, Baums IB. 2014.** The extended phenotypes of marine symbioses: ecological and evolutionary consequences of intraspecific genetic diversity in coral-algal associations. *Frontiers in Microbiology* **5**:445 DOI 10.3389/fmicb.2014.00445.

**Parkinson JE, Coffroth MA, LaJeunesse TC. 2015.** New species of Clade B *Symbiodinium* (Dinophyceae) from the greater Caribbean belong to different functional guilds: *S. aenigmaticum* sp. nov., *S. antillogorgium* sp. nov., *S. endomadracis* sp. nov., and *S. pseudominutum* sp. nov. *Journal of Phycology* **51(5)**:850–858 DOI 10.1111/jpy.12340.

**Pettay DT, LaJeunesse TC. 2009.** Microsatellite loci for assessing genetic diversity, dispersal and clonality of coral symbionts in "stress-tolerant" Clade D *Symbiodinium*. *Molecular Ecology Resources* **9**:1022–1025 DOI 10.1111/j.1755-0998.2009.02561.x.

**Pettay DT, LaJeunesse TC. 2007.** Microsatellites from Clade B *Symbiodinium* spp. specialized for Caribbean corals in the genus Madracis. *Molecular Ecology Notes* **7**:1271–1274 DOI 10.1111/j.1471-8286.2007.01852.x.

**Pettay DT, Wham DC, Smith RT, Iglesias-Prieto R, LaJeunesse TC. 2015.** Microbial invasion of the Caribbean by an Indo-Pacific coral zooxanthella. *Proceedings of the National Academy of Sciences of the United States of America* **112**:7513–7518 DOI 10.1073/pnas.1502283112.

**Pinho-Gomes AC, Vassallo A, Thompson K, Womersley K, Norton R, Woodward M. 2021.** Representation of women among editors in chief of leading medical journals. *JAMA Network Open* **4**:e2123026 DOI 10.1001/jamanetworkopen.2021.23026.

**Pinzón JH, Devlin-Durante MK, Weber MX, Baums IB, LaJeunesse TC. 2011.** Microsatellite loci for *Symbiodinium* A3 (*S. fitti*) a common algal symbiont among Caribbean *Acropora* (stony corals) and Indo-Pacific giant clams (*Tridacna*). *Conservation Genetics Resources* **3**:45–47 DOI 10.1007/s12686-010-9283-5.

**Pochon X, Garcia-Cuetos L, Baker AC, Castella E, Pawlowski J. 2007.** One-year survey of a single Micronesian reef reveals extraordinarily rich diversity of *Symbiodinium* types in soritid foraminifera. *Coral Reefs* **26**:867–882 DOI 10.1007/s00338-007-0279-x.

**Pochon X, LaJeunesse TC. 2021.** *Miliolidium* n. gen, a new Symbiodiniacean genus whose members associate with soritid foraminifera or are free-living. *Journal of Eukaryotic Microbiology* **68**:e12856 DOI 10.1111/jeu.12856.

**Pochon X, Montoya-Burgos JI, Stadelmann B, Pawlowski J. 2006.** Molecular phylogeny, evolutionary rates, and divergence timing of the symbiotic dinoflagellate genus *Symbiodinium*. *Molecular Phylogenetics and Evolution* **38**:20–30 DOI 10.1016/j.ympev.2005.04.028.

**Pochon X, Putnam HM, Burki F, Gates RD. 2012.** Identifying and characterizing alternative molecular markers for the symbiotic and free-living dinoflagellate genus *Symbiodinium*. *PLOS ONE* **7**:e29816 DOI 10.1371/journal.pone.0029816.

**Pochon X, Putnam HM, Gates RD. 2014.** Multi-gene analysis of *Symbiodinium* dinoflagellates: a perspective on rarity, symbiosis, and evolution. *PeerJ* **2**:e394 DOI 10.7717/peerj.394.

**Pochon X, Stat M, Takabayashi M, Chasqui L, Chauka LJ, Logan DDK, Gates RD. 2010.** Comparison of endosymbiotic and free-living *Symbiodinium* (Dinophyceae) diversity in a Hawaiian reef environment. *Journal of Phycology* **46**:53–65 DOI 10.1111/j.1529-8817.2009.00797.x.

**Pochon X, Wecker P, Stat M, Berteaux-Lecellier V, Lecellier G. 2019.** Towards an in-depth characterization of Symbiodiniaceae in tropical giant clams via metabarcoding of pooled multi-gene amplicons. *PeerJ* **7**:e6898 DOI 10.7717/peerj.6898.

**Porto I, Granados C, Restrepo JC, Sánchez JA. 2008.** Macroalgal-associated dinoflagellates belonging to the genus *Symbiodinium* in Caribbean reefs. *PLOS ONE* **3**:e2160 DOI 10.1371/journal.pone.0002160.

**Prada C, McIlroy SE, Beltrán DM, Valint DJ, Ford SA, Hellberg ME, Coffroth MA. 2014.** Cryptic diversity hides host and habitat specialization in a gorgonian-algal symbiosis. *Molecular Ecology* **23**:3330–3340 DOI 10.1111/mec.12808.

**Putman AI, Carbone I. 2014.** Challenges in analysis and interpretation of microsatellite data for population genetic studies. *Ecology and Evolution* **4**:4399–4428 DOI 10.1002/ece3.1305.

**Putnam HM, Davidson JM, Gates RD. 2016.** Ocean acidification influences host DNA methylation and phenotypic plasticity in environmentally susceptible corals. *Evolutionary Applications* **9**:1165–1178 DOI 10.1111/eva.12408.

**Putnam HM, Gates RD. 2015.** Preconditioning in the reef-building coral *Pocillopora damicornis* and the potential for trans-generational acclimatization in coral larvae under future climate change conditions. *Journal of Experimental Biology* **218**:2365–2372 DOI 10.1242/jeb.123018.

**Quattrini AM, Rodríguez E, Faircloth BC, Cowman PF, Brugler MR, Farfan GA, Hellberg ME, Kitahara MV, Morrison CL, Paz-García DA, Reimer JD, McFadden CS. 2020.** Palaeoclimate ocean conditions shaped the evolution of corals and their skeletons through deep time. *Nature Ecology & Evolution* **4**:1531–1538 DOI 10.1038/s41559-020-01291-1.

**Quigley KM, Baker AC, Coffroth MA, Willis BL. 2018.** Bleaching resistance and the role of algal endosymbionts. In: van Oppen MJH, Lough JM, eds. *Coral Bleaching: Patterns, Processes, Causes, and Consequences*. Heidelberg, Berlin: Springer, 111–151.

**Quigley KM, Bay LK, Willis BL. 2017.** Temperature and water quality-related patterns in sediment-associated *Symbiodinium* communities impact symbiont uptake and fitness of juveniles in the genus *Acropora*. *Frontiers in Marine Science* **4**:401 DOI 10.3389/fmars.2017.00401.

**Quigley KM, Davies SW, Kenkel CD, Willis BL, Matz MV, Bay LK. 2014.** Deep-sequencing method for quantifying background abundances of *Symbiodinium* types: exploring the rare *Symbiodinium* biosphere in reef-building corals. *PLOS ONE* **9**:e94297 DOI 10.1371/journal.pone.0094297.

**Quigley KM, Willis BL, Bay LK. 2017.** Heritability of the *Symbiodinium* community in vertically- and horizontally-transmitting broadcast spawning corals. *Scientific Reports* **7**:8219 DOI 10.1038/s41598-017-08179-4.

**Quigley KM, Willis BL, Bay LK. 2016.** Maternal effects and *Symbiodinium* community composition drive differential patterns in juvenile survival in the coral Acropora tenuis. *Royal Society Open Science* **3**:160471 DOI 10.1098/rsos.160471.

**Ramírez-Castañeda V. 2020.** Disadvantages in preparing and publishing scientific papers caused by the dominance of the English language in science: the case of Colombian researchers in biological sciences. *PLOS ONE* **15**:e0238372 DOI 10.1371/journal.pone.0238372.

**Ramsby BD, Hill MS, Thornhill DJ, Steenhuizen SF, Achlatis M, Lewis AM, LaJeunesse TC. 2017.** Sibling species of mutualistic *Symbiodinium* Clade G from bioeroding sponges in the western Pacific and western Atlantic oceans. *Journal of Phycology* **53**:951–960 DOI 10.1111/jpy.12576.

**Ramsby BD, Shirur KP, Iglesias-Prieto R, Goulet TL. 2014.** *Symbiodinium* photosynthesis in Caribbean octocorals. *PLOS ONE* **9**:e106419 DOI 10.1371/journal.pone.0106419.

**Reich HG, Kitchen SA, Stankiewicz KH, Devlin-Durante M, Fogarty ND, Baums IB. 2021.** Genomic variation of an endosymbiotic dinoflagellate (*Symbiodinium "fitti"*) among closely related coral hosts. *Molecular Ecology* **30**:3500–3514 DOI 10.1111/mec.15952.

**Reich HG, Rodriguez IB, LaJeunesse TC, Ho TY. 2020.** Endosymbiotic dinoflagellates pump iron: differences in iron and other trace metal needs among the Symbiodiniaceae. *Coral Reefs* **39**:915–927 DOI 10.1007/s00338-020-01911-z.

**Reimer JD, Herrera M, Gatins R, Roberts MB, Parkinson JE, Berumen ML. 2017.** Latitudinal variation in the symbiotic dinoflagellate *Symbiodinium* of the common reef zoantharian

*Palythoa tuberculosa* on the Saudi Arabian coast of the Red Sea. *Journal of Biogeography* **44**:661–673 DOI 10.1111/jbi.12795.

**Reimer JD, Shah MMR, Sinniger F, Yanagi K, Suda S. 2010.** Preliminary analyses of cultured *Symbiodinium* isolated from sand in the oceanic Ogasawara Islands, Japan. *Marine Biodiversity* **40**:237–247 DOI 10.1007/s12526-010-0044-1.

**Richards AL, Merrill AE, Coon JJ. 2015.** Proteome sequencing goes deep. *Current Opinion in Chemical Biology* **24**:11–17 DOI 10.1016/j.cbpa.2014.10.017.

**Roach TNF, Dilworth J, Christian Martin H, Jones AD, Quinn RA, Drury C. 2021.** Metabolomic signatures of coral bleaching history. *Nature Ecology & Evolution* **5**:495–503 DOI 10.1038/s41559-020-01388-7.

**Robison JD, Warner ME. 2006.** Differential impacts of photoacclimation and thermal stress on the photobiology of four different phylotypes of *Symbiodinium* (Pyrrhophyta). *Journal of Phycology* **42**:568–579 DOI 10.1111/j.1529-8817.2006.00232.x.

**Rodriguez-Casariego JA, Cunning R, Baker AC, Eirin-Lopez JM. 2021.** Symbiont shuffling induces differential DNA methylation responses to thermal stress in the coral *Montastraea cavernosa. Molecular Ecology* **31**:588–602 DOI 10.1111/mec.16246.

**Rodriguez-Casariego JA, Ladd MC, Shantz AA, Lopes C, Cheema MS, Kim B, Roberts SB, Fourqurean JW, Ausio J, Burkepile DE, Eirin-Lopez JM. 2018.** Coral epigenetic responses to nutrient stress: Histone H2A.X phosphorylation dynamics and DNA methylation in the staghorn coral *Acropora cervicornis. Ecology and Evolution* **48**:224 DOI 10.1002/ece3.4678.

**Rodríguez-Casariego JA, Mercado-Molina AE, Garcia-Souto D, Ortiz-Rivera IM, Lopes C, Baums IB, Sabat AM, Eirin-Lopez JM. 2020.** Genome-wide DNA methylation analysis reveals a conserved epigenetic response to seasonal environmental variation in the staghorn coral *Acropora cervicornis. Frontiers in Marine Science* **7**:822 DOI 10.3389/fmars.2020.560424.

**Rodríguez L, López C, Casado-Amezua P, Ruiz-Ramos DV, Martínez B, Banaszak A, Tuya F, García-Fernández A, Hernández M. 2019.** Genetic relationships of the hydrocoral *Millepora alcicornis* and its symbionts within and between locations across the Atlantic. *Coral Reefs* **38**:255–268 DOI 10.1007/s00338-019-01772-1.

**Rohart F, Gautier B, Singh A, Lê Cao KA. 2017.** *mixOmics*: an R package for 'omics feature selection and multiple data integration. *PLOS Computational Biology* **13**:e1005752 DOI 10.1371/journal.pcbi.1005752.

**Rossbach S, Hume BCC, Cárdenas A, Perna G, Voolstra CR, Duarte CM. 2021.** Flexibility in Red Sea *Tridacna maxima*-Symbiodiniaceae associations supports environmental niche adaptation. *Ecology and Evolution* **11**:3393–3406 DOI 10.1002/ece3.7299.

**Rowan R, Knowlton N. 1995.** Intraspecific diversity and ecological zonation in coral-algal symbiosis. *Proceedings of the National Academy of Sciences of the United States of America* **92**:2850–2853 DOI 10.1073/pnas.92.7.2850.

**Rowan R, Knowlton N, Baker A, Jara J. 1997.** Landscape ecology of algal symbionts creates variation in episodes of coral bleaching. *Nature* **388**:265–269 DOI 10.1038/40843.

**Rowan R, Powers DA. 1992.** Ribosomal RNA sequences and the diversity of symbiotic dinoflagellates (zooxanthellae). *Proceedings of the National Academy of Sciences of the United States of America* **89(8)**:3639–3643 DOI 10.1073/pnas.89.8.3639.

**Rowan R, Powers DA. 1991.** A molecular genetic classification of zooxanthellae and the evolution of animal-algal symbioses. *Science* **251(4999)**:1348–1351 DOI 10.1126/science.251.4999.1348.

**Rubin ET, Enochs IC, Foord C, Mayfield AB, Kolodziej G, Basden I, Manzello DP. 2021.** Molecular mechanisms of coral persistence within highly urbanized locations in the Port of Miami, Florida. *Frontiers in Marine Science* **8**:695236 DOI 10.3389/fmars.2021.695236.

**Russnak V, Rodriguez-Lanetty M, Karsten U. 2021.** Photophysiological tolerance and thermal plasticity of genetically different Symbiodiniaceae endosymbiont species of Cnidaria. *Frontiers in Marine Science* **8**:657348 DOI 10.3389/fmars.2021.657348.

**Saad OS, Lin X, Ng TY, Li L, Ang P, Lin S. 2020.** Genome size, rDNA copy, and qPCR assays for Symbiodiniaceae. *Frontiers in Microbiology* **11**:847 DOI 10.3389/fmicb.2020.00847.

**Sampayo EM, Dove S, LaJeunesse TC. 2009.** Cohesive molecular genetic data delineate species diversity in the dinoflagellate genus *Symbiodinium*. *Molecular Ecology* **18**:500–519 DOI 10.1111/j.1365-294X.2008.04037.x.

**Sampayo EM, Ridgway T, Bongaerts P, Hoegh-Guldberg O. 2008.** Bleaching susceptibility and mortality of corals are determined by fine-scale differences in symbiont type. *Proceedings of the National Academy of Sciences of the United States of America* **105**:10444–10449 DOI 10.1073/pnas.0708049105.

**Santos SR, Coffroth MA. 2003.** Molecular genetic evidence that dinoflagellates belonging to the genus *Symbiodinium freudenthal* are haploid. *Biological Bulletin* **204**:10–20 DOI 10.2307/1543491.

**Santos SR, Gutierrez-Rodriguez C, Coffroth MA. 2003.** Phylogenetic identification of symbiotic dinoflagellates via length heteroplasmy in domain V of chloroplast large subunit (*cp23S*)-ribosomal DNA sequences. *Marine Biotechnology* **5**:130–140 DOI 10.1007/s10126-002-0076-z.

**Santos SR, Shearer TL, Hannes AR, Coffroth MA. 2004.** Fine-scale diversity and specificity in the most prevalent lineage of symbiotic dinoflagellates (*Symbiodinium*, Dinophyceae) of the Caribbean. *Molecular Ecology* **13**:459–469 DOI 10.1046/j.1365-294x.2003.02058.x.

**Santos SR, Taylor DJ, Coffroth MA. 2001.** Genetic comparisons of freshly isolated versus cultured symbiotic dinoflagellates: implications for extrapolating to the intact symbionts. *Journal of Phycology* **37**:900–912 DOI 10.1046/j.1529-8817.2001.00194.x.

**Schoenberg DA, Trench RK. 1980.** Genetic variation in *Symbiodinium* (=*Gymnodinium*) *microadriaticum* Freudenthal, and specificity in its symbiosis with marine invertebrates. I. Isoenzyme and soluble protein patterns of axenic cultures of *Symbiodinium microadriaticum*. *Proceedings of the Royal Society of London* **207**:405–427 DOI 10.1098/rspb.1980.0031.

**Schoenberg DA, Trench RK. 1976.** Specificity of symbioses between marine cnidarians and zooxanthellae. In: Mackie GO, ed. *Coelenterate Ecology and Behavior*. Boston, MA, USA: Springer, 423–432.

**Seppey M, Manni M, Zdobnov EM. 2019.** BUSCO: assessing genome assembly and annotation completeness. In: Kollmar M, ed. *Gene Prediction: Methods and Protocols*. New York, NY: Springer, 227–245.

**Shah S, Chen Y, Bhattacharya D, Chan CX. 2020.** Sex in Symbiodiniaceae dinoflagellates: genomic evidence for independent loss of the canonical synaptonemal complex. *Scientific Reports* **10**:9792 DOI 10.1038/s41598-020-66429-4.

**Shoguchi E, Beedessee G, Tada I, Hisata K, Kawashima T, Takeuchi T, Arakaki N, Fujie M, Koyanagi R, Roy MC, Kawachi M, Hidaka M, Satoh N, Shinzato C. 2018.** Two divergent *Symbiodinium* genomes reveal conservation of a gene cluster for sunscreen biosynthesis and recently lost genes. *BMC Genomics* **19**:458 DOI 10.1186/s12864-018-4857-9.

**Shoguchi E, Shinzato C, Kawashima T, Gyoja F, Mungpakdee S, Koyanagi R, Takeuchi T, Hisata K, Tanaka M, Fujiwara M, Hamada M, Seidi A, Fujie M, Usami T, Goto H, Yamasaki S, Arakaki N, Suzuki Y, Sugano S, Toyoda A, Kuroki Y, Fujiyama A, Medina M, Coffroth MA, Bhattacharya D, Satoh N. 2013.** Draft assembly of the *Symbiodinium minutum* nuclear genome reveals dinoflagellate gene structure. *Current Biology* **23**:1399–1408 DOI 10.1016/j.cub.2013.05.062.

**Shoguchi E, Yoshioka Y, Shinzato C, Arimoto A, Bhattacharya D, Satoh N. 2020.** Correlation between organelle genetic variation and RNA editing in dinoflagellates associated with the coral *Acropora digitifera*. *Genome Biology and Evolution* **12**:203–209 DOI 10.1093/gbe/evaa042.

**Silbiger NJ, Stubler AD. 2019.** Unprofessional peer reviews disproportionately harm underrepresented groups in STEM. *PeerJ* 7:e8247 DOI 10.7717/peerj.8247.

**Silverstein RN, Correa AMS, Baker AC. 2012.** Specificity is rarely absolute in coral-algal symbiosis: implications for coral response to climate change. *Proceedings of the Royal Society B: Biological Sciences* **279**:2609–2618 DOI 10.1098/rspb.2012.0055.

**Silverstein RN, Correa AMS, LaJeunesse TC, Baker AC. 2011.** Novel algal symbiont (*Symbiodinium* spp.) diversity in reef corals of Western Australia. *Marine Ecology Progress Series* **422**:63–75 DOI 10.3354/meps08934.

**Simão FA, Waterhouse RM, Ioannidis P, Kriventseva EV, Zdobnov EM. 2015.** BUSCO: assessing genome assembly and annotation completeness with single-copy orthologs. *Bioinformatics* **31**:3210–3212 DOI 10.1093/bioinformatics/btv351.

**Smith EG, D'Angelo C, Salih A, Wiedenmann J. 2013.** Screening by coral green fluorescent protein (GFP)-like chromoproteins supports a role in photoprotection of zooxanthellae. *Coral Reefs* **32**:463–474 DOI 10.1007/s00338-012-0994-9.

**Smith EG, Gurskaya A, Hume BCC, Voolstra CR, Todd PA, Bauman AG, Burt JA. 2020.** Low Symbiodiniaceae diversity in a turbid marginal reef environment. *Coral Reefs* **39**:545–553 DOI 10.1007/s00338-020-01956-0.

**Smith EG, Ketchum RN, Burt JA. 2017.** Host specificity of *Symbiodinium* variants revealed by an ITS2 metahaplotype approach. *The ISME Journal* **11**:1500–1503 DOI 10.1038/ismej.2016.206.

**Snyder GA, Browne WE, Traylor-Knowles N, Rosental B. 2020.** Fluorescence-activated cell sorting for the isolation of scleractinian cell populations. *Journal of Visualized Experiments* **159**:e60446 DOI 10.3791/60446.

**Sproles AE, Oakley CA, Matthews JL, Peng L, Owen JG, Grossman AR, Weis VM, Davy SK. 2019.** Proteomics quantifies protein expression changes in a model cnidarian colonised by a thermally tolerant but suboptimal symbiont. *The ISME Journal* **13**:2334–2345 DOI 10.1038/s41396-019-0437-5.

**Stat M, Baker AC, Bourne DG, Correa AMS, Forsman Z, Huggett MJ, Pochon X, Skillings D, Toonen RJ, van Oppen MJH, Gates RD. 2012.** Molecular delineation of species in the coral holobiont. *Advances in Marine Biology* **63**:1–65 DOI 10.1016/B978-0-12-394282-1.00001-6.

**Stat M, Carter D, Hoegh-Guldberg O. 2006.** The evolutionary history of *Symbiodinium* and scleractinian hosts—symbiosis, diversity, and the effect of climate change. *Perspectives in Plant Ecology, Evolution and Systematics* **8**:23–43 DOI 10.1016/j.ppees.2006.04.001.

**Stat M, Morris E, Gates RD. 2008.** Functional diversity in coral-dinoflagellate symbiosis. *Proceedings of the National Academy of Sciences of the United States of America* **105**:9256–9261 DOI 10.1073/pnas.0801328105.

**Stefanoudis PV, Licuanan WY, Morrison TH, Talma S, Veitayaki J, Woodall LC. 2021.** Turning the tide of parachute science. *Current Biology* **31**:R184–R185 DOI 10.1016/j.cub.2021.01.029.

**Stephens TG, Ragan MA, Bhattacharya D, Chan CX. 2018.** Core genes in diverse dinoflagellate lineages include a wealth of conserved dark genes with unknown functions. *Scientific Reports* **8**:17175 DOI 10.1038/s41598-018-35620-z.

**Suggett DJ, Goyen S, Evenhuis C, Szabó M, Pettay DT, Warner ME, Ralph PJ. 2015.** Functional diversity of photobiological traits within the genus *Symbiodinium* appears to be governed by the interaction of cell size with cladal designation. *New Phytologist* **208**:370–381 DOI 10.1111/nph.13483.

**Suggett DJ, Moore CM, Hickman AE, Geider RJ. 2009.** Interpretation of fast repetition rate (FRR) fluorescence: signatures of phytoplankton community structure versus physiological state. *Marine Ecology Progress Series* **376**:1–19 DOI 10.3354/meps07830.

**Suggett DJ, Nitschke MN, Hughes DJ, Bartels N, Camp EF, Dilernia N, Edmondson J, Fitzgerald S, Grima A, Sage A, Warner ME. 2022.** Toward bio-optical phenotyping of reef-forming corals using light-induced fluorescence transient-fast repetition rate fluorometry. *Limnology and Oceanography, Methods* **20(3)**:131–191 DOI 10.1002/lom3.10479.

**Suggett DJ, Warner ME, Leggat W. 2017.** Symbiotic dinoflagellate functional diversity mediates coral survival under ecological crisis. *Trends in Ecology & Evolution* **32**:735–745 DOI 10.1016/j.tree.2017.07.013.

**Sunagawa S, Acinas SG, Bork P, Bowler C, Tara Oceans Coordinators, Eveillard D, Gorsky G, Guidi L, Iudicone D, Karsenti E, Lombard F, Ogata H, Pesant S, Sullivan MB, Wincker P, de Vargas C. 2020.** Tara Oceans: towards global ocean ecosystems biology. *Nature Reviews Microbiology* **18(8)**:428–445 DOI 10.1038/s41579-020-0364-5.

**Suyama Y, Matsuki Y. 2015.** MIG-seq: an effective PCR-based method for genome-wide single-nucleotide polymorphism genotyping using the next-generation sequencing platform. *Scientific Reports* **5(1)**:16963 DOI 10.1038/srep16963.

**Suzuki MM, Bird A. 2008.** DNA methylation landscapes: provocative insights from epigenomics. *Nature Reviews Genetics* **9(6)**:465–476 DOI 10.1038/nrg2341.

**Swain TD, Lax S, Backman V, Marcelino LA. 2020.** Uncovering the role of Symbiodiniaceae assemblage composition and abundance in coral bleaching response by minimizing sampling and evolutionary biases. *BMC Microbiology* **20(1)**:124 DOI 10.1186/s12866-020-01765-z.

**Sweet MJ, Scriven LA, Singleton I, Gadd M, Sariaslani S. 2012.** Chapter five—microsatellites for microbiologists. In: Gadd GM, Sariaslani S, eds. *Advances in Applied Microbiology*. Vol. 81. Cambridge: Academic Press, 169–207.

**Sweet MJ. 2014.** *Symbiodinium* diversity within *Acropora muricata* and the surrounding environment. *Marine Ecology* **35**:343–353 DOI 10.1111/maec.12092.

**Sweet M, Villela H, Keller-Costa T, Costa R, Romano S, Bourne DG, Cárdenas A, Huggett MJ, Kerwin AH, Kuek F, Medina M, Meyer JL, Müller M, Pollock FJ, Rappé MS, Sere M, Sharp KH, Voolstra CR, Zaccardi N, Ziegler M, Peixoto R. 2021.** Insights into the cultured bacterial fraction of corals. *mSystems* **6**:e01249-01220 DOI 10.1128/mSystems.01249-20.

**Taffe MA, Gilpin NW. 2021.** Racial inequity in grant funding from the US National Institutes of Health. *eLife* **10**:e65697 DOI 10.7554/eLife.65697.

**Takabayashi M, Adams LM, Pochon X, Gates RD. 2012.** Genetic diversity of free-living *Symbiodinium* in surface water and sediment of Hawai'i and Florida. *Coral Reefs* **31**:157–167 DOI 10.1007/s00338-011-0832-5.

**Takabayashi M, Santos SR, Cook CB. 2004.** Mitochondrial DNA phylogeny of the symbiotic dinoflagellates (*Symbiodinium*, Dinophyta). *Journal of Phycology* **40**:160–164 DOI 10.1111/j.0022-3646.2003.03-097.x.

**Takishita K, Ishikura M, Koike K, Maruyama T. 2003.** Comparison of phylogenies based on nuclear-encoded SSU rDNA and plastid-encoded *psbA* in the symbiotic dinoflagellate genus *Symbiodinium*. *Phycologia* **42**:285–291 DOI 10.2216/i0031-8884-42-3-285.1.

**Taubert N, Bruns A, Lenke C, Stone G. 2021.** Waiving article processing charges for least developed countries: a keystone of a large-scale open access transformation. *Insights Imaging* **34**:1–10 DOI 10.1629/uksg.526.

**Tautz D. 1989.** Hypervariability of simple sequences as a general source for polymorphic DNA markers. *Nucleic Acids Research* **17**:6463–6471 DOI 10.1093/nar/17.16.6463.

Tchernov D, Gorbunov MY, de Vargas C, Narayan Yadav S, Milligan AJ, Häggblom M, Falkowski PG. 2004. Membrane lipids of symbiotic algae are diagnostic of sensitivity to thermal bleaching in corals. *Proceedings of the National Academy of Sciences of the United States of America* **101**:13531–13535 DOI 10.1073/pnas.0402907101.

Tedersoo L, Tooming-Klunderud A, Anslan S. 2018. PacBio metabarcoding of Fungi and other eukaryotes: errors, biases and perspectives. *New Phytologist* **217**:1370–1385 DOI 10.1111/nph.14776.

Terraneo TI, Fusi M, Hume BCC, Arrigoni R, Voolstra CR, Benzoni F, Forsman ZH, Berumen ML. 2019. Environmental latitudinal gradients and host-specificity shape Symbiodiniaceae distribution in Red Sea Porites corals. *Journal of Biogeography* **46**:2323–2335 DOI 10.1111/jbi.13672.

Thornhill DJ, Howells EJ, Wham DC, Steury TD, Santos SR. 2017. Population genetics of reef coral endosymbionts (*Symbiodinium*, Dinophyceae). *Molecular Ecology* **26**:2640–2659 DOI 10.1111/mec.14055.

Thornhill DJ, LaJeunesse TC, Kemp DW, Fitt WK, Schmidt GW. 2006. Multi-year, seasonal genotypic surveys of coral-algal symbioses reveal prevalent stability or post-bleaching reversion. *Marine Biology* **148**:711–722 DOI 10.1007/s00227-005-0114-2.

Thornhill DJ, LaJeunesse TC, Santos SR. 2007. Measuring rDNA diversity in eukaryotic microbial systems: how intragenomic variation, pseudogenes, and PCR artifacts confound biodiversity estimates. *Molecular Ecology* **16**:5326–5340 DOI 10.1111/j.1365-294X.2007.03576.x.

Thornhill DJ, Lewis AM, Wham DC, LaJeunesse TC. 2014. Host-specialist lineages dominate the adaptive radiation of reef coral endosymbionts. *Evolution* **68**:352–367 DOI 10.1111/evo.12270.

Thornhill DJ, Xiang Y, Fitt WK, Santos SR. 2009. Reef endemism, host specificity and temporal stability in populations of symbiotic dinoflagellates from two ecologically dominant Caribbean corals. *PLOS ONE* **4**:e6262 DOI 10.1371/journal.pone.0006262.

Thurber RV, Payet JP, Thurber AR, Correa AMS. 2017. Virus-host interactions and their roles in coral reef health and disease. *Nature Reviews Microbiology* **15**:205–216 DOI 10.1038/nrmicro.2016.176.

Titlyanov EA, Kiyashko SI, Titlyanova TV, Yakovleva IM. 2009. δ13C and δ15N in tissues of reef building corals and the endolithic alga *Ostreobium quekettii* under their symbiotic and separate existence. *Galaxea Journal of Coral Reef Studies* **11**:169–175 DOI 10.3755/galaxea.11.169.

Tivey TR, Parkinson JE, Weis VM. 2020. Host and symbiont cell cycle coordination is mediated by symbiotic state, nutrition, and partner identity in a model cnidarian-dinoflagellate symbiosis. *mBio* **11**:e02626 DOI 10.1128/mBio.02626-19.

Torda G, Donelson JM, Aranda M, Barshis DJ, Bay L, Berumen ML, Bourne DG, Cantin N, Foret S, Matz M, Miller DJ, Moya A, Putnam HM, Ravasi T, van Oppen MJH, Thurber RV, Vidal-Dupiol J, Voolstra CR, Watson SA, Whitelaw E, Willis BL, Munday PL. 2017. Rapid adaptive responses to climate change in corals. *Nature Climate Change* **7**:627–636 DOI 10.1038/nclimate3374.

Tortorelli G, Oakley CA, Davy SK, van Oppen MJH, McFadden GI. 2022. Cell wall proteomic analysis of the cnidarian photosymbionts *Breviolum minutum* and *Cladocopium goreaui*. *Journal of Eukaryotic Microbiology* **69**:e12870 DOI 10.1111/jeu.12870.

Traylor-Knowles N. 2021. Unlocking the single-cell mysteries of a reef-building coral. *Cell* **184**:2802–2804 DOI 10.1016/j.cell.2021.05.007.

Trench RK, Blank RJ. 1987. *Symbiodinium microadriaticum* Freudenthal, *S. goreauii* sp. nov., *S. kawagutii* sp. nov. and *S. pilosum* sp. nov.: gymnodinioid dinoflagellate symbionts of marine invertebrates. *Journal of Phycology* **23**:469–481 DOI 10.1111/j.1529-8817.1987.tb02534.x.

Turland N, Kusber W-H, Wiersema JH, Barrie FR, Greuter W, Hawksworth DL, Herendeen PS, Marhold K, McNeill J. 2018. *International code of nomenclature for algae, fungi, and plants (Shenzhen Code), Regnum vegetabile*. Schmitten, Germany: Koeltz Botanical Books.

Turnham KE, Wham DC, Sampayo E, LaJeunesse TC. 2021. Mutualistic microalgae co-diversify with reef corals that acquire symbionts during egg development. *The ISME Journal* **15**:3271–3285 DOI 10.1038/s41396-021-01007-8.

Ulstrup KE, van Oppen MJH. 2003. Geographic and habitat partitioning of genetically distinct zooxanthellae (*Symbiodinium*) in *Acropora* corals on the Great Barrier Reef. *Molecular Ecology* **12**:3477–3484 DOI 10.1046/j.1365-294X.2003.01988.x.

Umeki M, Yamashita H, Suzuki G, Sato T, Ohara S, Koike K. 2020. Fecal pellets of giant clams as a route for transporting Symbiodiniaceae to corals. *PLOS ONE* **15**:e0243087 DOI 10.1371/journal.pone.0243087.

Valenzuela-Toro AM, Viglino M. 2021. How Latin American researchers suffer in science. *Nature* **598**:374–375 DOI 10.1038/d41586-021-02601-8.

van Oppen MJH, Bongaerts P, Frade P, Peplow LM, Boyd SE, Nim HT, Bay LK. 2018. Adaptation to reef habitats through selection on the coral animal and its associated microbiome. *Molecular Ecology* **27**:2956–2971 DOI 10.1111/mec.14763.

van Oppen MJH, Leong JA, Gates RD. 2009. Coral-virus interactions: a double-edged sword? *Symbiosis* **47**:1–8 DOI 10.1007/BF03179964.

van Oppen MJH, Mieog JC, Sánchez CA, Fabricius KE. 2005. Diversity of algal endosymbionts (zooxanthellae) in octocorals: the roles of geography and host relationships. *Molecular Ecology* **14**:2403–2417 DOI 10.1111/j.1365-294X.2005.02545.x.

Varasteh T, Shokri M, Rajabi-Maham H, Behzadi S, Hume B. 2018. *Symbiodinium thermophilum* symbionts in *Porites harrisoni* and *Cyphastrea microphthalma* in the northern Persian Gulf, Iran. *Journal of the Marine Biological Association of the United Kingdom* **98(8)**:2067–2073 DOI 10.1017/S0025315417001746.

Vega de Luna F, Dang KV, Cardol M, Roberty S, Cardol P. 2019. Photosynthetic capacity of the endosymbiotic dinoflagellate *Cladocopium* sp. is preserved during digestion of its jellyfish host *Mastigias papua* by the anemone *Entacmaea medusivora*. *FEMS Microbiology Ecology* **95**:fiz141 DOI 10.1093/femsec/fiz141.

Venera-Ponton DE, Diaz-Pulido G, Rodriguez-Lanetty M, Hoegh-Guldberg O. 2010. Presence of *Symbiodinium* spp. in macroalgal microhabitats from the southern Great Barrier Reef. *Coral Reefs* **29**:1049–1060 DOI 10.1007/s00338-010-0666-6.

Voolstra CR, Aranda M, Zhan Y, Dekker J. 2021a. *Symbiodinium microadriaticum* (coral microalgal endosymbiont). *Trends in Genetics* **37**:1044–1045 DOI 10.1016/j.tig.2021.08.008.

Voolstra CR, Buitrago-López C, Perna G, Cárdenas A, Hume BCC, Rädecker N, Barshis DJ. 2020. Standardized short-term acute heat stress assays resolve historical differences in coral thermotolerance across microhabitat reef sites. *Global Change Biology* **26**:4328–4343 DOI 10.1111/gcb.15148.

Voolstra CR, Quigley KM, Davies SW, Parkinson JE, Peixoto RS, Aranda M, Baker AC, Barno AR, Barshis DJ, Benzoni F, Bonito V, Bourne DG, Buitrago-López C, Bridge TCL, Chan CX, Combosch DJ, Craggs J, Frommlet JC, Herrera S, Quattrini AM, Röthig T, Reimer JD, Rubio-Portillo E, Suggett DJ, Villela H, Ziegler M, Sweet M. 2021b. Consensus guidelines for advancing coral holobiont genome and specimen voucher deposition. *Frontiers in Marine Science* **8**:1029 DOI 10.3389/fmars.2021.701784.

Voolstra CR, Suggett DJ, Peixoto RS, Parkinson JE, Quigley KM, Silveira CB, Sweet M, Muller EM, Barshis DJ, Bourne DG, Aranda M. 2021c. Extending the natural adaptive

capacity of coral holobionts. *Nature Reviews Earth & Environment* 2:747–762
DOI 10.1038/s43017-021-00214-3.

**Voolstra CR, Sunagawa S, Schwarz JA, Coffroth MA, Yellowlees D, Leggat W, Medina M. 2009.**
Evolutionary analysis of orthologous cDNA sequences from cultured and symbiotic
dinoflagellate symbionts of reef-building corals (Dinophyceae: *Symbiodinium*). *Comparative
Biochemistry and Physiology Part D Genomics and Proteomics* 4:67–74
DOI 10.1016/j.cbd.2008.11.001.

**Voolstra CR, Valenzuela JJ, Turkarslan S, Cárdenas A, Hume BCC, Perna G, Buitrago-López C,
Rowe K, Orellana MV, Baliga NS, Paranjape S, Banc-Prandi G, Bellworthy J, Fine M,
Frias-Torres S, Barshis DJ. 2021d.** Contrasting heat stress response patterns of coral holobionts
across the Red Sea suggest distinct mechanisms of thermal tolerance. *Molecular Ecology*
30:4466–4480 DOI 10.1111/mec.16064.

**Wangpraseurt D, Larkum AWD, Franklin J, Szabó M, Ralph PJ, Kühl M. 2014.** Lateral light
transfer ensures efficient resource distribution in symbiont-bearing corals. *Journal of
Experimental Biology* 217:489–498 DOI 10.1242/jeb.091116.

**Wangpraseurt D, Larkum AWD, Ralph PJ, Kühl M. 2012.** Light gradients and optical
microniches in coral tissues. *Frontiers in Microbiology* 3:316 DOI 10.3389/fmicb.2012.00316.

**Wee HB, Kobayashi Y, Reimer JD. 2021.** Effects of temperature, salinity, and depth on
Symbiodiniaceae lineages hosted by *Palythoa tuberculosa* near a river mouth. *Marine Ecology
Progress Series* 667:43–60 DOI 10.3354/meps13706.

**Wham DC, Carmichael M, LaJeunesse TC. 2014.** Microsatellite loci for *Symbiodinium goreaui*
and other Clade C *Symbiodinium*. *Conservation Genetics Resources* 6:127–129
DOI 10.1007/s12686-013-0023-5.

**Wham DC, LaJeunesse TC. 2016.** *Symbiodinium* population genetics: testing for species
boundaries and analysing samples with mixed genotypes. *Molecular Ecology* 25:2699–2712
DOI 10.1111/mec.13623.

**Wham DC, Ning G, LaJeunesse TC. 2017.** *Symbiodinium glynnii* sp. nov., a species of
stress-tolerant symbiotic dinoflagellates from pocilloporid and montiporid corals in the Pacific
Ocean. *Phycologia* 56:396–409 DOI 10.2216/16-86.1.

**Wham DC, Pettay DT, LaJeunesse TC. 2011.** Microsatellite loci for the host-generalist
"zooxanthella" *Symbiodinium trenchi* and other Clade D *Symbiodinium*. *Conservation Genetics
Resources* 3:541–544 DOI 10.1007/s12686-011-9399-2.

**Wirshing HH, Baker AC. 2016.** On the difficulty of recognizing distinct *Symbiodinium* species in
mixed communities of algal symbionts. *Molecular Ecology* 25:2724–2726
DOI 10.1111/mec.13676.

**Wirshing HH, Feldheim KA, Baker AC. 2013.** Vectored dispersal of *Symbiodinium* by larvae of a
Caribbean gorgonian octocoral. *Molecular Ecology* 22:4413–4432 DOI 10.1111/mec.12405.

**Witze ES, Old WM, Resing KA, Ahn NG. 2007.** Mapping protein post-translational modifications
with mass spectrometry. *Nature Methods* 4:798–806 DOI 10.1038/nmeth1100.

**Xiang T. 2018.** Isolation of axenic *Symbiodinium* cultures v1. protocols.io
DOI 10.17504/protocols.io.qxzdxp6.

**Xiang T, Hambleton EA, DeNofrio JC, Pringle JR, Grossman AR. 2013.** Isolation of clonal
axenic strains of the symbiotic dinoflagellate *Symbiodinium* and their growth and host
specificity. *Journal of Phycology* 49:447–458 DOI 10.1111/jpy.12055.

**Yamashita H, Koike K. 2016.** Motility and cell division patterns among several strains of
*Symbiodinium*. *Galaxea Journal of Coral Reef Studies* 18:13–19 DOI 10.3755/galaxea.18.1_13.

**Yamashita H, Koike K. 2013.** Genetic identity of free-living *Symbiodinium* obtained over a broad latitudinal range in the Japanese coast. *Phycological Research* **61**:68–80 DOI 10.1111/pre.12004.

**Yang F, Li L, Lin S. 2020.** Methylation pattern and expression dynamics of methylase and photosystem genes under varied light intensities in *Fugacium kawagutii* (Symbiodiniaceae). *Journal of Phycology* **56**:1738–1747 DOI 10.1111/jpy.13070.

**Yerbury JJ, Yerbury RM. 2021.** Disabled in academia: to be or not to be, that is the question. *Trends in Neurosciences* **44**:507–509 DOI 10.1016/j.tins.2021.04.004.

**Yorifuji M, Yamashita H, Suzuki G, Kawasaki T, Tsukamoto T, Okada W, Tamura K, Nakamura R, Inoue M, Yamazaki M, Harii S. 2021.** Unique environmental Symbiodiniaceae diversity at an isolated island in the northwestern Pacific. *Molecular Phylogenetics and Evolution* **161**:107158 DOI 10.1016/j.ympev.2021.107158.

**Zaneveld JR, McMinds R, Vega Thurber R. 2017.** Stress and stability: applying the Anna Karenina principle to animal microbiomes. *Nature Microbiology* **2(9)**:17121 DOI 10.1038/nmicrobiol.2017.121.

**Zhang C, Lin S. 2019.** Initial evidence of functional siRNA machinery in dinoflagellates. *Harmful Algae* **81(5)**:53–58 DOI 10.1016/j.hal.2018.11.014.

**Ziegler M, Eguíluz VM, Duarte CM, Voolstra CR. 2018.** Rare symbionts may contribute to the resilience of coral-algal assemblages. *The ISME Journal* **12(1)**:161–172 DOI 10.1038/ismej.2017.151.

**Ziegler M, Roder C, Büchel C, Voolstra CR. 2015.** Niche acclimatization in Red Sea corals is dependent on flexibility of host-symbiont association. *Marine Ecology Progress Series* **533**:149–161 DOI 10.3354/meps11365.

**Ziegler M, Seneca FO, Yum LK, Palumbi SR, Voolstra CR. 2017.** Bacterial community dynamics are linked to patterns of coral heat tolerance. *Nature Communications* **8(1)**:14213 DOI 10.1038/ncomms14213.