# Peer review of "Building consensus around the assessment and interpretation of Symbiodiniaceae diversity"

_PeerJ, doi:10.7717/peerj.15023_

## Round 0.1 · original submission · Minor Revisions

I am sorry for the delay in responding to you on this submission. It has been quite a challenge to find highly qualified referees who were not part of the consensus statement to review it. We eventually found 3 referees, but one has let us know they will not be able to complete the review as soon as we had hoped, so I have decided to move forward with the 2 reviews that I have in hand. As you will see, both are detailed and highly supportive of the work, but also offer a number of constructive suggestions for improvement.

In reading through the reviews and the manuscript, I feel that the referees have raised valid points and made some valuable suggestions for improvement. Thus, I request that you undertake a revision of your submission that incorporates this feedback into your revised submission.

If you decide to undertake the suggested revisions, please ensure that all review comments (including those on the annotated manuscript) are addressed in a rebuttal letter that outlines how you have addressed each comment. Any edits or clarifications mentioned in the rebuttal letter should also be inserted into the revised manuscript where appropriate. It is a common mistake to address reviewer questions in the rebuttal letter but not in the revised manuscript. If a reviewer raised a question, then your readers will probably have the same question so you should ensure that the manuscript can stand alone without the rebuttal letter. Directions on how to prepare a rebuttal letter can be found at: https://peerj.com/benefits/academic-rebuttal-letters/ if you need additional guidance.

I would also note here that it is PeerJ policy that additional references suggested during the peer-review process need only be included if the authors are in agreement that they are relevant and useful.

I look forward to seeing your revised manuscript.

Reviewer 1 ·

Basic reporting

The manuscript “Building consensus around the assessment and interpretation of Symbiodiniaceae diversity” provides an overview of the current state of Symbiodiniaceae research and provide valuable recommendation for most of the controversies in the field. Controversies in identifying the different taxonomic levels of Symbiodiniaceae and a lack of consensus among the research can lead to wrong interpretations and slow progress in the field.
This publication is extremely important as it will function as a practical guide for research done in the field. The authors not just discuss the challenges and unresolved issues in Symbiodiniaceae genomics but also recommend and provide guidelines to increase diversity in the field.

Here are a few comments and suggestions:

201: There are tools to assist in the distinction (e.g. Symportal). I believe it would be worth mentioning some of them here.

258-261: Can you be more specific to whom these recommendations are for? I believe by pointing out who these recommendations are addressed for can make them more effective. Most of the recommendations listed in the paragraph are addressed to journals: increase diversity in editorial board, offering lower publication costs for certain countries, double blind review process, …

2687-270: Instead of recommend stablishing a database, I believe the authors should mention that the database has been started (wouldn’t it be the link in the line 1459)? If so, it should be mentioned. Otherwise, as it is written, it suggests the database does not exist.

309-314: In my opinion, the statement is not problematic per se if the sentence is accompanied by a another saying that these symbiont types belong to the same species. If the authors disagree, I suggest them to expand on this.

319-325: While morphological and ecological descriptions are listed in line 335, I wondered why they are not listed as one of the components for a valid species description. Are these 6 major components defined elsewhere? If so, a reference is necessary. Otherwise, I would like the authors to explain why morphological and ecological descriptions are not include in the list of what is needed for a valid description.

371: The authors mentioned that guidance for genomic data set has recently been put forth. I believe it would be beneficial to listing some of them in the text.

374-376: While I agree that adopting a multi gene approach improve resolution and accuracy of species identification, many researchers may not have the resources (time and funding) to run multiple markers on each sample. This is particularly challenging for researchers from developing countries, where next generation sequencing is still very costly and adding multiple markers might significantly affect the overall research budget. I think it is something that is worth mentioning and should be considered when providing guidance on the number of markers.

The spacing needs to be standardized across the manuscript. Check line 387, 347-350. Also, in line 884-938 spacing among lines decreases.

404: For coralimorphorpharians, a recent publication could be added: Jacobs, K.P., Hunter, C.L., Forsman, Z.H. et al. A phylogenomic examination of Palmyra Atoll’s corallimorpharian invader. Coral Reefs 41, 673–685 (2022). https://doi.org/10.1007/s00338-021-02143-5

419-468: One thing worth adding in this session is some recommendation on markers or approaches to confirm a culture is actually monoculture (or a sentence to direct the reader to that session of the manuscript).

758: There is value in identifying symbionts at different taxonomic levels other than species as there is so much we do not know about the potential physiological role of symbiont subspecies. De Souza, et al.(2022) for example found significant pattern of symbiont community according to environmental conditions. If the symbionts had been identified to the level of species only I believe about 4 species would be identified (the authors did not discuss this) and the pattern of geographical and environmental subtypes and types would not be identified. While identifying to the levels below the species level can lead to overestimation of diversity, contrary to what the sentence said in 758, I believe that identifying symbiont at the level of species only can also obscure patterns. In my opinion, we should not discourage the identification both at the species level, above and below; instead, I think we should recommend authors to always clearly state what level of taxonomic identification they are using, and what they mean when they use the term diversity in their studies. Either approach is important, but it is also important to explicitly state which is being used, and to interpret the data and make the conclusions based on the level of taxonomy being identified.

908- Exclude the parenthesis inside the parenthesis, similar reference in the text have been added without the parenthesis (as in line 1022). Additionally, I would exclude the “for each” in the end or rephrase.

1353-1387: I agree with all points. I would also add the importance of fostering diversity of career levels. Including diverse early career students in workshops and discussions (including workshops similar to the one that led to this manuscript) is a great opportunity for the student to learn, engage in discussions and network with many coral researchers in the field and can have huge impacts in their career.

1466:1468- I agree with the importance of having hybrid conferences to increase participation. One point worth mentioning about the planning of hybrid conferences is to be mindful of different time zones. For a hybrid meeting to be inclusive, considering the time zone of the participants is important so that they feel respected and there is no expectation of attending talks at extremely early or late hours. The virtual ICRS in 2021, while more inclusive due to the online component, was very hard to follow for many participants due to the time zone.

Experimental design

no comment

Validity of the findings

no comment

Additional comments

no comment

·

Basic reporting

The article is well written, in clear scientific English, with sufficient background, context and references provided. The structure of the article is professional, with relevant figures and tables as well as two highlight "box" texts.

The review will primarily be of interest to those actively studying Symbiodinaceae and their hosts. However, given the ecological importance of the group and the challenges present in their description and identification, I believe the consensus statements here will be useful to a broad swath of research in related fields.

There has not been a recent review of the field of this scope, but more importantly this has been an attempt at a consensus statement from a historically contentious field.

The introduction covers both the basics of the field and the rationale for this important consensus effort.

Experimental design

The study is within scope, with rigorous presentation of useful detail.

As in any review, there are areas that skim and areas that dive deeper, but overall the review achieves broad coverage at comparable depth.

Both referenced and structure are appropriate.

Validity of the findings

As is likely in a consensus document, I found very little that is controversial here.
On the contrary, there are multiple sections in the document where a researcher chasing a particular question could find a thoughtful presentation of methodological strengths and weaknesses which could guide the selection of robust methods. It was satisfying to read, and I would certainly recommend it as a starting place for those either beginning with Symbiodiniaceae studies or those facing particular challenges in their current approach.

The conclusion does not specifically identity the "unresolved questions/ gaps/ future directions", but given the scope of the text there are many, many such areas identified in the body of the document, and I do not fault the authors for using the conclusion to summarize the entire endeavor.

Additional comments

As should be apparent from my comments above, I found the consensus review a satisfying document that was a long time coming, and have no major "deal-breaker" comments to share.

That said, I think we should not understate the importance of the included tables (and expanded versions of them) as 'rosetta stones' to translate among various marker results/designations. This is especially true in integrating the results of more taxonomically thorough work to guide the more phenomenological survey/experimental/ works that often are running many more samples. In some cases the prior source of conflicts in the field have been methodological, and having the ability to robustly translate from psbA sequences, to ITS2 types, to described species could go a long way to avoid this pitfall in the future, while maintaining the ability to efficiently identify Symbiodiniaceae present in many samples. I applaud the steps to simplify this process here, but would encourage associated researchers to expand on this effort.

My only comment would be to note that while there was a reference to the potential utility of genomic signatures of selection, without reference to any publication that employed them in the Symbiodiniaceae. Here I include references to two such studies that may be of interest to the readership.
(I also note after checking, that the first of these is indeed cited for other aspects of the paper, but not specifically in the section covering genomic patterns of selection).

(1) Liu, Huanle, Timothy G. Stephens, Raúl A. González-Pech, Victor H. Beltran, Bruno Lapeyre, Pim Bongaerts, Ira Cooke et al. "Symbiodinium genomes reveal adaptive evolution of functions related to coral-dinoflagellate symbiosis." Communications biology 1, no. 1 (2018): 1-11.

(2) Ladner, Jason T., Daniel J. Barshis, and Stephen R. Palumbi. "Protein evolution in two co-occurring types of Symbiodinium: an exploration into the genetic basis of thermal tolerance in Symbiodiniumclade D." BMC evolutionary biology 12, no. 1 (2012): 1-13.

---

## Round 0.2 · accepted · Accept

I have heard back from the most critical of the initial referees who is satisfied with the revisions and agrees that the paper is ready for publication without further comment. Thank you for your thorough and detailed response to the referees. I am happy to move this important contribution forward into production.